# On the Predictive Accuracy of Neural Temporal Point Process Models for Continuous-time Event Data

**Tanguy Bosser**                                    *tanguy.bosser@umons.ac.be*
*Department of Computer Science*
*University of Mons*

**Souhaib Ben Taieb**                               *souhaib.bentaieb@umons.ac.be*
*Department of Computer Science*
*University of Mons*

Reviewed on OpenReview: *https://openreview.net/forum?id=3OSISBQPrM*

## Abstract

Temporal Point Processes (TPPs) serve as the standard mathematical framework for modeling asynchronous event sequences in continuous time. However, classical TPP models are often constrained by strong assumptions, limiting their ability to capture complex real-world event dynamics. To overcome this limitation, researchers have proposed Neural TPPs, which leverage neural network parametrizations to offer more flexible and efficient modeling. While recent studies demonstrate the effectiveness of Neural TPPs, they often lack a unified setup, relying on different baselines, datasets, and experimental configurations. This makes it challenging to identify the key factors driving improvements in predictive accuracy, hindering research progress. To bridge this gap, we present a comprehensive large-scale experimental study that systematically evaluates the predictive accuracy of state-of-the-art neural TPP models. Our study encompasses multiple real-world and synthetic event sequence datasets, following a carefully designed unified setup. We thoroughly investigate the influence of major architectural components such as event encoding, history encoder, and decoder parametrization on both time and mark prediction tasks. Additionally, we delve into the less explored area of probabilistic calibration for neural TPP models. By analyzing our results, we draw insightful conclusions regarding the significance of history size and the impact of architectural components on predictive accuracy. Furthermore, we shed light on the miscalibration of mark distributions in neural TPP models. Our study aims to provide valuable insights into the performance and characteristics of neural TPP models, contributing to a better understanding of their strengths and limitations.

## 1 Introduction

From human social activity to natural phenomena, the evolution of a system of interest can often be characterized by a sequence of discrete events occurring at irregular time intervals. Online shop-

ping activity (Cai et al., 2018), earthquake occurrences (Ogata, 1998), measurement of electronic health records (Wang et al., 2016), and users activity on social media (Farajtabar et al., 2015) are typical examples where such sequences are frequently encountered. Given a sequence of observed historical events, a crucial challenge in numerous applications is to predict the *timing* of future events. Additionally, in cases where events are assigned a label, referred to as *marks*, it is necessary to also predict the *type* of event that is likely to occur. It is reasonable to assume that events are interdependent and that the future evolution of a system is directly influenced by past occurrences. For example, an individual might be inclined to purchase a particular item at a specific time on an e-commerce platform solely because a previous purchase made it necessary. Hence, modeling the intricate dynamics of event occurrences becomes crucial in predicting future events based on past observations.

Drawing upon solid theoretical foundations, the framework of Temporal Point Processes (TPPs) (Daley & Vere-Jones, 2007) has established itself as a suitable choice for modeling these sequences of asynchronous and time-dependent data. A TPP is fully characterized by its conditional intensity function, which provides the instantaneous unit rate of event arrivals based on the process history (Rasmussen, 2018). For event sequence modeling in a variety of domains, including finance (Hawkes, 2018), crime (Egesdal et al., 2010), or epidemiology (Rizoiu et al., 2018), a large number of earlier works examined classical parametric forms of TPPs, such as the Hawkes process (Hawkes, 1971). However, these classical parameterizations have faced criticism due to their limited flexibility in capturing complex event dynamics (Mei & Eisner, 2016). To address this, deep learning techniques have been introduced into the TPP literature, enabling more flexible models capable of capturing more complex temporal dependencies. Examples include recurrent neural networks (RNNs) (Mei & Eisner, 2016; Shchur et al., 2019) and self-attention mechanisms (Zhang et al., 2020; Zuo et al., 2020; Enguehard et al., 2020). Since its introduction by Du et al. (2016), the field of neural TPPs has experienced rapid development, with the emergence of numerous novel architectures and applications (Shchur et al., 2021).

Given a sequence of events, neural TPP models typically involve a combination of three main architectural components: 1) an event encoder, creating a fixed-sized embedding for each event in the sequence. 2) a history encoder, generating a summary of the history from past events' embeddings, and 3) a decoder parametrizing a function that fully characterizes the distribution of future events' arrival times and marks. Among other possibilities, improvements with respect to existing baselines are obtained by proposing alternatives to either of these components. For instance, one can replace an RNN-based history encoder with a self-attentive one, or choose to parametrize a certain TPP function that leads to useful properties, such as reduced computational costs or closed-form sampling. However, as pointed out by Shchur et al. (2021), "*new architectures often change all these components at once, which makes it hard to pinpoint the source of empirical gains*". Moreover, the baselines against which a newly proposed architecture is compared, as well as the datasets employed and the experimental setups, often differ from paper to paper, which renders a fair comparison even harder.

Additionally, modeling marked TPPs from data is a challenging problem from a statistical perspective in the sense that it requires joint modeling of discrete distributions over marks and continuous distributions over time. In practice, a model's ability to estimate the joint distribution of future arrival times and marks is often solely evaluated on its performance with respect to the negative log-likelihood (NLL). However, reporting a single NLL value encompasses the contributions of both arrival time and mark distributions, making it hard to evaluate model performance with respect to

each prediction task separately. Moreover, while the NLL enables the comparison of different baselines, it provides a limited understanding of the distribution shapes and the allocation of probability mass across their respective domains. Furthermore, while probabilistic calibration is a desirable property that any competent or ideal predictive distribution should possess (Dheur & Ben Taieb, 2023; Guo et al., 2017), the assessment of calibration in TPP models has been overlooked by the neural TPP community, both for the predictive distributions of time and marks.

In this paper, our objective is to address the aforementioned concerns by presenting the following contributions:

- We perform a large-scale experimental study to assess the predictive accuracy of state-of-the-art neural TPP models on 15 real-world event sequence datasets in a carefully designed unified setup. Our study also includes classical parametric TPP models as well as synthetic datasets. In particular, we study the influence of each major architectural component (event encoding, history encoder, decoder parametrization) for both time and mark prediction tasks. To the best of our knowledge, this is the largest comprehensive study for neural TPPs to date. Our study is fully reproducible and implemented in a common code base[1].

- We assess the probabilistic calibration of neural TPP models, both for the time and mark predictive distributions. To this end, we employ standard metrics and tools borrowed from the forecasting literature, namely the probabilistic calibration error and reliability diagrams. Probabilistic calibration is a desirable property that any competent or ideal predictive distribution should possess. Yet, to the best of our knowledge, this has been generally overlooked by the neural TPP community.

- Among other findings, we found that neural TPP models often do not fully leverage the complete information contained in all historical events. In fact, relying solely on a subset of the most recent observed occurrences can yield comparable performance to encoding the entire historical context. Furthermore, we demonstrate the high sensitivity of various decoder parameterizations to the event encoder, highlighting the significant gains in predictive accuracy that can be achieved through appropriate selection. In addition, while the distribution of arrival times is generally well-calibrated, our research reveals that classical parametric baselines exhibit better calibration of mark distributions compared to neural TPP models. Lastly, our study shows that several commonly used event sequence datasets within the TPP literature may not be suitable for accurately benchmarking neural TPP baselines.

## 2  Background and Notations

**Marked Temporal Point Processes** are stochastic processes whose realizations consist in a sequence of $n$ discrete events $\mathcal{S} = \{(t_1, k_1), ..., (t_n, k_n)\}$ observed within a fixed window $[0, T]$ with $T > 0$. Each event $e_i = (t_i, k_i)$ in the sequence represents an **arrival time** $t_i$ for the $i^{th}$ event, satisfying $0 \leq t_1 < ... < t_n \leq T$, and is associated with a mark $k_i$ belonging to the mark space $\mathcal{K}$. The mark space $\mathcal{K}$ can either be discrete, such as $\mathcal{K} = \{1, ..., K\}$ (Du et al., 2016), or continuous, i.e. $\mathcal{K} = \mathbb{R}$. Note that within this definition, the number of events $n$ itself is a random variable.

---

[1]https://github.com/tanguybosser/ntpp-tmlr2023

In the case of an unmarked scenario, events are solely characterized by their arrival times, i.e. $S = \{t_1, ..., t_n\}$. Without loss of generality, we will primarily focus on the marked setting for the remainder of this section, exclusively considering univariate discrete marks throughout the paper. A realization of a marked TPP can also be represented using a counting process $N(t) = \sum_{k=1}^{K} N_k(t)$, where $N_k(t) = |\{(t_i, k_i) \in \mathcal{S} | t_i < t\}|$ is the number of events with mark $k$ that have occurred prior to time $t$.

Let $e_{i-1} = (t_{i-1}, k_{i-1})$ be the last observed event. The occurrence of the next event $e_i$ in $(t_{i-1}, \infty[$ can be characterized by the conditional joint distribution of arrival-times and marks $f(t, k|\mathcal{H}_t)$, where $\mathcal{H}_t = \{(t_j, k_j) \in \mathcal{S} \mid t_j < t\}$ denotes the observed process' history. The conditional joint distribution can be decomposed as $f(t, k|\mathcal{H}_t) = f(t|\mathcal{H}_t)p(k|t, \mathcal{H}_t)$, where $f(t|\mathcal{H}_t)$ corresponds to the conditional density of time $t$, while $p(k|t, \mathcal{H}_t)$ is the probability of observing mark $k$, conditional on both $t$ and $\mathcal{H}_t$. Given the conditional cumulative distribution of arrival-times $F(t|\mathcal{H}_t)$, a marked TPP can be equivalently characterized by the *mark-wise conditional intensity function* (MCIF) $\lambda_k(t|\mathcal{H}_t)$, i.e:

$$\lambda_k(t|\mathcal{H}_t) = \frac{f(t, k|\mathcal{H}_t)}{1 - F(t|\mathcal{H}_t)}. \tag{1}$$

Furthermore, for $t > t_{i-1}$, we have (Rasmussen, 2018):

$$\lambda_k(t|\mathcal{H}_t) = \frac{f(t, k|\mathcal{H}_t)}{1 - F(t|\mathcal{H}_t)} = \lambda(t|\mathcal{H}_t)p(k|t, \mathcal{H}_t) \tag{2}$$

$$= \frac{\mathbb{E}\left[N_k(t + dt) - N_k(t)\right]|\mathcal{H}_t, t_i \notin [t_{i-1}, t]]}{dt}, \tag{3}$$

where $\lambda(t|\mathcal{H}_t) = \sum_{k=1}^{K} \lambda_k(t|\mathcal{H}_t)$ is the ground conditional intensity function of the process (GCIF). In essence, the MCIF gives the expected instantaneous occurrence rate of an event of mark $k \in \mathcal{K}$. Furthermore, as a heuristic interpretation, the MCIF can be seen as the instantaneous "probability" of observing the next event of mark $k$ in an infinitesimal interval around $t$, conditional on the past of the process up to but not including $t$:

$$\lambda_k(t|\mathcal{H}_t)dt \simeq \mathbb{P}\left[t_i \in [t, t + dt], k_i = k|\mathcal{H}_t, t_i \notin [t_{i-1}, t]\right]. \tag{4}$$

In the following, we will employ the notation '$*$' of Daley & Vere-Jones (2007) to indicate dependence on $\mathcal{H}_t$, i.e. $\lambda_k^*(t) = \lambda_k(t|\mathcal{H}_t)$. From the MCIF, we can define the *cumulative MCIF*,

$$\Lambda_k^*(t) = \int_{t_{i-1}}^{t} \lambda_k^*(s)ds, \tag{5}$$

allowing us to define the *conditional joint density* of time $t$ and mark $k$,

$$f^*(t, k) = \lambda_k^*(t)\exp\left(-\sum_{k=1}^{K} \Lambda_k^*(t)\right). \tag{6}$$

The proof of (6) can be found in Appendix E. Additionally, note that a realization of a marked TPP can be equivalently represented by the sequence $\{(\tau_1, k_1), ..., (\tau_n, k_n)\}$ where $\tau_1 = t_1$ and $\tau_i = t_i - t_{i-1}$, for $i > 1$, are the *inter-arrival times* associated to the events. We will use both notations interchangeably throughout the paper. Lastly, all expressions will be defined for $t_{i-1} < t \leq t_i$, unless stated otherwise.

**Parametrization of TPP models and Learning.** Defining a marked TPP model usually involves specifying a parametric form for any of the functions in (2), (5) or (6), provided that the chosen parametrization defines a valid joint distribution of arrival times and marks, $f^*(t, k)$. In this regard, a distinct set of requirements need to be fulfilled depending on the chosen function. We consider exclusively the setting of *non-terminating* TPP, meaning that a future event will arrive eventually with probability one. For a non-terminating TPP, if one decides to parametrize the MCIF, then the following conditions must be satisfied (Rasmussen, 2018):

$R_1^\lambda$: $\lambda_k^*(t) \geq 0$. $\qquad\qquad\qquad\qquad\qquad$ $R_2^\lambda$: $\lim_{t\to\infty} \int_{t_{i-1}}^t \lambda_k^*(s)ds = \infty$.

Alternatively, parametrizing the conditional joint density implies:

$R_1^f$: $f(t, k) > 0$. $\qquad\qquad\qquad\qquad\qquad$ $R_2^f$: $\int_{t_{i-1}}^\infty \sum_{k=1}^K f(t, k)dt = 1$.

Specifying a parametrization of the cumulative MCIF involves four constraints (Enguehard et al., 2020):

$R_1^\Lambda$: $\Lambda_k^*(t) > 0$. $\qquad\qquad\qquad\qquad\qquad$ $R_3^\Lambda$: $\lim_{t\to\infty} \Lambda_k^*(t) = \infty$.

$R_2^\Lambda$: $\Lambda_k^*(t_{i-1}) = \int_{t_{i-1}}^{t_{i-1}} \lambda_k^*(s)ds = 0$. $\qquad\qquad$ $R_4^\Lambda$: $d\Lambda_k^*(t)/dt \geq 0$,

where conditions $R_1^\Lambda$ and $R_4^\Lambda$ result from $\lambda_k^*(t) \geq 0$, while $R_3^\Lambda$ directly follows $R_2^\lambda$. In the setting of *terminating* TPP, where there is a probability $\pi$ that the process stops after the last observed event, $R_2^f, R_2^\lambda$ and $R_3^\Lambda$ are no longer satisfied.

A common example of parametrization is the homogeneous Poisson process (Daley & Vere-Jones, 2007; De et al., 2019) that defines a constant MCIF:

$$\lambda_k^*(t) = \mu_k \geq 0, \tag{7}$$

and hence implicitly assumes an exponential distribution with rate $\mu = \sum_{k=1}^K \mu_k$ for the inter-arrival times $\tau$:

$$f^*(\tau) = \mu \exp(-\mu\tau). \tag{8}$$

Another example is the well-known Hawkes process (Hawkes, 1971; Liniger, 2009), which defines the MCIF to account for the influence of previous events on the process' dynamics:

$$\lambda_k^*(t) = \mu_k + \sum_{k'=1}^K \sum_{\{(t_i, k'):t_i < t\}} \phi_{k,k'}(t - t_i), \tag{9}$$

where $\phi_{k,k'} : \mathbb{R}^+ \to \mathbb{R}^+$ is the so-called triggering kernel. For instance, choosing the exponential kernel $\phi_{k,k'}(t - t_i) = \alpha_{k,k'} \exp\left(-\beta_{k,k'}(t - t_i)\right) \mathbb{1}[t - t_i \geq 0]$ with $\alpha_{k,k'} > 0$ and $\beta_{k,k'} > 0$ allows to explicitly model self-excitation dynamics, for which the occurrence of an event increases the probability of observing future events, in a positive, additive and exponentially decaying fashion.

Let $\boldsymbol{\theta}$ be the set of learnable parameters for a valid parametrization of a marked TPP function. As an example, for the Hawkes process in (9), we have $\boldsymbol{\theta} = \{\boldsymbol{\mu}, \boldsymbol{\alpha}, \boldsymbol{\beta}\}$, where $\boldsymbol{\mu} \in \mathbb{R}_+^K$ and $\boldsymbol{\alpha}, \boldsymbol{\beta} \in \mathbb{R}_+^{K \times K}$.

The most common approach to learning $\boldsymbol{\theta}$ is achieved by maximum likelihood estimation, i.e. by minimizing the negative log-likelihood (NLL). Given a parametric form of $\lambda_k^*(t; \boldsymbol{\theta})$, $f^*(t, k; \boldsymbol{\theta})$ or $\Lambda_k^*(t; \boldsymbol{\theta})$ for all $k \in \mathcal{K}$, and a sequence $\mathcal{S}$ of $n$ events observed on $[0, T]$, the NLL objective writes:

$$\mathcal{L}(\boldsymbol{\theta}; \mathcal{S}) = -\sum_{i=1}^{n} \left[ \log \lambda_{k_i}^*(t_i; \boldsymbol{\theta}) - \Lambda^*(t_i; \boldsymbol{\theta}) \right] + \Lambda^*(T; \boldsymbol{\theta}) \tag{10}$$

$$= -\sum_{i=1}^{n} \left[ \log f^*(t_i; \boldsymbol{\theta}) + \log p^*(k_i | t_i; \boldsymbol{\theta}) \right] + \Lambda^*(T; \boldsymbol{\theta}), \tag{11}$$

where the term $\Lambda^*(T; \boldsymbol{\theta}) = \sum_{k=1}^{K} \Lambda_k^*(T; \boldsymbol{\theta})$ accounts for the fact that no event has been observed in the interval $(t_n, T]$.

**Prediction tasks with marked TPPs**. The expression (11) presented above highlights that learning a marked TPP model from an event sequence involves two main estimation tasks: (**T1**) estimating the conditional density of arrival times $f^*(t)$, and (**T2**) estimating the conditional distribution of marks $p^*(k|t)$. Once we have estimates for $f^*(t)$ and $p^*(k|t)$, we can effectively address queries such as: *When is the next event likely to occur? What will be the type of the next event, given that it occurs at a certain time t? How long until an event of type k occurs?* .

Additionally, by utilizing $f^*(t; \boldsymbol{\theta})$ and $p^*(k|t; \boldsymbol{\theta})$, we can compute an estimate for the next expected arrival time (**T3**) as:

$$\tilde{t} = \mathbb{E}[t] = \int_{t_{i-1}}^{\infty} t f^*(t; \boldsymbol{\theta}) dt, \tag{12}$$

and estimate the mark of the event at time $t$ (**T4**) as:

$$\tilde{k} = \underset{k \in \mathcal{K}}{\operatorname{argmax}} \, p^*(k|t; \boldsymbol{\theta}) = \underset{k \in \mathcal{K}}{\operatorname{argmax}} \, \frac{\lambda_k^*(t; \boldsymbol{\theta})}{\lambda^*(t; \boldsymbol{\theta})} = \underset{k \in \mathcal{K}}{\operatorname{argmax}} \, \lambda_k^*(t; \boldsymbol{\theta}). \tag{13}$$

In this paper, our focus will be exclusively on **T1**, which corresponds to the *time prediction* task, as well as **T2** and **T4**, which are referred to as the *mark prediction* tasks. We will not generate point estimates for the next expected arrival time (**T3**) since most of the considered models do not allow for estimating (12) in closed form. To maintain clarity in notation, we will omit the explicit dependency of the parametrizations on $\boldsymbol{\theta}$ throughout the remainder of the paper.

## 3 Neural Temporal Point Processes

While simple parametric forms of marked TPPs, such as the Hawkes process, have been used to model event sequences in a variety of real-world scenarios, they have been criticized for their lack of flexibility. For instance, events may inhibit the occurrences of future events rather than excite them, or their influence might not be strictly additive. While previous works proposed modifications of the Hawkes process to encompass such complementary effects, e.g. excitation and inhibition (Chen et al., 2019; Costa et al., 2020; Duval et al., 2022), others relied on the expressiveness of neural networks to capture more complex event dynamics Mei & Eisner (2016). The resulting framework, called *Neural* TPPs, defines models that can be characterized by three major components (Shchur et al., 2021). Recall that $\mathcal{S} = \{e_1, ..., e_n\}$, where $e_i = (t_i, k_i)$. A neural TPP model involves:

1. An **event encoder** which, for each $e_i \in \mathcal{S}$, generates a fixed size embedding[2] $\boldsymbol{e}_i \in \mathbb{R}^{d_e}$;

2. A **history encoder**, which for each $e_i \in \mathcal{S}$, generates a fixed size history embedding $\boldsymbol{h}_i \in \mathbb{R}^{d_h}$ from past event representations $\{\boldsymbol{e}_1, ..., \boldsymbol{e}_{i-1}\}$;

3. A **decoder**, which given a query time $t$ and its associated embedding $\boldsymbol{e}^t$ for $t_{i-1} \leq t < t_i$, parametrizes a function characterizing the TPP (i.e. $\lambda_k^*(t)$, $\Lambda_k^*(t)$, or $f^*(t, k)$) using $\boldsymbol{e}^t$, and/or $\boldsymbol{h}_i$.

The general modeling pipeline is shown in Figure 1. In the following, we will give more details about each of its major components.

### 3.1 Event encoding

The task of event encoding consists in generating a representation $\boldsymbol{e}_i \in \mathbb{R}^{d_e}$ for each event in $\mathcal{S}$, to be fed to the history encoder, and eventually to the decoder, at a later stage. This task essentially boils down to finding an embedding $\boldsymbol{e}_i^t \in \mathbb{R}^{d_t}$ for the (inter-)arrival time $t_i$, and an embedding $\boldsymbol{e}_i^k \in \mathbb{R}^{d_k}$ for the associated mark $k_i$. The event embedding $\boldsymbol{e}_i$ is finally obtained through some combination of $\boldsymbol{e}_i^t$ and $\boldsymbol{e}_i^k$, e.g. via concatenation:

$$\boldsymbol{e}_i = \begin{bmatrix} \boldsymbol{e}_i^t \\ \boldsymbol{e}_i^k \end{bmatrix}. \tag{14}$$

**Encoding the (inter-)arrival times**. A straightforward approach to obtain $\boldsymbol{e}_i^t$ is to select the raw inter-arrival times as time embeddings $\boldsymbol{e}_i^t = \tau_i$ or their logarithms $\boldsymbol{e}_i^t = \log \tau_i$ (Omi et al., 2019; Shchur et al., 2019; Mei & Eisner, 2016; Du et al., 2016). Inspired by the positional encoding of Transformer architectures (Vaswani et al., 2017) and their extension to temporal data (Kazemi et al., 2019), other works exploit more expressive representations by encoding the arrival times as vectors of sinusoïdals (Enguehard et al., 2020):

$$\boldsymbol{e}_i^t = \bigoplus_{j=0}^{d_t/2-1} \sin\left(\alpha_j t_i\right) \oplus \cos\left(\alpha_j t_i\right), \tag{15}$$

where $\alpha_j \propto 1000^{\frac{-2j}{d_t}}$ and $\oplus$ is the concatenation operator. Variants of sinusoïdal encoding can also be found in Zhang et al. (2020) and Zuo et al. (2020). Alternatively, learnable embeddings can be obtained by feeding the inter-arrival times to a fully connected layer (FC) (Enguehard et al., 2020):

$$\boldsymbol{e}_i^t = \text{FC}\left(\tau_i\right), \tag{16}$$

where $\text{FC} : \mathbb{R}^+ \to \mathbb{R}^{d_t}$.

**Encoding the mark**: When marks are available, the most common approach to obtain the mark embeddings is achieved by specifying a learnable embedding matrix $\mathbf{E}^k \in \mathbb{R}^{d_k \times K}$ (Du et al., 2016; Mei & Eisner, 2016; Shchur et al., 2019; Zhang et al., 2020; Zuo et al., 2020; Enguehard et al., 2020). Given the one-hot encoding $\boldsymbol{k}_i \in \{0, 1\}^K$ of mark $k_i$, its embedding $\boldsymbol{e}_i^k \in \mathbb{R}^{d_k}$ is retrieved as

$$\boldsymbol{e}_i^k = \mathbf{E}^k \boldsymbol{k}_i. \tag{17}$$

---

[2]Note that $e_i$ is used to designate an event, while the bold notation $\boldsymbol{e}_i$ is used for the event representation.

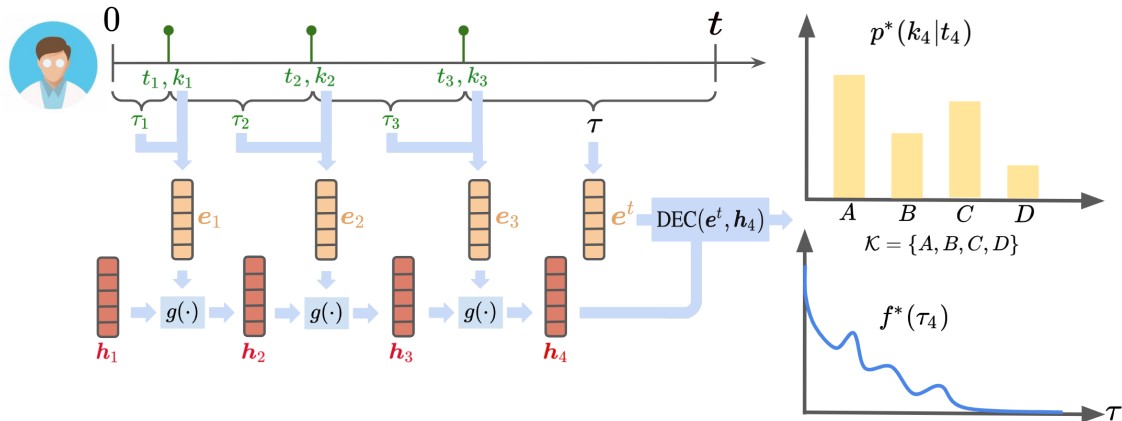

Figure 1: General workflow of neural parametrizations for marked TPPs. Events inter-arrival times $\tau_i$ and marks $k_i$ are encoded into a vector $\boldsymbol{e}_i$, which is in turn used to generate the history embeddings $\boldsymbol{h}_i$ through some form of auto-regressive mechanisms $g(\cdot)$ (e.g. GRU). A query-time embedding $\boldsymbol{e}^t$ and the history embeddings are then fed to the decoder to estimate the conditional density $f^*(\tau_{i+1})$ and conditional distribution $p^*(k_{i+1}|t_{i+1})$ of the next event.

### 3.2   History encoding

The general principle behind history encoding in the context of neural TPP approaches is to construct a fixed-size embedding $\boldsymbol{h}_i \in \mathbb{R}^{d_h}$ for event $e_i$ from its sequence of past events representations $\{\boldsymbol{e}_1, ..., \boldsymbol{e}_{i-1}\}$, using some form of auto-regressive mechanism or set aggregator. Naturally, the main goal when constructing $\boldsymbol{h}_i$ is to capture as closely as possible relevant patterns in the process history, in order to accurately estimate the distributions of arrival times and marks of future events.

**Recurrent architectures.** A natural choice to handle ordered sequences of tokens is achieved by employing a form of recurrent architecture, such as an LSTM or a GRU (Shchur et al., 2019; Du et al., 2016; Mei & Eisner, 2016). In this context, starting from an initial state $\boldsymbol{h}_1$ initialized randomly, the history embedding $\boldsymbol{h}_i$ of an event $e_i$ is constructed by sequentially updating the history embeddings at the previous time steps by using the next event's representation:

$$\boldsymbol{h}_2 = g(\boldsymbol{h}_1, \boldsymbol{e}_1),$$
$$...$$
$$\boldsymbol{h}_i = g(\boldsymbol{h}_{i-1}, \boldsymbol{e}_{i-1}), \tag{18}$$

where $\boldsymbol{h}_1$ is the initial state and $g : \mathbb{R}^{d_h} \to \mathbb{R}^{d_h}$ refers to the update function of the chosen recurrent layer.

**Self-attention encoders.** As an alternative to recurrent architectures, the self-attention mechanism of Transformers (Vaswani et al., 2017) computes the $\boldsymbol{h}_i$'s for each $i$ independently as follows

(Zhang et al., 2020; Zuo et al., 2020; Enguehard et al., 2020):

$$\boldsymbol{h}_i = \text{SA}(\boldsymbol{q}_i, \mathbf{K}_i, \mathbf{V}_i) = \mathbf{W}_2 \sigma_R \big(\mathbf{W}_1 \hat{\boldsymbol{h}}_i^\top + \boldsymbol{b}_1 \big) + \boldsymbol{b}_2, \tag{19}$$

$$\hat{\boldsymbol{h}}_i = \text{Softmax}\Big(\frac{\boldsymbol{q}_i^\top \mathbf{K}_i}{\sqrt{d_q}}\Big) \mathbf{V}_i^\top, \tag{20}$$

where $\text{SA}(\cdot)$ refers to the self-attention mechanism between a *query* vector $\boldsymbol{q}_i$, a *key* matrix $\mathbf{K}_i$ and a *value* matrix $\mathbf{V}_i$ defined as

$$\boldsymbol{q}_i = \mathbf{W}_Q \boldsymbol{e}_{i-1}, \quad \mathbf{K}_i = \mathbf{W}_K[\boldsymbol{e}_1, ..., \boldsymbol{e}_{i-1}], \quad \mathbf{V}_i = \mathbf{W}_V[\boldsymbol{e}_1, ..., \boldsymbol{e}_{i-1}], \tag{21}$$

where $\mathbf{W}_Q, \mathbf{W}_K \in \mathbb{R}^{d_q \times d_e}$, $\mathbf{W}_V \in \mathbb{R}^{d_h \times d_e}$, $\mathbf{W}_2$, $\mathbf{W}_1 \in \mathbb{R}^{d_h \times d_h}$, $\boldsymbol{b}_1$, $\boldsymbol{b}_2 \in \mathbb{R}^{d_h}$, and $\sigma_R$ is the ReLU activation function. In essence, the history embedding $\boldsymbol{h}_i$ is therefore constructed as a weighted sum of the representations of the past events, where the attention weights are obtained by measuring a similarity score (e.g. dot product) between the projection of $\boldsymbol{e}_{i-1}$, and the projections of $\{\boldsymbol{e}_1, ..., \boldsymbol{e}_{i-1}\}$.

As pointed out by Zuo et al. (2020), recurrent architectures may fail to capture long-term dependencies and are difficult to train due to vanishing and exploding gradients. Moreover, due to their inherently sequential nature, recurrent models forbid parallel processing and are usually trained using truncated backpropagation through time, which only returns an approximation of the true gradients. Conversely, architectures based on a self-attention mechanism can compute the history embeddings $\boldsymbol{h}_i$ for each $i$ in parallel, leading to improved computational efficiency. However, since each $\boldsymbol{h}_i$ depends on all events preceding $e_i$, computing such embeddings for all $L$ events of a sequence scales in $O(L^2)$ time for architectures based on a self-attention mechanism, while it scales in $O(L)$ time for their recurrent counterparts (Shchur et al., 2021).

### 3.3 Decoders

As mentioned in Section 2, fully characterizing a marked TPP can be achieved by parametrizing either of the functions $\lambda_k^*(t)$, $\Lambda_k^*(t)$ or $f^*(k, t)$, as any can be uniquely retrieved from the others. While being mathematically equivalent, a specific choice among these functions leads to particular advantages, challenges, and constraints, which will be discussed below. Nonetheless, given the encoding $\boldsymbol{e}$ of a query event $e = (t, k)$ with $t > t_i$ and its history embedding $\boldsymbol{h}_i$, the parametrization is almost systematically carried out by a neural network. We will consider the following state-of-the-art decoders in our study:

- The Exponential Constant decoder (**EC**) (Upadhyay et al., 2018) parametrizes a constant MCIF between two events using a feed-forward network on $\boldsymbol{h}_i$ as in (22).

- The MLP decoder (**MLP/MC**) (Enguehard et al., 2020) parametrizes the MCIF as in (26) using a feed-forward network on $\boldsymbol{e}$ and $\boldsymbol{h}_i$.

- The Self-Attention decoder (**SA/MC**) (Enguehard et al., 2020) parametrizes the MCIF as in (28) by letting $\boldsymbol{e}$ attend to all $\{\boldsymbol{h}_1, ..., \boldsymbol{h}_i\}$.

- The Neural Hawkes decoder (**NH**) (Mei & Eisner, 2016) parametrizes the MCIF as in (34) using a set of continuous-time LSTM equations.

- The RMTPP decoder (**RMTPP**) (Du et al., 2016) separately parametrizes the GCIF as in (30), and a mark distribution independent of time given the history as in (31).

- The LogNormMix decoder (**LNM**) (Shchur et al., 2019) separately parametrizes the conditional density of inter-arrival times as a mixture of log-normals as in (44), and the mark distribution similarly to RMTPP.

- The LogNorm decoder (**LN**) is defined similarly to LNM, but with a single mixture component.

- The FullyNN decoder (**FNN**) (Omi et al., 2019) parametrizes the cumulative MCIF as in (52) using a feed-forward network on $\boldsymbol{e}$ and $\boldsymbol{h}_i$.

- The Cumulative Self-Attention decoder (**SA/CM**) (Enguehard et al., 2020) parametrizes the cumulative MCIF as in (56) by letting $\boldsymbol{e}$ attend to all $\{\boldsymbol{h}_1, ..., \boldsymbol{h}_i\}$.

- A Hawkes decoder (**Hawkes**) with exponential kernels, which parametrizes the MCIF as in (9).

- A simple Poisson decoder (**Poisson**), which parametrizes a constant MCIF as in (7).

In the following, we will describe and discuss each of the above decoders, except the Poisson and Hawkes decoders which have already been introduced in Section 2.

**EC decoder.** The MCIF is assumed to be constant between two events and is parametrized using a feed-forward network on the history embedding, i.e.

$$\lambda_k^*(t) = \lambda_k^* = \sigma_{S,k}\left(\boldsymbol{w}_k^\mathsf{T}(\sigma_R(\mathbf{W}_1\boldsymbol{h}_i + \boldsymbol{b}_1) + b_k\right), \tag{22}$$

where $\mathbf{W}_1 \in \mathbb{R}^{d_{in} \times d_h}$ and $\boldsymbol{b}_1 \in \mathbb{R}^{d_{in}}$, and $\boldsymbol{w}_k \in \mathbb{R}^{d_{in}}$ and $b_k$ are mark-specific weights and biases, respectively. $\sigma_{S,k}$ is a mark-specific Softplus activation function:

$$\sigma_{S,k}(x) = s_k\log\left(1 + \exp\left(\frac{x}{s_k}\right)\right), \tag{23}$$

with $s_k \in \mathbb{R}_+$ ensuring $R_1^\lambda$. The MCIF being independent of time between two events, its cumulative is given by

$$\Lambda_k^*(t) = (t - t_{i-1})\lambda_k^*(t), \tag{24}$$

and the distribution of inter-arrival times is exponential with rate $\lambda^* = \sum_{k=1}^K \lambda_k^*$:

$$f^*(\tau) = \lambda^* \exp(-\lambda^*\tau). \tag{25}$$

**MLP/MC decoder.** Although the EC decoder can capture more general patterns than the Hawkes decoder, it does not allow evolving dynamics between consecutive events. To circumvent this limitation, the MLP/MC decoder takes as input the concatenation of $\boldsymbol{e}^t$ (which is a function of $t$) and $\boldsymbol{h}_i$. In contrast to the EC decoder, the MCIF can vary between consecutive events:

$$\lambda_k^*(t) = \mu_k + \sigma_{S,k}\left(\boldsymbol{w}_k^\mathsf{T}(\sigma_R(\mathbf{W}_1[\boldsymbol{h}_i, \boldsymbol{e}^t] + \boldsymbol{b}_1) + b_k\right). \tag{26}$$

where $\mu_k \in \mathbb{R}_+$ ensures that $R_2^\lambda$ is met. The Softplus activation forbids the computation of the cumulative MCIF in closed form, requiring numerical integration techniques, such as Monte Carlo, to approximate its expression:

$$\Lambda_k^*(t) \simeq \frac{t - t_{i-1}}{n_s} \sum_{j=1}^{n_s} \lambda_k^*(s_j), \tag{27}$$

where $s_j \sim \mathcal{U}[t_{i-1}, t]$ and $n_s$ is the number of Monte Carlo samples.

**SA/MC decoder**. The MCIF is parametrized by re-employing the same set of equations as (20), with the only difference being that the query vector is constructed using $e^t$. By doing so, this allows the query event $e$ to attend to previous event representations in $\mathcal{H}_t$:

$$\lambda_k^*(t) = \mu_k + \sigma_{S,k}\big(w_k^\mathsf{T}(\mathrm{SA}(q, \mathbf{K}_i, \mathbf{V}_i) + b_k)\big), \tag{28}$$

$$q = \mathbf{W}_Q e^t, \quad \mathbf{K}_i = \mathbf{W}_K[h_1, ..., h_i], \quad \mathbf{W}_V = \mathbf{V}_i[h_1, ..., h_i], \tag{29}$$

where $\mathbf{W}_Q \in \mathbb{R}^{d_q \times d_t}$, $\mathbf{W}_K \in \mathbb{R}^{d_q \times d_h}$ and $\mathbf{W}_V \in \mathbb{R}^{d_h \times d_z}$, $d_z$ being the output dimension of the attention mechanism. Similarly to the MLP/MC decoder, the cumulative MCIF must be approximated by numerical integration techniques, e.g. as in (27).

**RMTPP decoder.** Instead of specifying the MCIF, the RMTPP decoder separately parametrizes the GCIF $\lambda^*(t)$ and the mark distribution $p^*(k)$ as:

$$\lambda^*(t) = \exp\big(w^t(t - t_{i-1}) + (w^h)^\mathsf{T} h_i + b\big), \tag{30}$$

$$p^*(k) = \mathrm{Softmax}(\mathbf{W}^h h_i + b)_k, \tag{31}$$

where $w^t \in \mathbb{R}_+$, $w^h \in \mathbb{R}^{d_h}$, $b \in \mathbb{R}$, $\mathbf{W}^h \in \mathbb{R}^{K \times d_h}$, and $b \in \mathbb{R}^K$. The exponential transformation in (30), along with the positivity of $w^t$, ensure that $R_1^\lambda$ and $R_2^\lambda$ are met. By assuming the distribution of marks to be independent of the time given the history of the process, the RMTPP decoder makes a strong simplifying assumption, which has been criticized in previous works (Enguehard et al., 2020). However, the exponential in (30) allows us to directly compute the cumulative GCIF in closed form:

$$\Lambda^*(t) = \frac{1}{w^t}\Big(\exp\big(w^t(t - t_{i-1}) + (w^h)^\mathsf{T} h_i + b\big) - \exp\big((w^h)^\mathsf{T} h_i + b\big)\Big). \tag{32}$$

Moreover, as pointed out by Shchur et al. (2019), the RMTPP decoder defines a Gompertz distribution (Wienke, 2010) on the inter-arrival times, with a density given by

$$f^*(\tau) = \beta\eta \exp\left(\beta\tau - \eta \exp(\beta\tau) + \eta\right), \tag{33}$$

with shape $\eta = \frac{\exp\big((w^h)^\mathsf{T} h_i + b\big)}{w^t}$ and scale $\beta = w^t$.

**NH decoder.** The MCIF is parametrized using a fully-connected layer on a history embedding $h_i(t)$ that is allowed to vary between consecutive events, i.e:

$$\lambda_k^*(t) = \sigma_{S,k}\Big((w_k)^\mathsf{T} h_i(t)\Big). \tag{34}$$

In contrast to a classical discrete-time LSTM, which would only update the history embedding $h_i$ when the event at time $t_i$ is observed, a continuous-time LSTM allows the history embedding to

exponentially decay in the interval $]t_{i-1}, t_i]$, making it a function of time. Once $(t_i, k_i)$ is observed, the continuous-time LSTM will apply a discrete update to $\boldsymbol{h}_i(t_i)$, using a modified set of discrete-time LSTM equations (37)-(43). Specifically, the parametrization of the MCIF can be summarized as

$$\boldsymbol{h}_i(t) = \boldsymbol{o}_i \odot \Big( 2\sigma_{Si}\big(2\boldsymbol{c}(t)\big) - 1 \Big), \tag{35}$$

$$\boldsymbol{c}(t) = \bar{\boldsymbol{c}}_i + (\boldsymbol{c}_i - \bar{\boldsymbol{c}}_i)\exp\big(-\boldsymbol{\delta}_i(t - t_{i-1})\big), \tag{36}$$

$$\boldsymbol{i}_i = \sigma_{Si}(\mathbf{W}_i \boldsymbol{k}_{i-1} + \mathbf{U}_i \boldsymbol{h}_{i-1}(t_i) + \boldsymbol{b}_i), \tag{37}$$

$$\boldsymbol{f}_i = \sigma_{Si}(\mathbf{W}_f \boldsymbol{k}_{i-1} + \mathbf{U}_f \boldsymbol{h}_{i-1}(t_i) + \boldsymbol{b}_f), \tag{38}$$

$$\boldsymbol{z}_i = 2\sigma_{Si}(\mathbf{W}_z \boldsymbol{k}_{i-1} + \mathbf{U}_z \boldsymbol{h}_{i-1}(t_i) + \boldsymbol{b}_z) - 1, \tag{39}$$

$$\boldsymbol{o}_i = \sigma_{Si}(\mathbf{W}_o \boldsymbol{k}_{i-1} + \mathbf{U}_o \boldsymbol{h}_{i-1}(t_i) + \boldsymbol{b}_o), \tag{40}$$

$$\boldsymbol{c}_i = \boldsymbol{f}_i \odot \boldsymbol{c}(t_{i-1}) + \boldsymbol{i}_i \odot \boldsymbol{z}_i, \tag{41}$$

$$\bar{\boldsymbol{c}}_i = \boldsymbol{f}_i \odot \bar{\boldsymbol{c}}_{i-1} + \boldsymbol{i}_i \odot \boldsymbol{z}_i, \tag{42}$$

$$\boldsymbol{\delta}_i = \sigma_S(\mathbf{W}_d \boldsymbol{k}_{i-1} + \mathbf{U}_d \boldsymbol{h}_{i-1}(t_i) + \boldsymbol{b}_d), \tag{43}$$

where $\boldsymbol{k}_{i-1} \in \{0,1\}^K$ is the one-hot encoding of mark $k_{i-1}$, $\boldsymbol{w}_k \in \mathbb{R}^{d_h}$, $\mathbf{W}_i$, $\mathbf{W}_f$, $\mathbf{W}_z$, $\mathbf{W}_o$, $\mathbf{W}_d \in \mathbb{R}^{d_h \times K}$, $\mathbf{U}_i$, $\mathbf{U}_f$, $\mathbf{U}_z$, $\mathbf{U}_o$, $\mathbf{U}_d \in \mathbb{R}^{d_h \times d_h}$, and $\boldsymbol{b}_i$, $\boldsymbol{b}_f$, $\boldsymbol{b}_z$, $\boldsymbol{b}_o$, $\boldsymbol{b}_d \in \mathbb{R}^{d_h}$. $\sigma_{Si}$ is the Sigmoïd activation function, and $\sigma_S$ is the unmarked formulation of the Softplus activation, i.e. $\sigma_S(x) = s \log(1 + \exp(x/s))$ with $s \in \mathbb{R}_+$. Here also, one must rely on numerical integration techniques to estimate the cumulative MCIF.

**LNM decoder.** The conditional density of inter-arrival times is parametrized as a mixture of log-normal distributions, i.e:

$$f^*(\tau) = \sum_{m=1}^{M} p_m \frac{1}{\tau \sigma_m \sqrt{2\pi}} \exp\Big( -\frac{(\log \tau - \mu_m)^2}{2\sigma_m^2} \Big), \tag{44}$$

where $p_m = \text{Softmax}\big(\mathbf{W}_p \boldsymbol{h}_i + \boldsymbol{b}_p\big)_m$ corresponds to the probability that $\tau_i$ was generated by the $m^{th}$ mixture component, while $\mu_m = (\mathbf{W}_\mu \boldsymbol{h}_i + \boldsymbol{b}_\mu)_m$, $\sigma_m = \exp(\mathbf{W}_\sigma \boldsymbol{h}_i + \boldsymbol{b}_\sigma)_m$ are the mean and standard deviation of the $m^{th}$ mixture component, respectively. $\mathbf{W}_p, \mathbf{W}_\mu, \mathbf{W}_\sigma \in \mathbb{R}^{M \times d_h}$ and $\boldsymbol{b}_p, \boldsymbol{b}_\mu, \boldsymbol{b}_\sigma \in \mathbb{R}^M$, $M$ being the number of mixture components. Defined similarly to equation (31), the mark distribution $p^*(k)$ is assumed to be conditionally independent of the inter-arrival times given the history of the process. Although not available in closed-form, the cumulative distribution of a mixture of log-normals can be approximated with high precision (Abramowitz & Stegun, 1965):

$$F^*(\tau) = \sum_{m=1}^{M} \frac{1}{2}\Big[1 + \text{erf}\Big(\frac{\log \tau - \mu_m}{\sigma_m \sqrt{2}}\Big)\Big], \tag{45}$$

where $\text{erf}(x) = \frac{2}{\sqrt{\pi}} \int_0^x e^{-s^2} ds$ is the Gaussian error function. From $F^*(\tau)$, we can retrieve the cumulative GCIF as:

$$\Lambda^*(t) = -\log\Big(1 - F^*(\tau)\Big). \tag{46}$$

**FNN decoder.** A major bottleneck of models parametrizing the MCIF resides in its cumulative not always being available in closed-form, thus requiring expensive numerical integration techniques. An elegant way of addressing this challenge is to directly parametrize the cumulative MCIF, from

which $\lambda_k^*(t)$ can be easily retrieved through differentiation. The original definition of FullyNN involved parameterizing $\Lambda^*(t)$ using a fully-connected network that operated on both the inter-arrival times and the history embeddings. In the context of a marked setting, this can be expressed as follows:

$$\Lambda_k^*(t) = \sigma_{S,k}\big(\boldsymbol{w}_k^\top(\tanh\big(\boldsymbol{w}^t(t - t_{i-1}) + \mathbf{W}^h \boldsymbol{h}_i + \boldsymbol{b}\big) + b_k\big), \tag{47}$$

where each weight is constrained to be positive, ensuring condition $R_1^\Lambda$ and $R_4^\Lambda$, i.e. $\boldsymbol{w}_k \in \mathbb{R}_+^{d_{in}}$, $\boldsymbol{w}^t \in \mathbb{R}_+^{d_{in}}$, $\mathbf{W}^h \in \mathbb{R}_+^{d_{in} \times d_h}$, $\boldsymbol{b} \in \mathbb{R}_+^{d_{in}}$ and $b_k \in \mathbb{R}_+$. However, as pointed out by Shchur et al. (2019), equation (47) fails to satisfy $R_2^\Lambda$:

$$\Lambda_k^*(t_{i-1}) = \sigma_S\big(\boldsymbol{w}_k^\top(\tanh\big(\mathbf{W}^h \boldsymbol{h}_i + \boldsymbol{b}_1\big) + b_k\big) > 0, \tag{48}$$

which in turn yields $F^*(\tau = 0) > 0$. In other terms, the original FNN model attributes non-zero probability mass to null inter-arrival times. The original formulation also fails to satisfy $R_3^\Lambda$ due to the saturation of the tanh activation function:

$$\lim_{t \to \infty} \Lambda_k^*(t) = \sigma_S\left(\sum_{d=1}^{d_{in}} w_{k,d} + b_k\right) < \infty. \tag{49}$$

To prevent both these shortcomings, Enguehard et al. (2020) proposed to replace the tanh activation function with a Gumbel-Softplus activation:

$$\sigma_{GS,k}(x) = \big[1 - \big(1 + \alpha_k \exp(x)\big)^{-\frac{1}{\alpha_k}}\big]\big[1 + \sigma_{S,k}(x)\big], \tag{50}$$

with $\alpha_k \in \mathbb{R}_+$. The Gumbel-Softplus activation function being non-saturating (i.e. $\lim_{x \to \infty} \sigma_{GS,k}(x) = \infty$), using it as a replacement for the tanh activation function in (52) indeed satisfies $R_3^\Lambda$. Finally, $R_2^\Lambda$ can be satisfied by defining $\Lambda_k^*(t)$ as

$$\Lambda_k^*(t) = G_k^*(t) - G_k^*(t_{i-1}), \tag{51}$$

$$G_k^*(t) = \sigma_{S,k}\big(\boldsymbol{w}_k^\top(\sigma_{GS,k}\big(\mathbf{W}_1^t \boldsymbol{e}^t + \mathbf{W}_1^h \boldsymbol{h}_i + \boldsymbol{b}_1\big) + b_k\big), \tag{52}$$

which defines the generalized corrected version of FullyNN. Note that $G_k^*(t)$ takes now as input any encoding of a query time $\boldsymbol{e}^t$ with $\mathbf{W}_1^t \in \mathbb{R}^{d_{in} \times d_t}$. However, to ensure that $R_4^\Lambda$ remains satisfied, $\boldsymbol{e}^t$ must be monotonic in its input. Therefore, when parametrizing a cumulative decoder, the temporal event encoding in (15) cannot be used, and the weights of encoding (16) must be constrained to be positive. From the cumulative MCIF, $\lambda_k^*(t)$ can finally be retrieved through differentiation:

$$\lambda_k^*(t) = \frac{d}{dt}\Lambda_k^*(t). \tag{53}$$

**SA/CM decoder.** The SA/CM decoder shares a similar set of equations as in (28) to parametrize the cumulative MCIF. However, alike FullyNN, several modifications of the latter are required to ensure that the decoder meets the various constraints imposed by $\Lambda_k^*(t)$. First, the Softmax activation is replaced by a Sigmoid to satisfy $R_4^\Lambda$. Then, the ReLU activation is replaced by a Gumbel-Softplus, which prevents

$$\frac{d^2\Lambda_k^*(t)}{dt^2} = \frac{d\lambda_k^*(t)}{dt} = 0. \tag{54}$$

In other terms, a cumulative model defined with a ReLU activation is equivalent to the EC decoder in (22) (Enguehard et al., 2020). Moreover, given the saturation of the Sigmoid function, a term $\mu_k(t - t_i)$ with $\mu_k \in \mathbb{R}_+$ is added to the cumulative MCIF to satisfy $R_4^\Lambda$. Note that including this term is equivalent to adding $\mu_k$ directly to the MCIF. Finally, the Cumulative Self-Attention decoder is given by

$$\Lambda_k^*(t) = G_k^*(t) - G_k^*(t_{i-1}), \tag{55}$$

$$G_k^*(t) = \mu_k(t - t_{i-1}) + \sigma_{S,k}\big(\boldsymbol{w}_k^\mathsf{T}(\mathrm{SA}\big(\boldsymbol{q}, \mathbf{K}_i, \mathbf{V}_i\big) + b_k\big), \tag{56}$$

$$\mathrm{SA}(\boldsymbol{q}, \mathbf{K}_i, \mathbf{V}_i) = \mathbf{W}_2 \sigma_{GS,k}\big(\mathbf{W}_1 \hat{\boldsymbol{z}}_i^\mathsf{T} + \boldsymbol{b}_1\big) + \boldsymbol{b}_2, \tag{57}$$

$$\hat{\boldsymbol{z}}_i = \mathrm{Sigmoid}\Big(\frac{\boldsymbol{q}^\mathsf{T}\mathbf{K}_i}{\sqrt{d_q}}\Big)\mathbf{V}_i^\mathsf{T},$$

where $\mathbf{W}_1$, $\mathbf{W}_2 \in \mathbb{R}_+^{d_z \times d_z}$, $\boldsymbol{b}_1, \boldsymbol{b}_2 \in \mathbb{R}_+^{d_z}$, and where each entry of $\mathbf{W}_K$, $\mathbf{W}_V$, $\mathbf{W}_Q$, $\boldsymbol{w}_k$ and $b_k$ is now constrained to be positive. The query $\boldsymbol{q}$, keys $\mathbf{K}_i$, and values $\mathbf{V}_i$ are given in (29). As before, the MCIF can be retrieved through differentiation.

## 4 Related Work

**Neural Temporal Point Processes.** Simple parametric forms of TPP models, such as the self-exciting Hawkes process (Hawkes, 1971), or the self-correcting process (Isham & Westcott, 1979), rely on strong modeling assumptions, which inherently limits their flexibility. To capture complex dynamics of real-world processes, the ML community eventually turned to the latest advances in neural modeling and proposed new TPP models based on various neural-network architectures. For instance, Du et al. (2016) proposed to model the inter-arrival time distribution as a Gompertz distribution with a history encoding based on a discrete-time RNN. On a similar line of work, Mei & Eisner (2016) extended this idea by parametrizing the MCIF using a modified continuous-time RNN that allows the history to evolve between consecutive events. Given their huge success as sequence encoders in NLP, Zuo et al. (2020) and Zhang et al. (2020) employed a self-attention mechanism to encode the process history. Since differentiation is often easier to carry out than integration, FullyNN (Omi et al., 2019) instead proposed to directly parametrize the cumulative GCIF using a feed-forward network, avoiding expensive numerical integration techniques. Inspired by FullyNN, Enguehard et al. (2020) corrected and extended the FullyNN model to the marked case, and proposed a generic self-attention decoder that can be employed to parametrize the cumulative MCIF, but also the MCIF itself. Instead of modeling the (cumulative) MCIF, Xiao et al. (2017b) directly modeled the conditional density of inter-arrival times with a Gaussian distribution, while Shchur et al. (2019) relies on a mixture of log-normals whose parameters are obtained from an RNN-encoded history. To further alleviate the strong assumption of independence of times and marks given the history, Waghmare et al. (2022) proposed to model conditional mixtures of log-normals for each mark separately. Finally, Ben Taieb (2022) proposed to parametrize the conditional quantile function using recurrent neural splines, enabling analytical sampling of inter-arrival times. Alternatives to NLL optimization techniques for training neural TPPs include $\beta-$VAE objectives (Boyd et al., 2020), CRPS (Ben Taieb, 2022), reinforcement learning (De et al., 2019), noise contrastive estimation (Guo et al., 2018; Mei et al., 2020) and adversarial learning (Xiao et al., 2017a; 2018). For surveys on recent advances in neural TPP modeling, refer to Shchur et al. (2021) and Yan (2019).

**Experimental studies**. Most papers in the neural TPP literature mainly focused on proposing methodological improvements for modeling streams of event data, often inspired by contemporary advances in neural network representation learning. While these contributions are paramount to driving future progress in the field of TPP, few studies have been carried out to identify real sources of empirical gains across neural architectures and highlight future interesting research directions. The empirical study of Lin et al. (2021) is the closest to our work. While they also compared the impact of various history encoders and decoders, they did not discuss the influence of different event encoding mechanisms. However, we found that specific choices for this architectural component can lead to drastic performance gains. Additionally, they did not include simple parametric models to their baselines, such as the Hawkes decoder. Considering these models in an experimental study allows however to fairly evaluate the true gains brought by neural architectures. We also assess the calibration of neural and non-neural TPP models on the distribution of arrival times and marks, and our experiments are conducted across a wider range of real-world datasets. Finally, our results are supported by rigorous statistical tests. Lin et al. (2022) also conducted empirical comparisons in the context of neural TPPs but their attention was focused on deep generative models which are orthogonal to our work.

## 5 Experimental study of neural TPP models

We carry out a large-scale experimental study to assess the predictive accuracy of state-of-the-art neural TPP models on 15 real-world event sequence datasets in a carefully designed unified setup. We also consider classical parametric TPP models, such as Hawkes and Poisson processes, as well as synthetic Hawkes datasets. In particular, we study the influence of each major architectural component (event encoding, history encoder, decoder parametrization) for both time and mark prediction tasks. The next section summarizes the variations of the architectural components we have considered in our experimental study. Section 5.2 presents the datasets including summary statistics and pre-processing steps. Section 5.3 describes our experimental setup. Finally, Section 5.4 presents the evaluation metrics and statistical tools used to assess the accuracy of the considered models, and Section 5.5 describes the procedure employed to aggregate the results across multiple datasets.

### 5.1 Models

To ease the understanding of the following sections, a brief summary of the different architectures considered in our experiments is presented below. Table 1 summarizes the correspondence between all variations of event encoding, history encoder, and decoder, and their mathematical expressions.

**Event encoding mechanisms.** For the event encoding mechanism, we consider the raw inter-arrival times $\tau$ (**TO**), the logarithms of inter-arrival times (**LTO**), a temporal encoding of arrival times (**TEM**), and a learnable encoding of inter-arrival times (**LE**). Additionally, we include all their variants resulting from the concatenation of the mark embeddings $e^k$, i.e. **CONCAT** (TO and $e^k$), **LCONCAT** (LTO an $e^k$), **TEMWL** (TEM and $e^k$), and **LEWL** (LE and $e^k$).

**History encoders**. To encode a process' history, we employ a **GRU**, and a self-attention mechanism (**SA**). Additionally, we consider a constant history encoder (**CONS**), which systematically outputs $h_i = 1_{d_h}$. Note that a decoder equipped with this encoder parametrizes a function in-

Table 1: Summary of the various architectures components considered in the comparative study.

| Component | Name | Acronym | Parametrization |
|---|---|---|---|
| | Times | TO | $\boldsymbol{e}_i = \tau_i$ |
| | Log-times | LTO | $\boldsymbol{e}_i = \log \tau_i$ |
| | Concatenate | CONCAT | $\boldsymbol{e}_i^t = \tau_i,\ \boldsymbol{e}_i^k$ as in (17), $\boldsymbol{e}_i$ as in (14) |
| Event encoding | Log-concatenate | LCONCAT | $\boldsymbol{e}_i^t = \log \tau_i,\ \boldsymbol{e}_i^k$ as in (17), $\boldsymbol{e}_i$ as in (14 |
| | Temporal | TEM | $\boldsymbol{e}_i$ as in (15) |
| | Temporal with labels | TEMWL | $\boldsymbol{e}_i^t$ as in (15), $\boldsymbol{e}_i^k$ as in (17), $\boldsymbol{e}_i$ as in (14) |
| | Learnable | LE | $\boldsymbol{e}_i$ as in (16) |
| | Learnable with labels | LEWL | $\boldsymbol{e}_i^t$ as in (16), $\boldsymbol{e}_i^k$ as in (17), $\boldsymbol{e}_i$ as in (14) |
| | GRU | GRU | $\boldsymbol{h}_i = \mathrm{GRU}(\boldsymbol{e}_1, ..., \boldsymbol{e}_{i-1})$ as in (18) |
| Encoder | Self-attention | SA | $\boldsymbol{h}_i$ as in (20) |
| | Constant | CONS | $\boldsymbol{h}_i = \mathbf{1}_{d_h}$ |

| Component | Name | Acronym | Parametrization | Closed-form MLE |
|---|---|---|---|---|
| | Exponential constant | EC | $\lambda_k^*$ as in (22) | ✔ |
| | MLP | MLP/MC | $\lambda_k^*(t)$ as in (26) | ✗ |
| | FullyNN | FNN | $\Lambda_k^*(t)$ as in (52) | ✔ |
| | LogNormMix | LNM | $f^*(\tau)$ as in (44), $p^*(k)$ in (31) | ✔ |
| | LogNorm | LN | $f^*(\tau)$ as in (44) with $M=1$, $p^*(k)$ in (31) | ✔ |
| Decoder | RMTPP | RMTPP | $\lambda^*(t)$ as in (30), $p^*(k)$ in (31) | ✔ |
| | Neural Hawkes | NH | $\lambda_k^*(t)$ as in (34) | ✗ |
| | Self-attention | SA/MC | $\lambda_k^*(t)$ as in (28) | ✗ |
| | Cumulative Self-attention | SA/CM | $\Lambda_k^*(t)$ as in (56) | ✔ |
| | Hawkes | Hawkes | $\lambda_k^*(t)$ as in (9) | ✔ |
| | Poisson | Poisson | $\lambda_k^*(t)$ as in (7) | ✔ |

dependent of the process' history and reduces essentially to a renewal process (Lindqvist et al., 2003).

**Decoders**. The decoders considered in this experimental study can be classified on the basis of the function that they parametrize, as well as on the assumptions they make regarding the distribution of inter-arrival times and marks. As described in Section 3.3, decoders that parametrize the MCIF are the **EC** , **MLP/MC** , **SA/MC**, **NH**, **Hawkes** and **Poisson** decoders. On the other hand, decoders that directly parametrize the cumulative MCIF are the **FNN** and **SA/CM** decoders. Finally, **RMTPP** separately parametrizes the GCIF and the distribution of marks, while **LNM** and its single mixture version, **LN**, separately parametrize the density of inter-arrival times and the distribution of marks. For these last three decoders, the distribution of marks is assumed to be independent of the time, conditional on the history of the process.

Although not required to define a valid parametrization of a marked TPP, we also consider for completeness the setting where a constant baseline intensity term $\mu_k$ (**B**) is added to the MCIF of the EC and RMTPP decoders. We define a model as a specific choice of event encoding mechanism, history encoder, and decoder, e.g. GRU-MLP/MC-TO corresponds to a model using the GRU encoder to build $\boldsymbol{h}_i$, the MLP/MC decoder to parametrizes $\lambda_k^*(t)$, and where the events are encoding using the TO event encoding. While in most cases, any variation of a component can be seamlessly associated with any other variation of other components, some combinations are either impossible or meaningless. Indeed, all cumulative decoders (SA/CM and FullyNN) cannot be trained with

the TEM or TEMWL event encodings, as both would violate the monotonicity constraint of the cumulative MCIF. Moreover, as all event encodings are irrelevant with respect to the CONS history encoder, all CONS-EC models are equivalent. For the NH decoder, we stick to the original model definition, as it can be hardly disentangled into different components. A complete list of the considered combinations is given in Table 14 in Appendix C.

We further classify the different models into three categories: *parametric*, *semi-parametric*, and *non-parametric*. Parametric TPP models include classical (i.e. non-neural) architectures, characterized by strong modeling assumptions and hence, low flexibility. We consider the Hawkes and Poisson decoders as parametric baselines. On the other hand, semi-parametric models include architectures that still make assumptions regarding the distribution of inter-arrival times, but their parameters are obtained from the output of a neural network. All models equipped with LNM, LN, RMTPP, or EC decoders are deemed semi-parametric. All remaining baselines, i.e. FNN, MLP/MC, SA/MC, SA/CM, and NH, are considered non-parametric models.

## 5.2 Datasets

**Real-world datasets.** A total of 15 real-world datasets containing sequences of various lengths are used in our experiments, among which 7 possess marked events. A brief description is presented below, while their general statistics are summarized in Tables 2 and 3.

- **Marked Datasets**

  - **LastFM** [3] (Kumar et al., 2019) : Records of users listening to songs. Each sequence corresponds to a user, and the artist of the song is the mark.
  - **MOOC** [3] (Kumar et al., 2019): Records of student's actions on an online course system. Each sequence corresponds to a user, and the type of action is the mark.
  - **Wikipedia** [3] (Kumar et al., 2019) : Records of Wikipedia pages' edits. Each sequence corresponds to a Wikipedia page, and marks relate to the user that edited the corresponding page.
  - **MIMIC2** [4] (Du et al., 2016) : Clinical records of patients of an intensive care unit for seven years. Each sequence corresponds to a patient, and marks describe the type of disease.
  - **Github** [5] (Trivedi et al., 2018) : Records of users' actions on the open-source platform Github during the year 2013. Each sequence corresponds to a user, and the marks describe the action performed (Watch, Star, Fork, Push, Issue, Comment Issue, Pull Request, Commit).
  - **Stack Overflow** [7] (Du et al., 2016) : Records of the time users received a specific badge on the question-answering website Stack Overflow. Each sequence corresponds to a user, and the mark is the badge received.
  - **Retweets** [4] (Mei & Eisner, 2016) : Streams of retweet events following the creation of an original tweet. Each sequence corresponds to a tweet, and the category (i.e. "small", "medium", "large") to which a retweeter belongs corresponds to the mark.

---

[3] https://github.com/srijankr/jodie/
[4] https://github.com/babylonhealth/neuralTPPs
[5] https://github.com/uoguelph-mlrg/LDG

Table 2: Marked datasets statistics after pre-processing. "Sequences (%)" and "Events (%)" respectively correspond to the number of sequences and events that remain in the dataset after the pre-processing step. "MSL" is the mean sequence length.

| | Sequences | Sequences (%) | Events | Events (%) | MSL | Max Len. | Min Len. | Marks |
|---|---|---|---|---|---|---|---|---|
| Wikipedia | 590 | 0.59 | 30472 | 0.19 | 51.6 | 1163 | 2 | 50 |
| MOOC | 7047 | 1.00 | 351160 | 0.89 | 49.8 | 416 | 2 | 50 |
| LastFM | 856 | 0.92 | 193441 | 0.15 | 226.0 | 6396 | 2 | 50 |
| MIMIC2 | 599 | 0.84 | 1812 | 0.68 | 3.0 | 32 | 2 | 43 |
| Github | 173 | 1.00 | 20657 | 1.00 | 119.4 | 4698 | 3 | 8 |
| Stack Overflow | 7959 | 1.00 | 569688 | 0.99 | 71.6 | 735 | 40 | 22 |
| Retweets | 24000 | 1.00 | 2610102 | 1.00 | 108.8 | 264 | 50 | 3 |

Table 3: Unmarked datasets statistics after pre-processing. "MSL" is the mean sequence length.

| | Sequences | Events | MSL | Max Len. | Min Len. |
|---|---|---|---|---|---|
| Reddit Subs. | 1094 | 1235128 | 1129.0 | 2658 | 362 |
| Reddit Ask Comments | 1355 | 400933 | 295.9 | 2137 | 4 |
| Taxi | 182 | 17904 | 98.4 | 140 | 12 |
| Twitter | 1804 | 29862 | 16.6 | 169 | 2 |
| Yelp Toronto | 300 | 215146 | 717.2 | 2868 | 424 |
| Yelp Airport | 319 | 9716 | 30.5 | 55 | 9 |
| Yelp Mississauga | 319 | 17621 | 55.2 | 107 | 3 |
| PUBG | 3001 | 229703 | 76.5 | 97 | 26 |

- **Unmarked Datasets**
    - **Twitter** [6] (Shchur et al., 2020) : Records of tweets made by a user over several years.
    - **PUBG** [6] (Shchur et al., 2020) : Records of players' death in the online game PUBG. Each sequence corresponds to a user.
    - **Yelp Airport** [6], **Mississauga** [6] (Shchur et al., 2020), and **Toronto** [7] (Shchur et al., 2019) : Records of users' reviews on the platform Yelp for the McCarran airport, Toronto city and Mississauga city respectively. Each sequence corresponds to a user.
    - **Reddit Ask Comments** [6](Shchur et al., 2020) : Records of comments in reply to Reddit threads within 24hrs of the original post submission. Each sequence corresponds to a thread.
    - **Reddit Subs** [6](Shchur et al., 2020) : Records of submissions to a political sub-Reddit in the period from 01.01.2017 to 31.12.2019. Each sequence corresponds to a 24hrs window.
    - **Taxi** [6] (Shchur et al., 2020) : Records of taxi pick-ups in the South of Manhattan.

---

[6]https://github.com/shchur/triangular-tpp
[7]https://github.com/shchur/ifl-tpp

**Pre-processing.** Some marked datasets, such as Wikipedia and LastFM, originally presented a very large amount of marks, which distributions turn out to be highly spread across their respective domains. This observation raises two issues: (1) Some marks are therefore highly under-represented, rendering the task of learning their underlying distribution difficult, and (2), as each mark is associated with an MCIF, the computational requirements drastically increase with the number of marks, which is even more exacerbated when Monte Carlo samples need to be drawn. With the incentive to avoid either of these two bottlenecks, each marked dataset is filtered to only contain events belonging to the 50 most represented marks. The resulting sequences containing less than two events are further removed from the dataset, which makes the number of distinct marks in MIMIC2 drop from 75 to 43. Finally, to avoid numerical instabilities, the arrival times of events are scaled in the interval [0,10]. Specifically, we compute $t_{i,\text{scaled}} = 10t_i/t_{\max}$, where $t_{\max}$ is the largest observed timestamp in the dataset. Unmarked datasets do not go through any processing steps, at the exception of the scaling and removal of sequences containing less than two events.

As observed in Tables 2 and 3, the considered datasets are relatively diverse in their characteristics. Indeed, some datasets, such as Yelp Toronto or LastFM, possess a relatively short number of very long sequences, while others, such as Twitter or MOOC, show the exact opposite characteristics. Figure 2 shows the distributions of the inter-arrival times logarithms across all sequences for all real-world datasets, as well as the mark distribution across all sequences for marked datasets. As observed, the distribution of the pooled (log) inter-arrival times differ significantly from one dataset to the other, in some cases presenting characteristics such as multimodality (LastFM, MOOC, Wikipedia, Yelp Toronto, PUBG) or large variance (MOOC, Wikipedia, Github, Yelp Toronto). The distribution of pooled marks also shows different characteristics. While the marks look evenly spread across their domains on LastFM, MOOC, and Wikipedia, their distribution appears sharper on Github, MIMIC2, Stack Overflow, and Retweets. Such diversity should be empirically beneficial, as it would allow us to assess the models' performance across a wider range of real-life applications. In Figures 12, 13 and 14 of Appendix D, we show the distribution of inter-arrival times and marks for some randomly sampled sequences in each dataset.

**Synthetic datasets.** We generate a synthetic dataset from the multidimensional Hawkes process with the exponential kernel as in (9) with the following parameter values:

$$\boldsymbol{\mu} = \begin{pmatrix} 0.2 \\ 0.6 \\ 0.1 \\ 0.7 \\ 0.9 \end{pmatrix} \qquad \boldsymbol{\alpha} = \begin{pmatrix} 0.25 & 0.13 & 0.13 & 0.13 & 0.13 \\ 0.13 & 0.35 & 0.13 & 0.13 & 0.13 \\ 0.13 & 0.13 & 0.2 & 0.13 & 0.13 \\ 0.13 & 0.13 & 0.13 & 0.3 & 0.13 \\ 0.13 & 0.13 & 0.13 & 0.13 & 0.25 \end{pmatrix} \qquad \boldsymbol{\beta} = \begin{pmatrix} 4.1 & 0.5 & 0.5 & 0.5 & 0.5 \\ 0.5 & 2.5 & 0.5 & 0.5 & 0.5 \\ 0.5 & 0.5 & 6.2 & 0.5 & 0.5 \\ 0.5 & 0.5 & 0.5 & 4.9 & 0.5 \\ 0.5 & 0.5 & 0.5 & 0.5 & 4.1 \end{pmatrix},$$

where the matrix $\alpha$ was scaled to have a spectral radius of approximately 0.8, guaranteeing stationarity of the process (Bacry et al., 2020). The process essentially corresponds to a marked process with $K = 5$ marks, from which we simulate 5 distinct datasets of 1000 sequences using the library *tick* (Bacry et al., 2018).

## 5.3 Experimental Setup

For each real-world and synthetic dataset, we randomly split the sequences into train/validation/test splits of sizes 60%/20%/20%, respectively. The models are trained to minimize the NLL in (11) on the training sequences using mini-batch gradient descent (Suvrit et al., 2011). The NLL on

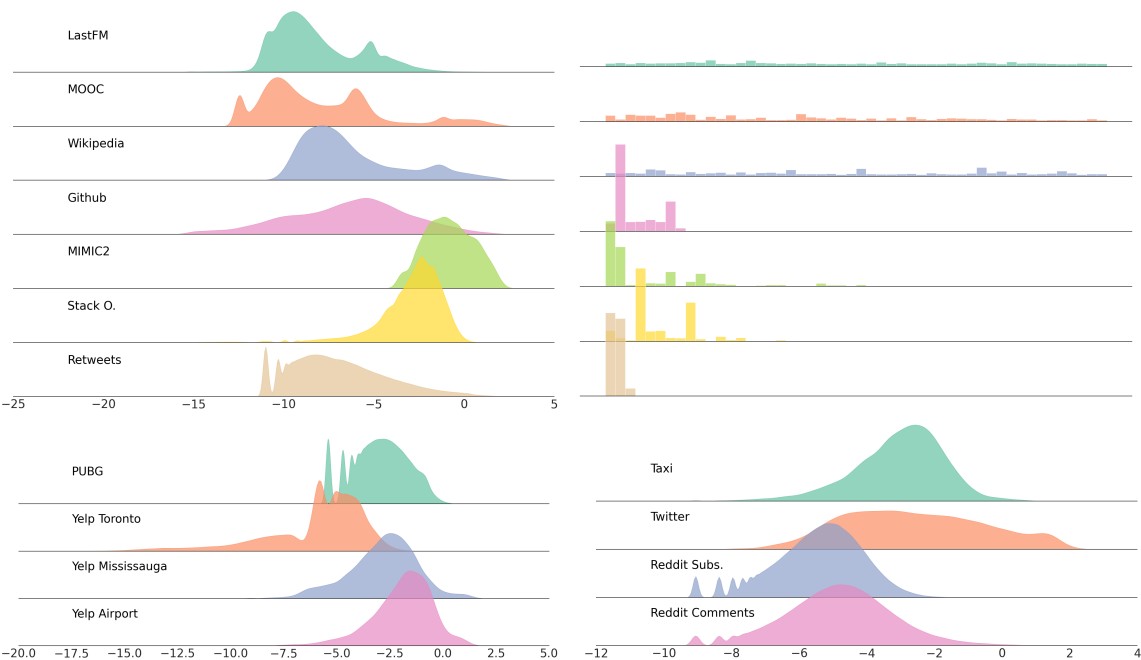

Figure 2: Distribution of log $\tau$ (top left) and mark distribution (top right) for marked datasets, after pre-processing. For unmarked datasets, only the distribution of log $\tau$ (bottom) is reported.

the validation sequences is evaluated at each epoch, and the training procedure is interrupted if the number of epochs reaches 500, or if no improvement is observed for 20 consecutive epochs. In the latter case, the model's parameters are reverted to their state of lowest validation loss. For all models, optimization is carried out using the Adam optimizer (Kingma & Ba, 2014) with an initial learning rate of $10^{-3}$. If no improvement in validation loss is observed for 5 consecutive epochs, the learning rate is divided by a factor of 2, and training continues. We repeat this protocol 5 times using different random train/validation/test splits.

We conduct experiments with different values of event encoding dimension, specifically $\{4, 8, 16, 32\}$, as well as varying the number of hidden units for fully-connected layers in $\{8, 16, 32\}$. For models utilizing GRU, SA encoder, and SA/MC decoder, we explore different numbers of hidden units in $\{8, 16, 32, 64\}$, and consider one or two layers. In the case of SA encoder and SA/MC decoder, we consider one or two heads. Additionally, the number of mixtures in LNM is explored in $\{8, 16, 32, 64\}$. To determine the model's hyperparameters, we follow a specific procedure. For each model and dataset split, we randomly select five hyperparameter configurations. The model is trained using each of these configurations, and we select the configuration with the lowest validation loss. These five best configurations (which may differ depending on the split) are then evaluated on the respective test set. Finally, we report the average test metrics as described in the following section.

### 5.4 Evaluation metrics

We consider a range of metrics to assess the performance of the model in terms of the time prediction task, which involves estimating $f^*(\tau)$, and the mark prediction task, which involves estimating $p^*(k|t)$ and predicting the mark of the next event. As discussed in Section 2, reporting a single NLL metric gathers the contributions of both inter-arrival times and marks, and in effect, obscures how a model actually performs on fitting the two distributions separately. Consequently, we split the NLL into **NLL-T** and **NLL-M** terms, highlighting the contribution of each term to the total NLL metric. Given a set of test sequences $S = \{\mathcal{S}_1, ..., \mathcal{S}_L\}$, where each sequence $\mathcal{S}_l$ contains $n_l$ events, the average test NLL is given by

$$\mathcal{L}(\boldsymbol{\theta}; S) = \underbrace{-\frac{1}{L}\sum_{l=1}^{L}\sum_{i=1}^{n_l}\log f^*(t_{l,i}; \boldsymbol{\theta}) + \Lambda^*(T; \boldsymbol{\theta})}_{\text{NLL-T}} \underbrace{-\frac{1}{L}\sum_{l=1}^{L}\sum_{i=1}^{n_l}\log p^*(k_{l,i}|t_{l,i}; \boldsymbol{\theta})}_{\text{NLL-M}}. \tag{58}$$

On the other hand, a model's ability to predict the next event's mark is measured with the **F1-score**.

**Calibration.** TPP models are probabilistic models which estimate predictive distributions over arrival times and marks. While achieving a low out-of-sample NLL score is crucial, it is also important to ensure that the predictive distributions are well-calibrated. Calibration, in the context of forecasting theory, refers to the statistical consistency between the predicted distribution and the observed outcomes (Gneiting et al., 2007). Formally, a model that outputs a predictive CDF $F^*(\tau)$ (which may be retrieved from $f^*(\tau)$, $\lambda^*(t)$ or $\Lambda^*(t)$) is (unconditionally) probabilistically calibrated if (Dawid, 1984; Kuleshov et al., 2018):

$$\mathbb{P}\big(F^*(\tau) \leq p\big) = p, \quad \forall p \in [0,1], \tag{59}$$

where the probability is taken over $\tau$ and $\mathcal{H}_t$. For example, if a predictive distribution is well-calibrated, it means that a 90% prediction interval for inter-arrival times would, on average, contains the observed inter-arrival times 90% of the time. Similarly, in the context of mark prediction, probabilistic calibration is defined as (Guo et al., 2017):

$$\mathbb{P}\big(\tilde{k} = \bar{k} \mid p^*(\tilde{k}|t) = p\big) = p, \quad \forall p \in [0,1], \tag{60}$$

where $\tilde{k}$ and $\bar{k}$ are the predicted and true mark at time $t$, respectively. Intuitively, when we say a model is calibrated with respect to the mark distribution, it means that 90% of the predictions made with a confidence level of 0.9 should match the observed mark, on average.

We measure probabilistic calibration for the arrival time distribution using the **Probabilistic Calibration Error** (PCE), defined as (Dheur & Ben Taieb, 2023):

$$\text{PCE} = \frac{1}{M}\sum_{m=1}^{M}\left|\sum_{i=1}^{n}\frac{\mathbb{1}[F^*(\tau_i) \leq p_m]}{n} - p_m\right|, \tag{61}$$

where $p_m = \frac{m}{M} \in [0,1]$ are specific probability levels, $n = \sum_{l=1}^{L} n_l$, and where we set $M = 50$. For the mark distribution, we report the **Expected Calibration Error** (ECE) (Naeini et al., 2015)

defined as:

$$\text{ECE} = \frac{1}{M} \sum_{j=1}^{M} \left| \text{acc}(B_j) - \text{conf}(B_j) \right|, \tag{62}$$

where each prediction $\tilde{k}$ is assigned to the $j^{th}$ out of $J$ bins (obtained by discretizing the interval $[0, 1]$ in $J$ equal-size bins) if $p^*(\tilde{k}|t)$ lies in the interval $[\frac{j-1}{J}, \frac{j}{J}]$. $B_j$ is the set of predictions that fell within bin $j$, $\text{acc}(B_j)$ corresponds to the model's accuracy within bin $j$, and $\text{conf}(B_j)$ is the average confidence level of all predictions that fell within bin $j$. We set $J = 10$, and lower PCE and ECE are better.

**Reliability diagrams.** A disadvantage with PCE and ECE metrics is that information regarding the calibration error at individual probability levels $p_1, ..., p_M$, or within individual bins $B_1, ..., B_J$, is lost. Reliability diagrams are visual tools that can be used to assess the probabilistic calibration of a model at a fine-grained level for both continuous and discrete distributions. For the distribution of inter-arrival times, a reliability diagram is obtained by plotting the empirical CDF $\sum_{i=1}^{n} \frac{\mathbb{1}[F^*(\tau_i) \leq p_m]}{n}$ in (61) against all probability levels $p_m$. For the distribution of marks, it is obtained by plotting $\text{acc}(B_j)$ against $\text{conf}(B_j)$ for all $B_j$. In both cases, a probabilistically calibrated model should align with the diagonal line, and any significant deviation from it corresponds to miscalibration (Gneiting et al., 2007; Guo et al., 2017).

**Statistical comparisons.** We further conduct statistical pairwise comparisons between all decoders, for each metric separately. First, Friedman test (Friedman, 1937; 1940) is employed to assess of at least one statistical difference among all decoders. If the null hypothesis is rejected at the $\alpha = 0.05$ significance level, we proceed with comparing each decoder against each other, using Holm's posthoc test (Holm, 1979) to account for multiple hypothesis testing. The outcome of the pairwise comparisons is displayed on critical difference (CD) diagrams, which show the average rank of a model on a metric of interest across all datasets, as well as groups of models that are not statistically different from one another at a given significance level. Refer to Demšar (2006); García & Herrera (2008) for additional details.

### 5.5 Results Aggregation

Given the high number of variations per model's component, the number of possible combinations renders the comparison of all individual models across every dataset unmanageable. To overcome this challenge, we aggregate the model results for each metric separately across all datasets as follows. When comparing different event encodings, we first group all models that are equipped with a specific decoder variation (e.g. MLP/MC). Then, among this decoder group, we further group all models by event encoding variations (e.g. TO-Any history encoder-MLP/MC). We then compute the average score of that event encoding-decoder group on a given dataset with respect to each metric. Finally, for each metric, we rank this encoding-decoder group against other encoding-decoder groups based on their average score on the same dataset. We apply this operation for each dataset separately and report the average and median scores, as well as the average rank of that component variation group across all datasets.

Moreover, as the scale of the NLLs (NLL-T and NLL-M) vary significantly from one dataset to another, we standardize their values on each dataset separately prior to applying the aggregation

procedure above. For each model, we compute its standardized NLL (-T or -M) as

$$\frac{\text{NLL}_d^m - m_d}{\text{IQR}}, \tag{63}$$

where $\text{NLL}_d^m$ is the NLL score of model $m$ on dataset $d$, while $m_d$ and IQR are the median and inter-quartile ranges of NLL scores for all models on dataset $d$, respectively.

## 6  Results and Discussion

Tables 4 and 5 present the average results across all marked datasets for different variations of decoders and event encoding/history encoders, respectively. Table 6 displays the results for the combination of architectural components that achieved the lowest NLL-T and NLL-M on average across all marked datasets and decoders. Appendix A provides the same results for unmarked datasets, while Appendix B includes additional raw metrics and standard errors for each dataset. In the tables, the "Mean" and "Median" columns represent the average and median aggregated scores, respectively. The "Rank" column indicates the average rank across all marked datasets, as explained in the previous section. For the subsequent discussion, we will focus on the "Mean" column.

We would like to note that while these tables include all marked datasets, we found that certain datasets (MIMIC2, Stack Overflow, Taxi, Reddit Subs, Reddit Comments, Yelp Toronto, and Yelp Mississauga) may not be suitable for benchmarking neural TPP models, as most decoders achieve competitive performance on them. We urge researchers to exercise caution when using these datasets in future studies, and we will discuss this concern later in this section. For completeness, we have included the results of the aggregation procedure with these datasets excluded in Appendix F, and we found no significant differences in the results.

**Analysis of the event encoding.** Comparing the results in Table 4, we aim to provide answers to the two following questions: (1) *Are vectorial representations of time better for estimating the inter-arrival time distribution?* (2) *Do we need to encode past mark occurrences to better model the distribution of future marks?*

(1) We observe that vectorial representations of time (i.e. TEM and LE in opposition to TO and LTO) improve the NLL-T and PCE scores for the EC and both SA decoders. Given that a model equipped with the EC decoder only uses the event encoding mechanism at the history encoding stage, this finding suggests that GRU and SA encoders rely on expressive transformations of time to capture patterns of event occurrences. However, we observed that the GRU encoder is rather stable with respect to the time encoding employed, while the performance of the SA encoder drastically decreases with TO or LTO encodings. Added to the fact that both SA decoders only perform well when using TEM and LE, we conclude that self-attention mechanisms must be combined with vectorial representations of time to yield good performance in the context of neural TPP models.

Furthermore, a log transformation of the inter-arrival times (i.e. LTO) improves performance on the same metrics for the RMTPP, FNN and MLP/MC decoders. Using the LTO encoding in combination with the RMTPP decoder effectively defines an inverse Weibull distribution for the inter-arrival times (Kleiber & Kotz, 2003). It appears to be a better fit for the data than the Gompertz distribution from the original formulation of RMTPP.

Moreover, for FNN and MLP/MC decoders, we found that the last Softplus activation prevents the gradient of the GCIF to take large values for very short inter-arrival times. However, in many of the considered datasets, most events occur in packs during extremely short time spans, requiring the GCIF to change quickly between two events. The LTO encoding in combination with the Softplus activation allows steeper gradients for short inter-arrival times and thus enables rapid changes in the GCIF. As a result, the FNN and MLP/MC decoders can reach lower NLL-T with the LTO encoding.

(2) Including a mark representation in the event encoding generally improves performance in terms of NLL-T and PCE when moving from TO to CONCAT and from LTO to LCONCAT. On the one hand, this observation suggests that information contained in previous marks does help the model to better estimate the arrival times of future events. However, we observe that in most cases, TEMWL and LEWL encodings show higher NLL-T compared to their TE and LE counterparts. Therefore, relevant information contained in previous marks appears less readily exploitable by the models when the mark embedding is concatenated to a vectorial representation of time.

All decoders improve substantially with respect to the NLL-M, ECE, and F1-score metrics when the mark is included in the event encoding. While expected, this finding confirms that expressive representations of past marks are paramount to predicting future marks.

**Analysis of the history encoder.** From the results of Table 5, we observe that models equipped with a GRU history encoder yield overall better performance with respect to all time and mark-specific metrics compared to ones equipped with a SA encoder. While self-attention mechanisms have gained increasing popularity since their introduction by (Vaswani et al., 2017) for sequence modeling tasks, we found that they are on average less suited than their RNN counterparts in the context of TPPs. Specifically, the GRU encoder is more stable with respect to the choice of event encoding mechanism, while the SA encoder requires vectorial event representation to achieve good performance. Furthermore, the constant (CONS) history-independent encoder systematically achieves the worst results with respect to all metrics. This observation confirms the common assumption that future event occurrences associated with real-world TPPs indeed depend on past arrival times and marks, and hence better predictive accuracy can be achieved with better representations of the observed history.

**Analysis of the decoder.** On Table 6, we report for each decoder separately the combination that yielded the best performance with respect to the NLL-T (top rows) and the NLL-M (bottom rows), on average across all marked datasets. While we previously explained that some variations of event encoding and history encoders worked on average better for a given decoder, it does not necessarily mean that the best combination includes that variation. Moreover, we find that a single combination does not perform equally well on both metrics separately. In the following, we will thus focus our discussion on the top-row models for time-related metrics (NLL-T, PCE), and on bottom-row models for the mark-related ones (NLL-M, ECE, F1-score).

As a first observation, we note that LNM achieves the lowest NLL-T, outperforming all other baselines. The difference in performance with the LN decoder further suggests that the assumption of log-normality for the distribution of the inter-arrival times is not a sufficient inductive bias by itself and that the additional flexibility granted by the mixture is necessary to achieve a lower NLL-T. Additionally, most neural baselines achieve on average a lower NLL-T than their parametric counterparts, indicating that high-capacity models are more amenable to capturing complex patterns in real-world event data.

Table 4: Average and median scores, as well as average ranks per decoder and variation of event encoding, for marked datasets. Best results are highlighted in bold. Among others, our key insights include that (1) the SA/MC and SA/CM decoders achieve significantly lower NLL-T and PCE when combined with vectorial representations of time (i.e. TEM and LE) compared to the TO and LTO encodings, (2) in comparison to the TO encoding, LTO significantly improves performance on the NLL-T and PCE for the RMTPP, FNN and MLP/MC decoders, and (3) including a representation of past observed marks when encoding the history (CONCAT, TEMWL, LEWL) is paramount to achieve good performance with respect to the NLL-M, ECE and F1-score for all decoders. Refer to text for more details, and to Section 5.5 for details on the aggregation procedure.

| | NLL-T | | | PCE | | | NLL-M | | | ECE | | | F1-score | | |
|---|---|---|---|---|---|---|---|---|---|---|---|---|---|---|---|
| | Mean | Median | Rank | Mean | Median | Rank | Mean | Median | Rank | Mean | Median | Rank | Mean | Median | Rank |
| EC-TO | 0.59 | 0.56 | 6.5 | 0.2 | 0.23 | 6.25 | 0.19 | 0.21 | 6.5 | 0.44 | 0.44 | 7.12 | 0.19 | 0.17 | 6.88 |
| EC-LTO | 0.7 | 0.57 | 7.25 | 0.2 | 0.23 | 6.38 | 0.15 | 0.17 | 5.5 | 0.44 | 0.45 | 7.0 | 0.19 | 0.17 | **7.25** |
| EC-CONCAT | 0.13 | 0.14 | **2.5** | **0.17** | **0.15** | **2.25** | -0.51 | -0.75 | 2.75 | 0.34 | 0.31 | **1.88** | **0.3** | 0.28 | 2.12 |
| EC-LCONCAT | 0.36 | 0.29 | 4.25 | 0.19 | 0.2 | 4.75 | -0.45 | -0.66 | 3.5 | 0.34 | 0.3 | 2.75 | 0.28 | 0.27 | 2.88 |
| EC-TEM | 0.2 | 0.17 | 4.38 | 0.18 | 0.15 | 4.5 | 0.18 | 0.2 | 6.38 | 0.42 | 0.42 | 6.12 | 0.2 | 0.18 | 5.88 |
| EC-TEMWL | 1.48 | 0.3 | 4.25 | 0.19 | 0.17 | 4.88 | **-0.67** | **-0.78** | **2.0** | **0.33** | **0.27** | 2.38 | 0.3 | **0.29** | 1.88 |
| EC-LE | **0.05** | **0.11** | 3.0 | 0.18 | 0.15 | 2.88 | 0.15 | 0.15 | 5.38 | 0.41 | 0.42 | 5.38 | 0.2 | 0.18 | 5.75 |
| EC-LEWL | 0.17 | 0.19 | 3.88 | 0.18 | 0.15 | 4.12 | -0.33 | -0.59 | 4.0 | 0.35 | 0.35 | 3.38 | 0.29 | 0.28 | 3.38 |
| LNM-TO | -0.95 | -0.57 | 6.0 | **0.02** | 0.02 | 5.38 | 0.06 | 0.11 | 6.88 | 0.44 | 0.45 | 6.88 | 0.19 | 0.17 | **7.62** |
| LNM-LTO | -0.58 | -0.59 | 6.25 | 0.02 | **0.01** | 5.5 | 0.03 | 0.06 | 6.38 | 0.44 | 0.46 | 7.25 | 0.19 | 0.17 | 6.5 |
| LNM-CONCAT | **-1.14** | **-0.98** | **2.0** | 0.02 | 0.01 | 4.0 | -2.25 | -1.47 | **1.38** | 0.25 | 0.21 | **1.75** | **0.39** | **0.35** | 1.5 |
| LNM-LCONCAT | -0.83 | -0.83 | 2.62 | 0.02 | 0.01 | **2.12** | -1.87 | -1.18 | 2.88 | 0.29 | 0.25 | 2.75 | 0.34 | 0.31 | 2.88 |
| LNM-TEM | -0.87 | -0.79 | 4.75 | 0.02 | 0.01 | 4.5 | 0.03 | 0.05 | 6.62 | 0.42 | 0.44 | 5.75 | 0.2 | 0.18 | 6.0 |
| LNM-TEMWL | -0.83 | -0.87 | 4.75 | 0.03 | 0.02 | 4.12 | -1.99 | **-1.49** | 2.25 | 0.28 | 0.25 | 2.12 | 0.36 | 0.32 | 2.25 |
| LNM-LE | -1.11 | -0.78 | 5.12 | 0.02 | 0.02 | 6.12 | 0.0 | 0.02 | 6.12 | 0.42 | 0.44 | 6.12 | 0.2 | 0.18 | 5.88 |
| LNM-LEWL | -0.95 | -0.92 | 4.5 | 0.02 | 0.01 | 4.25 | -1.89 | -1.18 | 3.5 | 0.3 | 0.28 | 3.38 | 0.32 | 0.29 | 3.38 |
| FNN-TO | 0.96 | 0.68 | 4.5 | 0.21 | 0.28 | 4.5 | 0.3 | 0.34 | 4.5 | 0.46 | 0.47 | 5.25 | 0.17 | 0.17 | **5.25** |
| FNN-LTO | -0.32 | -0.31 | 2.0 | **0.03** | **0.02** | 2.12 | 0.05 | 0.01 | 3.75 | 0.4 | 0.42 | 2.25 | 0.21 | 0.19 | 2.62 |
| FNN-CONCAT | 0.86 | 0.45 | 4.38 | 0.21 | 0.27 | 4.5 | 0.06 | 0.06 | 3.5 | 0.43 | 0.46 | 3.5 | 0.19 | 0.19 | 3.0 |
| FNN-LCONCAT | **-0.57** | **-0.66** | **1.0** | 0.03 | 0.02 | **1.62** | **-0.99** | **-0.69** | 2.12 | **0.31** | 0.35 | **1.75** | **0.26** | **0.23** | 1.38 |
| FNN-LE | 0.87 | 0.53 | 4.25 | 0.21 | 0.29 | 3.75 | 0.27 | 0.31 | 4.12 | 0.46 | 0.47 | 4.25 | 0.17 | 0.16 | 4.62 |
| FNN-LEWL | 0.95 | 0.41 | 4.88 | 0.21 | 0.29 | 4.5 | 0.14 | 0.12 | 3.62 | 0.45 | 0.46 | 4.0 | 0.18 | 0.18 | 4.12 |
| MLP/MC-TO | 0.11 | 0.22 | 6.0 | 0.16 | 0.16 | 6.38 | 0.36 | 0.33 | 6.75 | 0.41 | 0.4 | 6.75 | 0.2 | 0.18 | **7.25** |
| MLP/MC-LTO | -0.16 | -0.14 | 4.0 | **0.09** | **0.06** | **1.88** | 0.44 | 0.25 | 6.75 | 0.39 | 0.42 | 6.0 | 0.21 | 0.18 | 6.0 |
| MLP/MC-CONCAT | 0.0 | 0.06 | 4.88 | 0.16 | 0.13 | 5.62 | -0.16 | -0.26 | **2.62** | 0.33 | 0.34 | 3.38 | 0.28 | **0.29** | 3.12 |
| MLP/MC-LCONCAT | **-0.34** | **-0.28** | 2.25 | 0.09 | 0.07 | 2.0 | -0.19 | -0.42 | 3.25 | **0.29** | 0.28 | **2.38** | 0.28 | 0.28 | 2.88 |
| MLP/MC-TEM | 0.01 | -0.01 | 5.5 | 0.15 | 0.13 | 5.75 | 0.26 | 0.23 | 5.5 | 0.4 | 0.41 | 6.38 | 0.2 | 0.18 | 6.5 |
| MLP/MC-TEMWL | 0.29 | 0.36 | 7.25 | 0.18 | 0.17 | 7.62 | **-0.45** | **-0.45** | 2.62 | 0.34 | 0.33 | 2.75 | **0.3** | 0.29 | 2.5 |
| MLP/MC-LE | -0.2 | -0.19 | 3.12 | 0.13 | 0.11 | 3.12 | 0.37 | 0.22 | 5.62 | 0.38 | 0.37 | 6.0 | 0.21 | 0.19 | 5.5 |
| MLP/MC-LEWL | -0.18 | -0.16 | 3.0 | 0.13 | 0.11 | 3.62 | -0.42 | -0.35 | 2.88 | 0.29 | **0.26** | 2.38 | 0.3 | 0.28 | 2.25 |
| RMTPP-TO | 0.15 | 0.23 | 6.75 | 0.16 | 0.16 | 6.75 | 0.11 | 0.12 | 6.25 | 0.42 | 0.43 | 7.12 | 0.19 | 0.18 | **7.12** |
| RMTPP-LTO | -0.19 | -0.19 | 4.62 | 0.06 | 0.05 | 3.25 | 0.09 | 0.07 | 6.25 | 0.42 | 0.42 | 7.0 | 0.19 | 0.18 | 6.38 |
| RMTPP-CONCAT | -0.05 | -0.05 | 4.0 | 0.15 | 0.12 | 4.12 | -1.52 | -1.28 | 2.88 | 0.26 | 0.24 | 2.5 | 0.35 | 0.33 | 2.5 |
| RMTPP-LCONCAT | **-0.42** | **-0.34** | 2.25 | **0.04** | **0.04** | 2.0 | -1.65 | **-1.51** | 3.0 | 0.25 | **0.21** | 2.12 | 0.37 | 0.34 | 2.0 |
| RMTPP-TEM | -0.01 | 0.01 | 5.5 | 0.15 | 0.12 | 5.25 | 0.1 | 0.15 | 6.0 | 0.41 | 0.41 | 6.25 | 0.21 | 0.19 | 6.5 |
| RMTPP-TEMWL | 0.0 | 0.05 | 4.5 | 0.15 | 0.14 | 5.0 | **-2.1** | -1.42 | **1.5** | **0.24** | 0.21 | **1.88** | **0.4** | **0.37** | 1.62 |
| RMTPP-LE | -0.14 | 0.01 | 3.88 | 0.15 | 0.13 | 4.75 | 0.12 | 0.14 | 6.25 | 0.4 | 0.41 | 5.62 | 0.21 | 0.19 | 6.0 |
| RMTPP-LEWL | -0.03 | -0.0 | 4.5 | 0.15 | 0.13 | 4.88 | -1.19 | -1.03 | 3.88 | 0.3 | 0.29 | 3.5 | 0.32 | 0.32 | 3.88 |
| SA/CM-TO | 10.78 | 0.12 | 4.75 | 0.11 | 0.06 | 4.12 | 1.14 | 0.31 | 4.75 | 0.45 | 0.46 | 5.12 | 0.14 | 0.1 | **4.75** |
| SA/CM-LTO | 20.39 | -0.01 | 3.0 | 0.1 | 0.05 | 2.75 | 0.66 | 0.11 | 4.0 | 0.43 | **0.44** | 2.88 | 0.18 | 0.1 | 2.88 |
| SA/CM-CONCAT | -0.08 | -0.09 | 3.38 | 0.07 | 0.05 | 3.25 | 0.6 | 0.16 | 4.0 | 0.44 | 0.44 | 3.38 | 0.17 | **0.17** | 3.0 |
| SA/CM-LCONCAT | 20.53 | 0.05 | 4.12 | 0.12 | 0.06 | 3.62 | 0.61 | 0.08 | **2.5** | 0.43 | 0.45 | 3.25 | 0.18 | 0.11 | 3.0 |
| SA/CM-LE | **-0.37** | **-0.4** | **2.0** | **0.05** | **0.04** | **2.38** | 0.09 | 0.07 | 3.0 | 0.42 | 0.44 | 3.62 | 0.19 | 0.17 | 4.12 |
| SA/CM-LEWL | 2.14 | -0.07 | 3.75 | 0.09 | 0.08 | 4.88 | **-0.0** | **-0.01** | 2.75 | **0.41** | 0.44 | **2.75** | **0.2** | 0.16 | 3.25 |
| SA/MC-TO | 1.22 | 1.0 | 7.38 | 0.23 | 0.31 | 6.75 | 0.24 | 0.31 | 5.12 | 0.47 | 0.47 | 7.12 | 0.17 | 0.16 | **7.5** |
| SA/MC-LTO | 1.07 | 0.82 | 5.75 | 0.22 | 0.29 | 5.88 | 0.28 | 0.4 | 6.25 | 0.46 | 0.47 | 6.0 | 0.17 | 0.16 | 6.38 |
| SA/MC-CONCAT | 0.64 | 0.85 | 6.5 | 0.19 | 0.17 | 6.25 | 0.1 | 0.2 | 5.0 | 0.44 | 0.47 | 6.25 | 0.2 | 0.17 | 5.88 |
| SA/MC-LCONCAT | 0.53 | 0.74 | 5.12 | 0.19 | 0.16 | 6.25 | 0.16 | 0.23 | 5.38 | 0.43 | 0.47 | 5.25 | 0.2 | 0.17 | 5.5 |
| SA/MC-TEM | -0.42 | -0.4 | 3.25 | 0.1 | 0.07 | 2.75 | 0.15 | 0.05 | 5.12 | 0.39 | 0.4 | 4.25 | 0.2 | 0.18 | 4.12 |
| SA/MC-TEMWL | -0.28 | -0.26 | 4.25 | 0.11 | 0.08 | 4.25 | **-0.51** | **-0.34** | **1.88** | 0.32 | 0.32 | **1.88** | **0.28** | **0.25** | 1.5 |
| SA/MC-LE | **-0.76** | **-0.52** | **1.62** | **0.07** | **0.04** | **1.5** | 0.11 | -0.03 | 4.88 | 0.37 | 0.38 | 3.38 | 0.21 | 0.19 | 3.38 |
| SA/MC-LEWL | -0.48 | -0.45 | 2.12 | 0.08 | 0.07 | 2.38 | -0.29 | -0.23 | 2.38 | 0.34 | 0.32 | 1.88 | 0.25 | 0.24 | 1.75 |

Table 5: Average and median scores, as well as average ranks per decoder and variation of history encoder, for marked datasets. Best results are highlighted in bold. The key insight parsed from this Table is that models equipped with a GRU encoder (i.e. GRU-∗) show overall improved performance with respect to all metrics compared to ones equipped with a self-attention encoder (i.e. SA-∗). Refer to text for more details, and to Section 5.5 for details on the aggregation procedure.

| | Marked Datasets | | | | | | | | | | | | | | |
|---|---|---|---|---|---|---|---|---|---|---|---|---|---|---|---|
| | NLL-T | | | PCE | | | NLL-M | | | ECE | | | F1-score | | |
| | Mean | Median | Rank | Mean | Median | Rank | Mean | Median | Rank | Mean | Median | Rank | Mean | Median | Rank |
| CONS-EC | 1.74 | 1.34 | 3.0 | 0.3 | 0.35 | 2.83 | 0.63 | 0.56 | 2.67 | 0.48 | 0.49 | 3.0 | 0.14 | 0.06 | 3.0 |
| SA-EC | 0.73 | 0.63 | 2.0 | 0.25 | 0.27 | 2.17 | 0.04 | -0.04 | 2.0 | 0.43 | 0.42 | 2.0 | 0.2 | 0.15 | 1.83 |
| GRU-EC | **0.19** | **0.14** | **1.0** | **0.22** | **0.23** | **1.0** | **-0.17** | **-0.29** | **1.33** | **0.39** | **0.38** | **1.0** | **0.22** | **0.17** | **1.17** |
| CONS-LNM | -0.35 | -0.44 | 2.83 | **0.02** | 0.02 | 2.33 | 0.58 | 0.43 | 3.0 | 0.48 | 0.49 | 3.0 | 0.14 | 0.06 | 3.0 |
| SA-LNM | -0.61 | -0.68 | 1.83 | 0.02 | 0.02 | 2.17 | -0.63 | -0.39 | 1.83 | 0.39 | 0.39 | 1.83 | 0.24 | 0.17 | 1.83 |
| GRU-LNM | **-0.8** | **-0.83** | **1.33** | 0.03 | **0.01** | 1.5 | **-1.11** | **-0.83** | **1.17** | **0.35** | **0.37** | **1.17** | **0.26** | **0.2** | **1.17** |
| CONS-FNN | 1.02 | 0.72 | 3.0 | 0.2 | 0.24 | 2.67 | 0.5 | 0.25 | 2.83 | 0.46 | 0.46 | 2.83 | 0.17 | **0.11** | 2.5 |
| SA-FNN | 0.78 | 0.53 | 1.83 | 0.19 | 0.23 | 2.0 | 0.05 | 0.06 | 2.17 | 0.45 | 0.45 | 1.83 | **0.18** | 0.11 | 2.0 |
| GRU-FNN | **0.52** | **0.13** | **1.17** | **0.18** | **0.21** | **1.33** | **-0.17** | **-0.04** | **1.0** | **0.42** | **0.44** | **1.33** | 0.18 | 0.11 | **1.5** |
| CONS-MLP/MC | 0.5 | 0.52 | 3.0 | 0.19 | 0.23 | 2.83 | 0.4 | 0.3 | 2.33 | 0.39 | 0.4 | 2.0 | 0.22 | 0.17 | 2.33 |
| SA-MLP/MC | 0.13 | 0.13 | 2.0 | 0.17 | 0.21 | 2.0 | 0.04 | -0.02 | 2.33 | 0.38 | 0.39 | 2.67 | 0.22 | 0.17 | 2.33 |
| GRU-MLP/MC | **-0.15** | **-0.14** | **1.0** | **0.16** | **0.18** | **1.17** | **-0.03** | **-0.26** | **1.33** | **0.35** | **0.37** | **1.33** | **0.23** | **0.18** | **1.33** |
| CONS-RMTPP | 0.87 | 0.66 | 3.0 | 0.18 | 0.21 | 2.33 | 0.97 | 0.65 | 2.67 | 0.47 | 0.48 | 3.0 | 0.13 | 0.05 | 3.0 |
| SA-RMTPP | 0.13 | 0.17 | 1.83 | 0.17 | 0.19 | 2.17 | -0.38 | -0.36 | 2.17 | 0.38 | 0.39 | 1.83 | 0.24 | 0.18 | 1.83 |
| GRU-RMTPP | **-0.15** | **-0.17** | **1.17** | **0.15** | **0.17** | 1.5 | **-0.95** | **-0.66** | **1.17** | **0.34** | **0.37** | **1.17** | **0.26** | **0.22** | **1.17** |
| CONS-SA/CM | 0.01 | -0.06 | 3.0 | 0.1 | 0.12 | 2.83 | 0.51 | 0.36 | 2.67 | **0.44** | **0.44** | 2.17 | **0.18** | **0.11** | 2.17 |
| SA-SA/CM | -0.17 | -0.2 | 1.83 | 0.08 | **0.08** | 1.83 | **0.12** | 0.09 | 1.5 | 0.44 | 0.44 | **1.83** | 0.18 | 0.11 | **1.17** |
| GRU-SA/CM | **-0.22** | **-0.22** | **1.17** | **0.07** | 0.08 | 1.33 | 0.12 | **0.07** | 1.83 | 0.44 | 0.44 | 2.0 | 0.18 | 0.11 | 2.67 |
| CONS-SA/MC | 0.44 | 0.2 | 3.0 | 0.18 | 0.19 | 2.83 | 0.42 | 0.28 | 2.67 | 0.41 | **0.41** | 2.33 | **0.2** | **0.13** | 2.0 |
| SA-SA/MC | 0.07 | 0.03 | 1.67 | **0.16** | **0.16** | 1.83 | -0.05 | **-0.16** | 1.5 | **0.39** | 0.41 | **1.67** | 0.19 | 0.12 | 2.33 |
| GRU-SA/MC | **0.05** | **-0.04** | **1.33** | 0.16 | 0.16 | **1.33** | **-0.1** | -0.16 | 1.83 | 0.39 | 0.42 | 2.0 | 0.2 | 0.12 | **1.67** |

Figure 3 shows $\lambda^*(t)$ (orange) and $f^*(t)$ (green) between two events in a test sequence in LastFM, for the combinations of Table 6 that performed the best on the NLL-T. The greater the gap between $\lambda^*(t)$ and $f^*(t)$, the higher the cumulative GCIF between two events. We observe that most decoders learn to assign a high probability to very low inter-arrival times, which allows them to reach low NLL-T on datasets where events are highly clustered, such as LastFM. This behavior is only possible if the GCIF is allowed to vary rapidly once a new event is observed. As discussed previously, for decoders employing a Softplus activation, such as MLP/MC, FNN, or SA/CM, a steeper gradient for the GCIF can be obtained at low inter-arrival times by using the LTO or LCONCAT encodings.

With respect to the mark prediction task, we find that the Hawkes decoder achieves the overall best results, outperforming all baselines in terms of NLL-M, ECE, and F1-score. While the Hawkes decoder did not perform favorably on time-related metrics, its clear superiority compared to more complex models on the mark prediction task is rather intriguing. Likewise, the RMTPP and LN(M) decoders also show competitive performance on these metrics, despite their simplifying assumption of marks being independent of the time given the history of the process. The overall superiority of parametric and semi-parametric architectures on the mark prediction task suggests that non-parametric decoders may actually suffer from their high flexibility, making them hard to optimize in practice. This particularly stands out for the NH decoder which performs extremely badly on all metrics, only marginally outperforming the Poison decoder. Nonetheless, although LN, LNM,

Table 6: Average and median scores, and average ranks of the best combinations per decoder on the NLL-T (top rows) and NLL-M (bottom row) across all marked datasets. Best results are highlighted in bold.

| | Marked Datasets | | | | | | | | | | | | | | |
|---|---|---|---|---|---|---|---|---|---|---|---|---|---|---|---|
| | NLL-T | | | PCE | | | NLL-M | | | ECE | | | F1-score | | |
| | Mean | Median | Rank | Mean | Median | Rank | Mean | Median | Rank | Mean | Median | Rank | Mean | Median | Rank |
| GRU-EC-LE | -0.22 | -0.02 | 7.75 | 0.17 | 0.14 | 9.12 | -0.1 | -0.03 | 6.5 | 0.39 | 0.41 | 7.0 | 0.21 | 0.18 | 7.38 |
| GRU-LNM-TO | **-1.44** | **-0.93** | **1.5** | **0.02** | **0.01** | **1.88** | -0.06 | -0.07 | 6.5 | 0.41 | 0.42 | 6.75 | 0.21 | 0.18 | 7.12 |
| GRU-LN-LEWL | -0.54 | -0.56 | 5.25 | 0.07 | 0.04 | 5.75 | -1.56 | -1.38 | 2.75 | 0.28 | 0.28 | 4.5 | 0.33 | 0.35 | 2.88 |
| GRU-FNN-LCONCAT | -0.6 | -0.71 | 3.38 | 0.02 | 0.02 | 2.25 | -1.57 | -1.0 | 3.5 | 0.29 | 0.35 | 4.5 | 0.27 | 0.23 | 4.12 |
| GRU-MLP/MC-LCONCAT | -0.42 | -0.38 | 5.5 | 0.09 | 0.07 | 6.38 | -0.47 | -0.47 | 5.62 | 0.26 | 0.26 | 2.88 | 0.28 | 0.28 | 4.38 |
| GRU-RMTPP-LCONCAT | -0.52 | -0.41 | 4.75 | 0.04 | 0.03 | 4.88 | -1.88 | **-1.99** | **2.38** | 0.22 | 0.15 | 2.88 | 0.4 | 0.35 | 1.62 |
| GRU-SA/CM-LE | -0.46 | -0.42 | 6.12 | 0.05 | 0.03 | 3.75 | 0.1 | 0.07 | 7.75 | 0.43 | 0.45 | 8.12 | 0.19 | 0.18 | 8.75 |
| GRU-SA/MC-LE | -0.85 | -0.55 | 3.88 | 0.08 | 0.05 | 4.88 | 0.08 | -0.04 | 7.62 | 0.35 | 0.37 | 6.5 | 0.21 | 0.2 | 6.62 |
| Hawkes | 1.21 | -0.15 | 7.62 | 0.11 | 0.1 | 6.5 | **-3.42** | -1.01 | 4.12 | **0.15** | **0.13** | **2.12** | **0.45** | **0.43** | 2.38 |
| Poisson | 1.86 | 1.35 | 10.75 | 0.25 | 0.34 | 10.25 | 2.23 | 1.26 | 10.75 | 0.48 | 0.48 | 10.5 | 0.16 | 0.15 | 10.75 |
| NH | 1.32 | 1.07 | 9.5 | 0.24 | 0.32 | 10.38 | 0.25 | 0.41 | 8.5 | 0.48 | 0.47 | 10.25 | 0.17 | 0.16 | 10.75 |
| GRU-EC-TEMWL | 0.16 | 0.16 | 7.88 | 0.18 | 0.16 | 8.62 | -1.42 | -1.1 | 5.0 | 0.29 | 0.24 | 5.75 | 0.3 | 0.3 | 6.12 |
| GRU-LNM-CONCAT | **-1.41** | **-1.0** | **1.12** | **0.02** | **0.01** | **1.62** | -2.62 | -1.76 | 2.5 | 0.23 | 0.18 | 3.88 | 0.38 | 0.34 | 3.0 |
| GRU-LN-CONCAT | -0.51 | -0.57 | 4.25 | 0.07 | 0.04 | 4.88 | -2.59 | -1.81 | 3.12 | 0.25 | 0.25 | 4.12 | 0.33 | 0.34 | 4.38 |
| GRU-FNN-LCONCAT | -0.6 | -0.71 | 2.75 | 0.02 | 0.02 | 2.25 | -1.57 | -1.0 | 5.25 | 0.29 | 0.35 | 4.88 | 0.27 | 0.23 | 5.12 |
| GRU-MLP/MC-TEMWL | 0.11 | 0.23 | 7.5 | 0.17 | 0.16 | 7.12 | -0.73 | -0.64 | 7.0 | 0.32 | 0.33 | 6.38 | 0.32 | 0.3 | 5.5 |
| GRU-RMTPP-TEMWL + B | -0.19 | -0.11 | 5.62 | 0.14 | 0.11 | 5.62 | -2.67 | **-1.85** | **2.38** | 0.2 | 0.16 | 2.88 | 0.42 | 0.42 | 2.62 |
| GRU-SA/CM-LEWL | -0.23 | -0.23 | 5.5 | 0.08 | 0.07 | 5.12 | -0.18 | -0.11 | 8.12 | 0.41 | 0.43 | 8.5 | 0.21 | 0.2 | 8.88 |
| GRU-SA/MC-TEMWL | -0.29 | -0.3 | 4.88 | 0.11 | 0.09 | 5.25 | -0.55 | -0.55 | 7.38 | 0.31 | 0.31 | 6.25 | 0.28 | 0.26 | 6.5 |

RMTPP, and Hawkes do better than their non-parametric counterparts, we find that all decoders perform rather poorly on the mark prediction task.

Figure 4 shows the CD diagrams between the average ranks of all decoders at the $\alpha = 0.1$ significance level, on each metric separately. The lower the rank (further to the left on the top horizontal axis), the better the performance of a decoder with respect to that metric, while a bold black line groups decoders that are not significantly different from one another. As can be seen, there are clear differences between the models' average ranks (e.g. LNM on the NLL-T for marked datasets). The fact that Holm's posthoc tests do not allow us to conclude statistically significant differences between them can be explained by a large number of pairwise comparisons compared to the few available samples (datasets), which results in high adjusted p-values.

**On the calibration of TPP models.** Figure 5 shows the reliability diagrams for the inter-arrival time predictive distributions associated with the combinations presented in Table 6, averaged over all marked datasets. We observe a good probabilistic calibration for the LNM, FNN, MLP/MC, RMTPP, and SA/CM decoders, as demonstrated by their curves closely matching the diagonal. On the other hand, the other decoders present higher degrees of miscalibration. More precisely, their empirical CDF lays systematically above the diagonal line, which suggests that the predictive distributions attribute little probability mass to low inter-arrival times. This observation supports our argument about the behavior of $f^*(\tau)$ on Figure 3, i.e. the conditional density of calibrated models attributes most of the mass to short inter-arrival times.

Figure 6 shows the reliability diagrams for the mark distributions associated with the best-performing combinations (in terms of the NLL-M) in Table 6, aggregated over all marked datasets. Overall, we observe that the calibration of these decoders mirrors their performance on the NLL-M and F1-scores, i.e. better scores with respect to these metrics correspond to better calibration. Indeed, the Hawkes decoder is the best-calibrated model among our baselines, followed by RMTPP

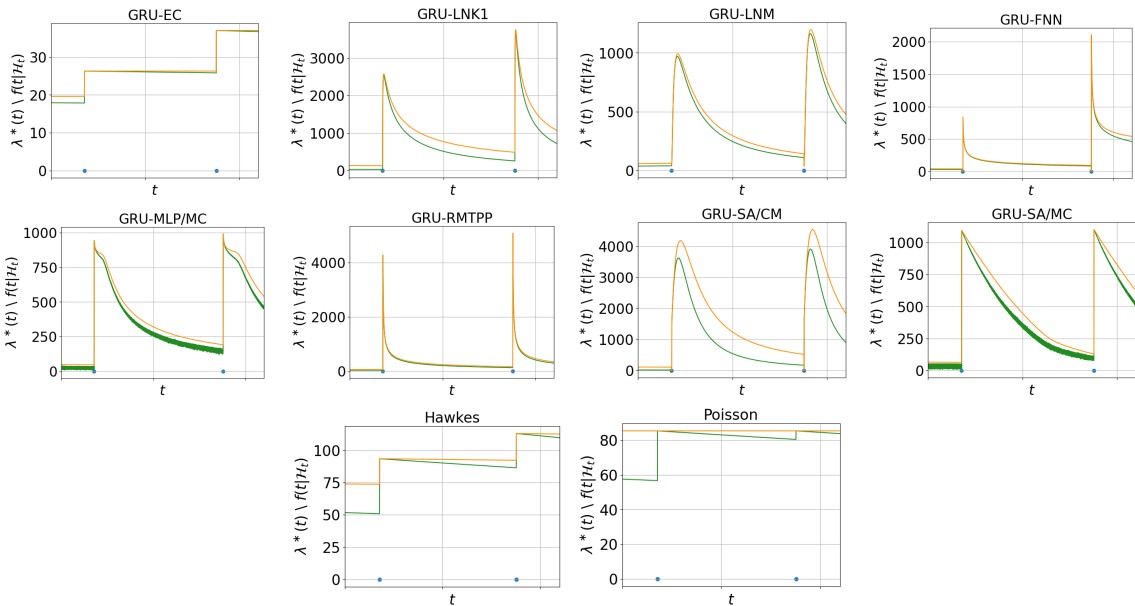

Figure 3: Evolution of $\lambda^*(t)$ (orange) and $f(t|\mathcal{H}_t)$ (green) between two events on the LastFM dataset. Arrival times correspond to 2.0975 and 2.0986, respectively.

and LNM. Nonetheless, we note that all decoders are usually over-confident in their predictions, as shown by the bin accuracies falling systematically below the diagonal line.

**Analysis of the history size.** When training a neural TPP model, it is typically assumed that the complete history $\mathcal{H}t_i$, spanning from the first to the last observed event, contains valuable information about the future dynamics of the process. In other words, the joint distribution of an event $e_i = (t_i, k_i)$ is commonly modeled as dependent on the entire history of the process up to $e_i$, denoted as $f(e_i|\mathcal{H}_{t_i}) = f(e_i|e_{i-1}, \ldots, e_1)$. However, in certain real-world scenarios, it may be more appropriate to explicitly assume a $q$-order Markovian property, where only $q$ of the last $i-1$ events preceding $e_i$ actually influence the distribution of $e_i$.

When comparing the impact of different encoders, we have found that effectively encoding the process history is crucial for modeling the joint distribution. However, it does not provide insights into the number of past events that truly contain useful information. To address this question, we train models using variations of both the GRU and self-attention history encoders. These encoders generate a history embedding $\boldsymbol{h}_i$ specifically for event $e_i$ using $\mathcal{H}^q_{t_i} = \{(t_j, k_j) \in \mathcal{H}|t_j < t_i, \ i - q \leq j < i\}$, which includes at most $q$ events preceding $e_i$.

In Figure 7, we show the evolution of the NLL-T and NLL-M (when relevant), by varying the maximal history size $q$ to which the history encoder has access to during training. We report our observations for the GRU encoder operating on this fixed-size window (GRU-Fixed), but similar results were obtained for a fixed self-attentive encoder (SA-Fixed). Compared to a completely masked history, which is equivalent to training the model with the CONS encoder (i.e. $q = 0$), we observe a substantial improvement in NLL-T for most models when only the last event is made

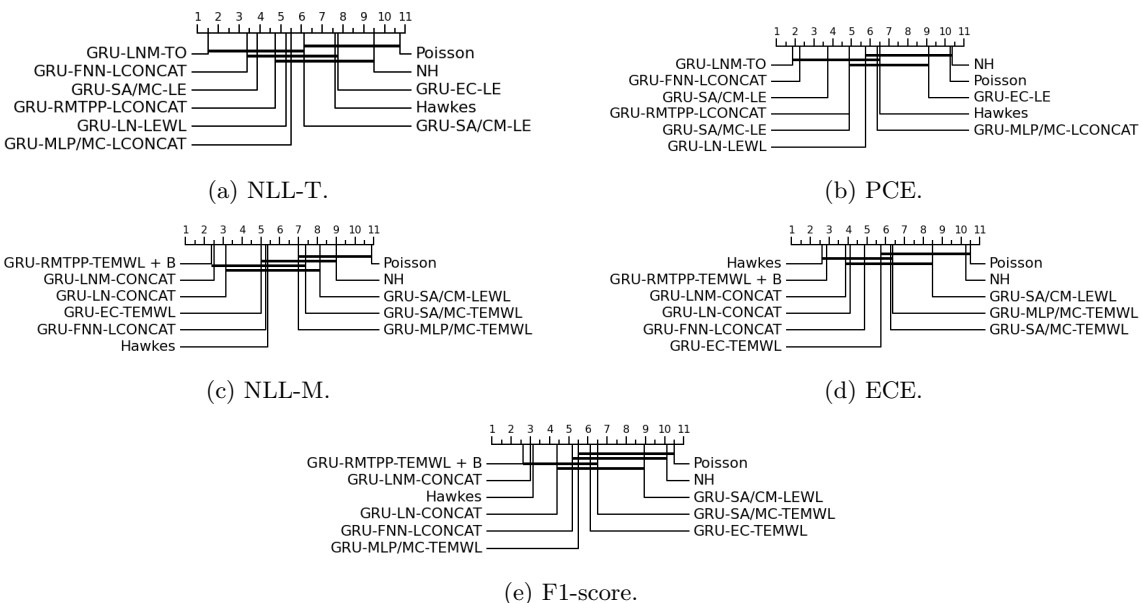

(a) NLL-T.

(b) PCE.

(c) NLL-M.

(d) ECE.

(e) F1-score.

Figure 4: Critical Distance (CD) diagrams per metric at the $\alpha = 0.10$ significance level for all decoders on marked datasets. The decoders' average ranks are displayed on top, and a bold line joins the decoders' variations that are not statistically different.

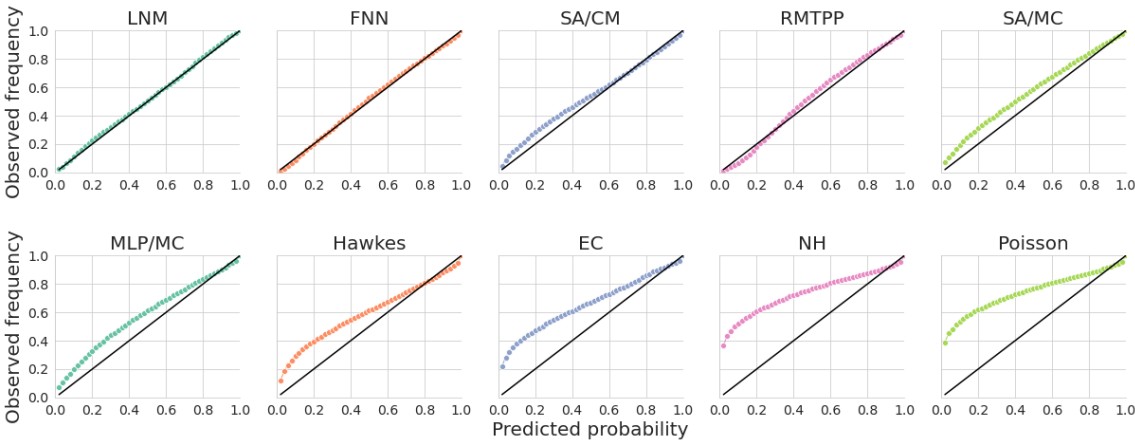

Figure 5: Reliability diagrams of the distribution of inter-arrival times for the decoder's combinations that performed the best on the NLL-T (top rows of Table 6), averaged over all marked datasets. The bold black line corresponds to perfect marginal calibration.

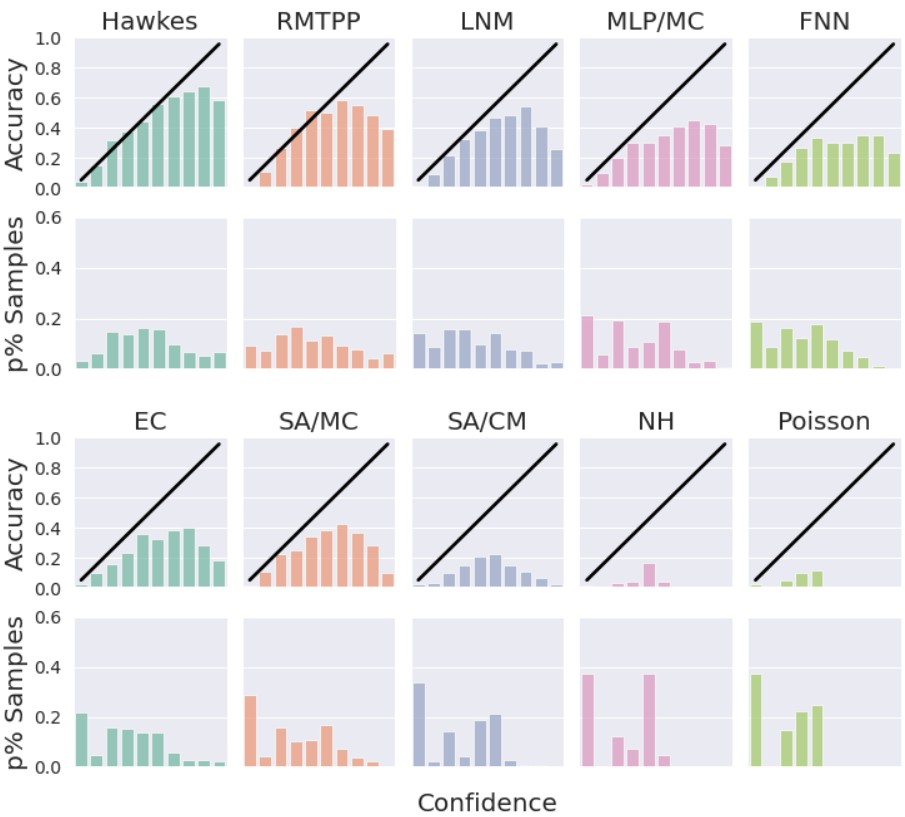

Figure 6: Reliability diagrams of the distribution of marks for various decoders, averaged over all marked datasets. Front rows depict a decoder's average accuracy per bin, while bottom rows show the average proportion of samples falling per bin. Bins aligning with the bold black line corresponds to perfect calibration.

available to the encoder. This indicates a process' past does indeed contain useful information. However, as observed on most datasets, additional context does not yield significant improvement, with an NLL-T that often quickly stabilizes as $q$ increases.

On the one hand, this finding suggests that real-world processes do possess a Markovian property and that going far in history does not bring valuable additional insight regarding the arrival time of the next event. On the other hand, it could also indicate that RNNs (and self-attention mechanisms) fail to capture dependencies among event occurrences in the context of TPPs and that more research is needed to design better alternatives to encode the history. In essence, masking part of the history translates to a reduction in model complexity, making them effectively less prone to overfitting on the training sequences. Additionally, the EC decoder appears to be more impacted on the NLL-T as $q$ increases than other baselines. Considering that this decoder can only leverage the history embedding to define the MCIF, we hypothesize that optimization may in this case force the encoder to extract as much information as possible from the patterns of previous marks occurrences.

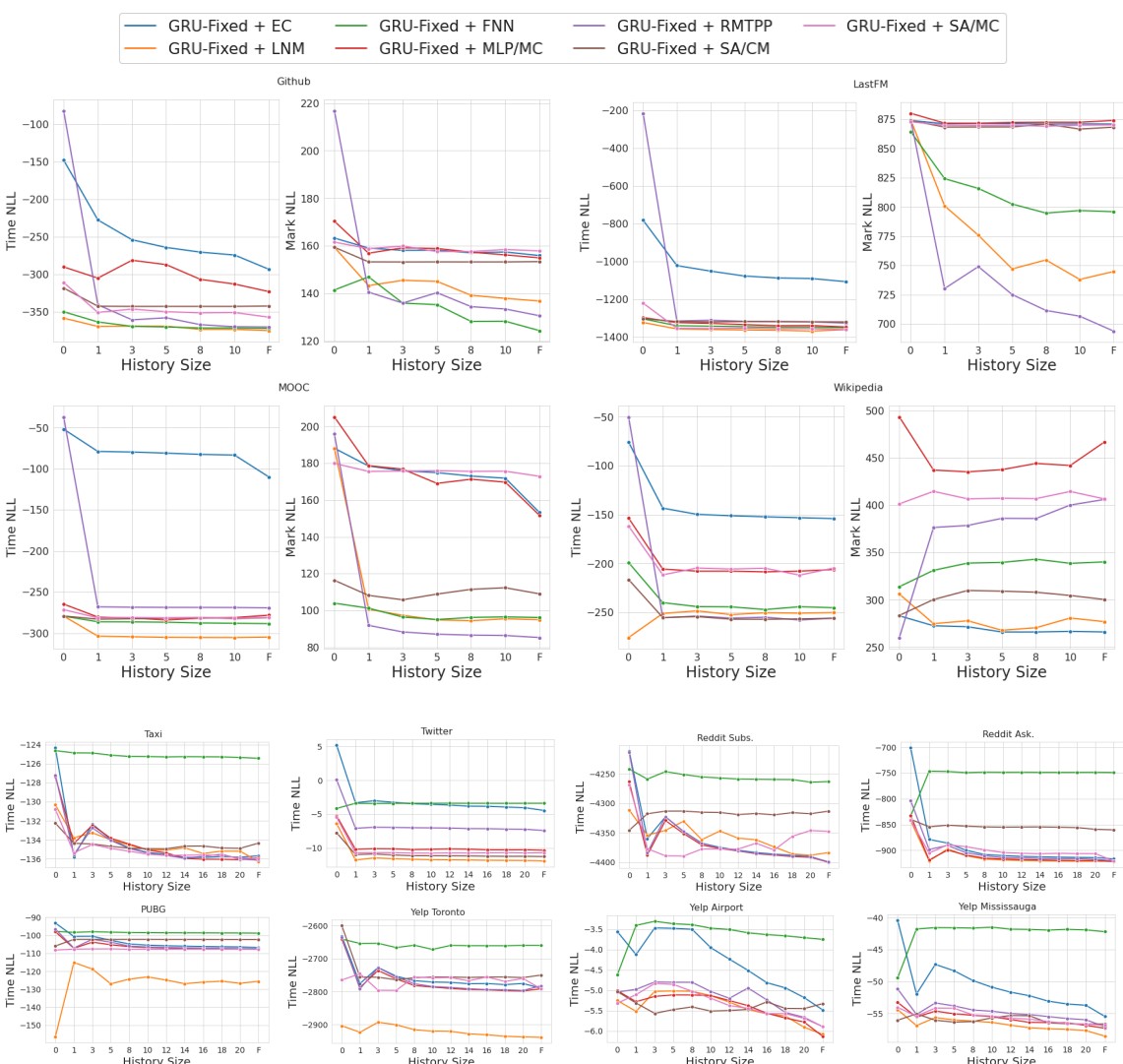

Figure 7: Evolution of models' performance with respect to the Time NLL, and the Mark NLL when available, as a function of the maximal number of events used to construct $h_i$ when using a GRU operating on a fixed-size window. 'F' refers to the unconstrained GRU, i.e. the encoder having access to the full history.

A similar statement can be made for the NLL-M. However, we find continuous improvement as $q$ increases for FNN, RMTPP, and LNM on Github and LastFM. As LNM and RMTPP parametrize the mark distribution solely from $h_i$, they also require expressive representations of the history to perform well with respect to the NLL-M.

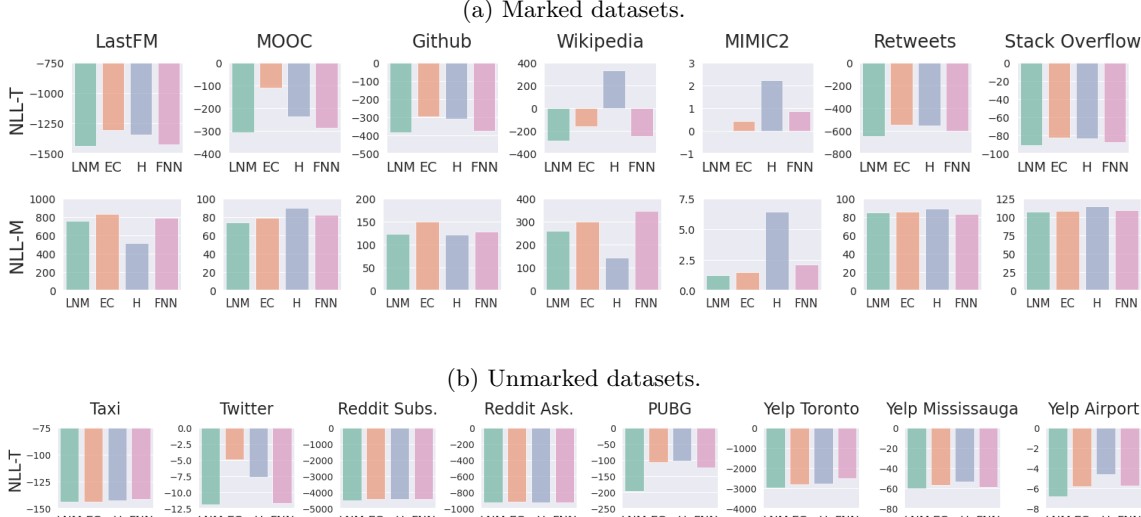

Figure 8: Performance OF LNM, EC, Hawkes (H), and SA/CM on the NLL-T and NLL-M for each dataset.

**On the adequacy of TPPs datasets.** A concern that is rarely addressed in the neural TPP literature relates to the validity of the current benchmark datasets for neural TPP models. Enguehard et al. (2020) raised some concerns regarding MIMIC2 and Stack Overflow as the simple time-independent EC decoder yielded competitive performance with more complex baselines on such datasets. Figure 8 reports both NLL-T and NLL-M (when available) for LNM, EC, Hawkes and FNN decoders[8] on all real-world datasets. We indeed observe that the EC decoder is competitive with LNM on the NLL-T and NLL-M for MIMIC2 and Stack Overflow, supporting their recommendation that future research should show a certain degree of caution when benchmarking new methods on these datasets. We further express additional concerns about Taxi, Reddit Subs and Reddit Ask Comments, Yelp Toronto, and Yelp Mississauga, on which all decoders achieve comparable performance. All remaining datasets appear to be appropriate benchmarks for evaluating neural TPP models, as higher variability is observed in the results.

**Computational times.** On Table 7, we report the computing time (in seconds) for each decoder averaged over 10 epochs for a single forward and backward pass on the training sequences of the MOOC dataset (standard deviation is given in parenthesis). All decoders were equipped with the GRU history encoder and the TEMWL event encoding, when relevant (i.e. not for Hawkes, Poisson, and NH), with similar hyper-parameters configurations. Additionally, the batch size was 32, and the models were trained on an NVIDIA RTX A5000. With respect to the results on both the time and mark prediction tasks, we find LNM to provide the best performance/runtime trade-off among all the decoders considered in the study. On the other hand, the NH decoder is extremely expensive to train, while generally performing worse than other neural baselines.

---

[8]Here also, we employed the combinations on the top row of Table 6 to compute the NLL-T on marked datasets, and the combinations of Table 10 to compute the NLL-T on unmarked datasets. The NLL-M was computed using the combinations on the bottom row of Table 6.

Table 7: Average execution time in seconds per decoder for a single forward and backward pass on the MOOC dataset.

| EC | LNM | MLP/MC | FNN | RMTPP | SA/CM | SA/MC | Hawkes | Poisson | NH |
|---|---|---|---|---|---|---|---|---|---|
| 1.21 (0.06) | 1.51 (0.06) | 1.66 (0.07) | 2.39 (0.07) | 1.32 (0.07) | 4.01 (0.08) | 2.47 (0.06) | 17.11 (0.27) | 0.66 (0.01) | 234.13 (1.36) |

## 7 Conclusion

We conducted a large-scale empirical study of state-of-the-art neural TPP models using multiple real-world and synthetic event sequence datasets in a carefully designed unified setup. Specifically, we studied the influence of major architectural components on predictive accuracy and highlighted that some specific combinations of architectural components can lead to significant improvements for both time and mark prediction tasks. Moreover, we assessed the rarely discussed topic of probabilistic calibration for neural TPP models and found that the mark distributions are often poorly calibrated despite time distributions being well calibrated. Additionally, while recurrent encoders are better at capturing a process' history, we found that solely encoding a few of the last observed events yielded comparable performance to an embedding of the complete history for most datasets and decoders. Finally, we confirmed the concerns of previous research regarding the commonly used datasets for benchmarking neural TPP models and raised concerns about others. We believe our findings will bring valuable insights to the neural TPP research community, and we hope will inspire future work.

## Acknowledgement

We would like to thank Victor Dheur and Sukanya Patra from the Big Data & Machine Learning lab at UMONS for providing helpful critical feedback in earlier versions of this paper. This work is supported by the ARC-21/25 UMONS3 Concerted Research Action funded by the Ministry of the French Community - General Directorate for Non-compulsory Education and Scientific Research.

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

# A   Results on unmarked datasets.

We report the comparison of different event encoding mechanisms and history encoders for unmarked datasets in Table 8 and Table 9, respectively, where we also added the worst score of each component's variation. The results of the combinations that performed best with respect to the NLL-T can be found in Table 10. Overall, our findings regarding unmarked datasets align with the ones described in Section 6. However, we note that most decoders show a lower PCE, and hence, improved calibration with respect to the distribution of inter-arrival time compared to marked datasets. The reliability diagrams displayed in Figure 9 for unmarked datasets do indeed confirm this observation.

In Figure 10, we present the CD diagrams illustrating the relationships between all combinations of Table 10 for the NLL-T and PCE metrics. While there is a noticeable distinction in the average ranks among the decoders, the available data for marked datasets do not provide sufficient evidence to establish statistical differences between them at the $\alpha = 0.1$ significance level.

Table 8: Average, median, and worst scores, as well as average ranks per decoder and variation of event encoding, for unmarked datasets. Refer to Section 5.5 for details on the aggregation procedure. Best scores are highlighted in bold.

| | | | | | | | | |
|---|---|---|---|---|---|---|---|---|
| | | Unmarked Datasets | | | | | | |
| | | NLL-T | | | | PCE | | |
| | Mean | Median | Worst | Rank | Mean | Median | Worst | Rank |
| EC-TO | 0.25 | 0.24 | 0.6 | 3.5 | 0.07 | 0.06 | 0.16 | 3.25 |
| EC-LTO | 0.25 | 0.26 | 0.64 | 3.5 | 0.07 | 0.06 | 0.16 | 3.62 |
| EC-TEM | **-0.21** | **-0.26** | **0.31** | **1.0** | **0.05** | **0.03** | **0.14** | **1.25** |
| EC-LE | -0.09 | -0.04 | 0.46 | 2.0 | 0.05 | 0.03 | 0.15 | 1.88 |
| LNM-TO | -1.15 | -0.5 | -0.26 | 3.0 | **0.01** | **0.01** | **0.01** | 3.12 |
| LNM-LTO | -1.17 | -0.57 | -0.25 | 3.38 | 0.01 | 0.01 | 0.01 | 2.25 |
| LNM-TEM | **-1.46** | **-0.79** | **-0.51** | **1.12** | 0.01 | 0.01 | 0.01 | **1.5** |
| LNM-LE | -0.98 | -0.59 | -0.44 | 2.5 | 0.01 | 0.01 | 0.01 | 3.12 |
| FNN-TO | 0.7 | 0.6 | 1.06 | 2.62 | 0.09 | 0.09 | 0.15 | 2.62 |
| FNN-LTO | **-0.19** | **-0.25** | 1.69 | **1.25** | **0.02** | **0.01** | **0.06** | **1.0** |
| FNN-LE | 0.56 | 0.5 | **1.03** | 2.12 | 0.08 | 0.07 | 0.15 | 2.38 |
| MLP/MC-TO | -0.02 | 0.0 | 0.1 | 3.75 | 0.05 | 0.04 | 0.08 | 3.62 |
| MLP/MC-LTO | -0.14 | -0.17 | 0.06 | 2.5 | **0.03** | 0.03 | 0.08 | **1.88** |
| MLP/MC-TEM | -0.23 | -0.21 | **-0.04** | 2.12 | 0.04 | 0.03 | 0.09 | 2.62 |
| MLP/MC-LE | **-0.3** | **-0.33** | -0.04 | **1.62** | 0.03 | **0.02** | **0.07** | 1.88 |
| RMTPP-TO | -0.07 | -0.07 | 0.11 | 3.75 | 0.04 | 0.03 | 0.08 | 3.75 |
| RMTPP-LTO | -0.21 | -0.13 | 0.2 | 2.75 | **0.02** | **0.02** | **0.03** | 2.0 |
| RMTPP-TEM | **-0.37** | **-0.32** | **-0.25** | **1.25** | 0.03 | 0.02 | 0.07 | **1.88** |
| RMTPP-LE | -0.3 | -0.28 | -0.14 | 2.25 | 0.03 | 0.02 | 0.08 | 2.38 |
| SA/CM-TO | 0.71 | 0.24 | 4.19 | 2.38 | 0.05 | 0.03 | 0.14 | 2.62 |
| SA/CM-LTO | 1.08 | 0.23 | 4.93 | 2.12 | 0.05 | 0.03 | 0.14 | 2.25 |
| SA/CM-LE | **-0.22** | **-0.18** | **0.19** | **1.5** | **0.02** | **0.01** | **0.05** | **1.12** |
| SA/MC-TO | 0.78 | 0.66 | 1.17 | 3.5 | 0.09 | 0.08 | 0.16 | 3.38 |
| SA/MC-LTO | 0.68 | 0.69 | 1.09 | 3.5 | 0.09 | 0.1 | 0.15 | 3.38 |
| SA/MC-TEM | -0.42 | -0.39 | **-0.18** | 1.88 | **0.02** | **0.01** | **0.07** | 1.62 |
| SA/MC-LE | **-0.5** | **-0.49** | 0.17 | **1.12** | 0.03 | 0.02 | 0.08 | 1.62 |

Table 9: Average, median, and worst scores, as well as average ranks per decoder and variation of history encoder, for unmarked datasets. Refer to Section 5.5 for details on the aggregation procedure. Best scores are highlighted in bold.

| | NLL-T | | | | PCE | | | |
|---|---|---|---|---|---|---|---|---|
| | Mean | Median | Worst | Rank | Mean | Median | Worst | Rank |
| CONS-EC | 1.03 | 0.87 | 2.14 | 3.0 | 0.1 | 0.09 | 0.2 | 3.0 |
| SA-EC | 0.48 | 0.54 | 0.85 | 2.0 | 0.08 | 0.07 | 0.17 | 2.0 |
| GRU-EC | **-0.38** | **-0.4** | **0.25** | **1.0** | **0.04** | **0.02** | **0.14** | **1.0** |
| CONS-LNM | -0.83 | -0.16 | 0.13 | 2.75 | **0.01** | **0.01** | 0.02 | 2.25 |
| SA-LNM | -0.69 | -0.26 | -0.14 | 2.25 | 0.01 | 0.01 | 0.02 | 2.25 |
| GRU-LNM | **-1.7** | **-0.92** | **-0.61** | **1.0** | 0.01 | 0.01 | **0.01** | **1.5** |
| CONS-FNN | 0.63 | 0.53 | 1.26 | 2.62 | **0.06** | **0.05** | 0.13 | 2.12 |
| SA-FNN | 0.46 | 0.35 | 1.06 | 2.25 | 0.06 | 0.06 | **0.1** | **1.62** |
| GRU-FNN | **0.25** | **0.16** | **0.94** | **1.12** | 0.06 | 0.06 | 0.11 | 2.25 |
| CONS-MLP/MC | 0.46 | 0.5 | 0.74 | 2.88 | 0.06 | 0.06 | 0.11 | 2.88 |
| SA-MLP/MC | 0.15 | 0.17 | 0.33 | 2.12 | 0.05 | 0.04 | 0.09 | 2.0 |
| GRU-MLP/MC | **-0.49** | **-0.47** | **-0.33** | **1.0** | **0.03** | **0.02** | **0.06** | **1.12** |
| CONS-RMTPP | 0.31 | 0.26 | 0.79 | 2.88 | 0.04 | 0.03 | **0.06** | 2.38 |
| SA-RMTPP | 0.04 | 0.01 | 0.29 | 2.12 | 0.04 | 0.03 | 0.07 | 2.38 |
| GRU-RMTPP | **-0.52** | **-0.54** | **-0.33** | **1.0** | **0.02** | **0.01** | 0.06 | **1.25** |
| CONS-SA/CM | 1.03 | 0.68 | 2.87 | 2.88 | 0.05 | 0.04 | 0.11 | 2.62 |
| SA-SA/CM | 0.53 | 0.26 | **2.7** | 1.62 | **0.04** | **0.03** | **0.1** | 1.88 |
| GRU-SA/CM | **0.51** | **0.2** | 3.0 | **1.5** | 0.04 | 0.03 | 0.1 | **1.5** |
| CONS-SA/MC | 0.52 | 0.49 | 0.86 | 2.88 | 0.07 | **0.06** | 0.11 | 2.62 |
| SA-SA/MC | 0.17 | 0.21 | **0.34** | 1.75 | **0.06** | 0.07 | **0.08** | **1.5** |
| GRU-SA/MC | **0.1** | **0.14** | 0.38 | **1.38** | 0.06 | 0.07 | 0.08 | 1.88 |

Table 10: Average, median, worst scores, and average ranks of the best combinations per decoder on the NLL-T across all unmarked datasets. Best results are highlighted in bold.

| | NLL-T | | | | PCE | | | |
|---|---|---|---|---|---|---|---|---|
| | Mean | Median | Worst | Rank | Mean | Median | Worst | Rank |
| GRU-EC-TEM + B | -0.41 | -0.46 | 0.16 | 5.5 | 0.04 | 0.02 | 0.13 | 7.12 |
| GRU-LNM-TEM | **-1.96** | **-0.95** | **-0.62** | **1.25** | **0.01** | **0.01** | **0.01** | **2.12** |
| GRU-LNK1-TEM | -0.25 | -0.25 | 0.41 | 7.0 | 0.03 | 0.03 | 0.07 | 7.75 |
| GRU-FNN-LTO + B | -0.48 | -0.59 | 1.48 | 4.12 | 0.01 | 0.01 | 0.05 | 3.38 |
| GRU-MLP/MC-LTO + B | -0.57 | -0.56 | -0.44 | 4.25 | 0.02 | 0.01 | 0.05 | 5.5 |
| GRU-RMTPP-LTO | -0.6 | -0.55 | -0.28 | 4.25 | 0.01 | 0.01 | 0.03 | 4.12 |
| SA-SA/CM-LE + B | -0.23 | -0.15 | 0.0 | 7.38 | 0.02 | 0.01 | 0.04 | 4.5 |
| GRU-SA/MC-LE + B | -0.56 | -0.55 | 0.4 | 4.12 | 0.03 | 0.02 | 0.1 | 5.25 |
| Hawkes | -0.22 | -0.16 | 0.16 | 7.12 | 0.02 | 0.01 | 0.06 | 5.25 |
| Poisson | 4.23 | 2.54 | 12.29 | 11.0 | 0.25 | 0.26 | 0.47 | 11.0 |
| NH | 2.29 | 1.11 | 8.2 | 10.0 | 0.16 | 0.15 | 0.3 | 10.0 |

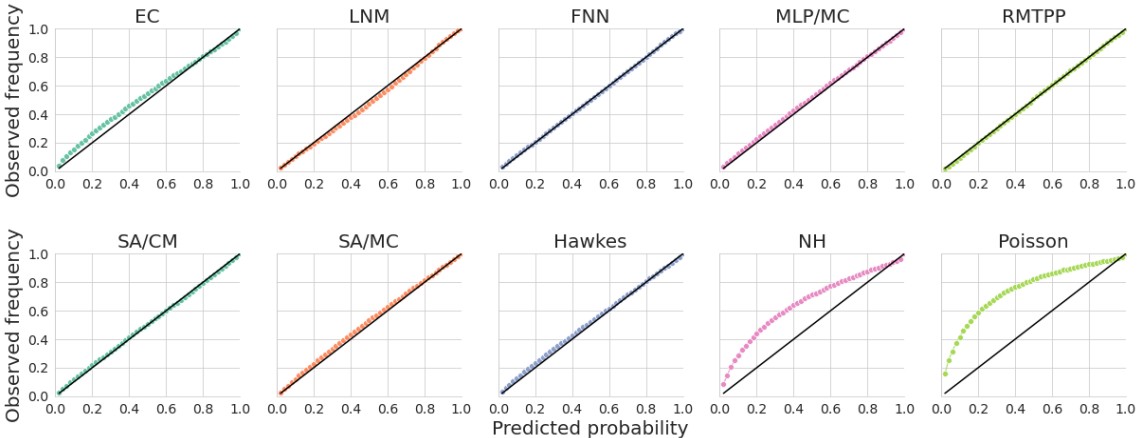

Figure 9: Reliability diagrams of the distribution of inter-arrival times for the decoder's combinations that performed the best on the NLL-T averaged over all unmarked datasets. The bold black line corresponds to perfect marginal calibration.

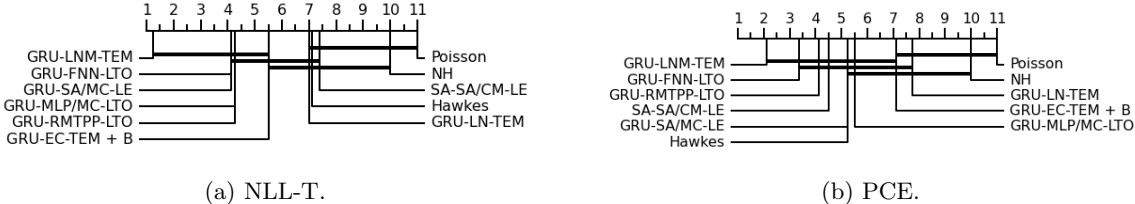

(a) NLL-T.                                               (b) PCE.

Figure 10: Critical Distance (CD) diagrams per metric at the $\alpha = 0.10$ significance level for all decoders on unmarked datasets. The decoders' average ranks are displayed on top, and a bold line joins the decoders' variations that are not statistically different.

## B  Additional results

In Table 11, we report the results with respect to all time-related metrics (NLL-T, PCE) for all models of Table 6 on marked datasets, while results with respect to marked related metrics (NLL-M, ECE, F1-score) are given in Table 12. In turn, results with respect to time-related metrics for models of Table 10 on unmarked datasets are reported in Table 13. Finally, we report in Figure 11 the standardized NLL-T and NLL-M for all models of Table 6 for all marked datasets.

Table 11: Results with respect to the NLL-T and PCE for the combinations of models presented in Table 6 on all marked datasets. Standard error across all splits is reported in parenthesis.

| | NLL-T | | | | | | | |
|---|---|---|---|---|---|---|---|---|
| | LastFM | MOOC | Wikipedia | Github | MIMIC2 | Hawkes | SO | Retweets |
| GRU-LNM-TO | -1363.43 (59.03) | -275.49 (3.53) | -242.45 (30.64) | -378.09 (57.75) | 0.13 (0.33) | -75.62 (1.01) | -79.21 (2.32) | -613.77 (14.25) |
| GRU-LN-TO | -1351.97 (60.26) | -275.49 (3.53) | -226.75 (31.05) | -369.38 (57.05) | 0.66 (0.07) | -73.68 (0.76) | -74.24 (1.42) | -582.47 (2.47) |
| GRU-FNN-LCONCAT | -1355.58 (59.91) | -287.65 (3.56) | -239.87 (29.03) | -371.57 (58.27) | 1.76 (0.11) | -78.33 (0.8) | -87.99 (1.41) | -596.29 (2.65) |
| GRU-SA/CM-LE | -1299.86 (57.52) | -280.73 (3.8) | -200.89 (47.85) | -331.96 (60.72) | 1.23 (0.23) | -78.01 (0.79) | -84.05 (1.3) | -545.31 (13.15) |
| GRU-RMTPP-LCONCAT | -1331.75 (58.87) | -268.91 (3.25) | -267.41 (24.59) | -382.4 (61.05) | 1.66 (0.08) | -78.49 (0.79) | -83.99 (1.47) | -570.73 (2.18) |
| GRU-SA/MC-LE | -1349.8 (60.0) | -277.66 (3.03) | -207.39 (51.79) | -348.82 (56.8) | 0.47 (0.12) | -78.5 (0.76) | -85.17 (1.33) | -581.05 (2.59) |
| GRU-MLP/MC-LCONCAT | -1338.38 (56.91) | -275.43 (3.55) | -186.58 (47.99) | -325.3 (57.09) | 1.82 (0.14) | -78.49 (0.79) | -87.82 (1.15) | -577.8 (1.7) |
| Hawkes | -1189.48 (55.2) | -235.9 (3.09) | 332.83 (93.92) | -308.42 (57.77) | 4.49 (0.24) | -78.66 (0.82) | -83.18 (1.4) | -554.6 (2.15) |
| GRU-EC-LE | -1102.45 (49.95) | -105.23 (3.43) | -153.79 (14.4) | -290.58 (55.37) | 0.91 (0.05) | -78.05 (0.78) | -82.38 (1.34) | -546.95 (2.86) |
| NH | -788.33 (33.0) | -76.64 (1.21) | -102.72 (12.9) | -168.39 (29.39) | 1.73 (0.11) | -72.54 (0.67) | -73.79 (1.06) | -236.29 (4.38) |
| Poisson | -747.35 (46.13) | -51.74 (1.01) | -57.32 (16.36) | -142.58 (26.06) | 2.98 (0.12) | -71.6 (0.7) | -73.6 (1.11) | -234.81 (0.5) |
| GRU-LNM-CONCAT | -1363.78 (59.77) | -289.3 (3.06) | -261.79 (37.08) | -367.41 (56.35) | 0.59 (0.22) | -76.57 (1.18) | -85.12 (3.03) | -621.33 (22.55) |
| GRU-LN-CONCAT | -1348.27 (57.29) | -280.3 (3.15) | -228.31 (31.13) | -367.41 (56.35) | 0.97 (0.05) | -73.68 (0.76) | -81.42 (1.34) | -583.42 (2.36) |
| GRU-SA/CM-LEWL | -1274.49 (50.07) | -268.28 (5.07) | -174.11 (29.42) | -316.81 (51.14) | 1.5 (0.18) | -77.52 (0.65) | -80.23 (1.5) | -555.59 (9.08) |
| GRU-RMTPP-TEMWL + B | -1118.02 (52.48) | -202.08 (9.0) | -136.01 (22.52) | -276.45 (51.41) | 1.34 (0.14) | -78.48 (0.79) | -82.96 (1.36) | -565.46 (3.72) |
| GRU-SA/MC-TEMWL | -1243.21 (47.96) | -257.48 (3.25) | -158.02 (27.61) | -322.72 (53.94) | 1.77 (0.11) | -78.04 (0.73) | -86.24 (1.29) | -569.34 (4.58) |
| GRU-MLP/MC-LEWL | -1192.81 (46.5) | -232.97 (1.31) | -135.52 (20.11) | -285.31 (51.34) | 1.41 (0.11) | -78.59 (0.79) | -85.9 (1.42) | -577.61 (2.41) |
| GRU-EC-TEMWL | -1002.13 (37.43) | -97.11 (1.01) | -144.57 (14.46) | -269.0 (51.2) | 1.82 (0.11) | -78.0 (0.76) | -82.78 (1.42) | -535.75 (1.39) |

| | PCE | | | | | | | |
|---|---|---|---|---|---|---|---|---|
| | LastFM | MOOC | Wikipedia | Github | MIMIC2 | Hawkes | SO | Retweets |
| GRU-LNM-TO | 0.02 (0.0) | 0.04 (0.0) | 0.22 (0.07) | 0.03 (0.01) | 0.05 (0.01) | 0.02 (0.01) | 0.03 (0.01) | 0.01 (0.0) |
| GRU-LN-TO | 0.03 (0.0) | 0.04 (0.0) | 0.28 (0.05) | 0.05 (0.0) | 0.07 (0.01) | 0.03 (0.0) | 0.05 (0.0) | 0.01 (0.0) |
| GRU-FNN-LCONCAT | 0.02 (0.0) | 0.02 (0.0) | 0.07 (0.01) | 0.02 (0.0) | 0.06 (0.0) | 0.0 (0.0) | 0.0 (0.0) | 0.01 (0.0) |
| GRU-SA/CM-LE | 0.05 (0.01) | 0.02 (0.0) | 0.18 (0.05) | 0.09 (0.01) | 0.05 (0.01) | 0.0 (0.0) | 0.0 (0.0) | 0.01 (0.0) |
| GRU-RMTPP-LCONCAT | 0.06 (0.0) | 0.07 (0.0) | 0.1 (0.01) | 0.02 (0.0) | 0.04 (0.0) | 0.01 (0.0) | 0.01 (0.0) | 0.03 (0.0) |
| GRU-SA/MC-LE | 0.06 (0.01) | 0.13 (0.0) | 0.24 (0.04) | 0.12 (0.01) | 0.04 (0.01) | 0.01 (0.0) | 0.01 (0.0) | 0.01 (0.0) |
| GRU-MLP/MC-LCONCAT | 0.08 (0.01) | 0.13 (0.01) | 0.27 (0.03) | 0.14 (0.01) | 0.05 (0.01) | 0.01 (0.0) | 0.01 (0.0) | 0.02 (0.0) |
| Hawkes | 0.21 (0.0) | 0.2 (0.0) | 0.19 (0.01) | 0.15 (0.01) | 0.06 (0.0) | 0.0 (0.0) | 0.01 (0.0) | 0.03 (0.0) |
| GRU-EC-LE | 0.27 (0.0) | 0.37 (0.01) | 0.35 (0.01) | 0.19 (0.01) | 0.08 (0.01) | 0.02 (0.0) | 0.02 (0.0) | 0.09 (0.0) |
| NH | 0.35 (0.01) | 0.4 (0.0) | 0.35 (0.01) | 0.3 (0.02) | 0.07 (0.0) | 0.04 (0.0) | 0.05 (0.0) | 0.36 (0.0) |
| Poisson | 0.38 (0.01) | 0.4 (0.0) | 0.34 (0.01) | 0.33 (0.02) | 0.07 (0.0) | 0.04 (0.0) | 0.04 (0.0) | 0.36 (0.0) |
| GRU-LNM-CONCAT | 0.02 (0.01) | 0.03 (0.01) | 0.16 (0.06) | 0.05 (0.01) | 0.03 (0.01) | 0.01 (0.01) | 0.03 (0.01) | 0.01 (0.0) |
| GRU-LN-CONCAT | 0.04 (0.01) | 0.04 (0.0) | 0.28 (0.06) | 0.05 (0.01) | 0.05 (0.01) | 0.03 (0.0) | 0.04 (0.0) | 0.01 (0.0) |
| GRU-SA/CM-LEWL | 0.09 (0.01) | 0.09 (0.02) | 0.28 (0.02) | 0.11 (0.01) | 0.05 (0.0) | 0.01 (0.0) | 0.02 (0.01) | 0.01 (0.0) |
| GRU-RMTPP-TEMWL + B | 0.25 (0.0) | 0.27 (0.01) | 0.32 (0.01) | 0.18 (0.01) | 0.03 (0.0) | 0.01 (0.0) | 0.01 (0.0) | 0.04 (0.0) |
| GRU-SA/MC-TEMWL | 0.17 (0.01) | 0.19 (0.01) | 0.33 (0.01) | 0.13 (0.01) | 0.05 (0.0) | 0.01 (0.0) | 0.01 (0.0) | 0.03 (0.0) |
| GRU-MLP/MC-LEWL | 0.21 (0.01) | 0.24 (0.01) | 0.33 (0.01) | 0.19 (0.01) | 0.03 (0.0) | 0.01 (0.0) | 0.01 (0.0) | 0.02 (0.0) |
| GRU-EC-TEMWL | 0.29 (0.01) | 0.39 (0.0) | 0.35 (0.01) | 0.19 (0.01) | 0.06 (0.0) | 0.02 (0.0) | 0.02 (0.0) | 0.12 (0.01) |

Table 12: Results with respect to the NLL-M, ECE and F1-score for the combinations of models presented in Table 6 on all marked datasets. Standard error across all splits is reported in parenthesis.

| | LastFM | MOOC | Wikipedia | Github | MIMIC2 | Hawkes | SO | Retweets |
|---|---|---|---|---|---|---|---|---|
| | | | | NLL-M | | | | |
| GRU-LNM-CONCAT | 762.03 (22.43) | 92.47 (1.85) | 259.12 (21.41) | 124.34 (18.24) | 2.51 (0.14) | 110.15 (0.6) | 107.09 (0.6) | 85.16 (0.83) |
| GRU-FNN-LCONCAT | 791.0 (31.28) | 103.16 (4.6) | 346.6 (47.58) | 128.11 (19.55) | 4.34 (0.15) | 110.72 (0.61) | 109.83 (0.87) | 83.38 (0.23) |
| GRU-SA/CM-LCONCAT | 864.68 (29.32) | 110.49 (1.87) | 292.59 (29.1) | 152.63 (21.43) | 4.37 (0.15) | 111.6 (0.54) | 115.49 (0.69) | 90.19 (0.17) |
| GRU-RMTPP-LCONCAT | 834.05 (40.33) | 89.92 (1.67) | 327.16 (77.34) | 122.9 (16.42) | 2.29 (0.23) | 110.34 (0.56) | 107.33 (0.34) | 82.63 (0.18) |
| GRU-SA/MC-LE | 869.85 (30.36) | 177.49 (2.28) | 407.99 (74.55) | 156.81 (22.86) | 4.55 (0.17) | 111.71 (0.54) | 117.3 (0.91) | 88.5 (0.21) |
| GRU-MLP/MC-LTO | 875.01 (31.01) | 163.62 (4.1) | 476.0 (136.36) | 166.51 (23.07) | 4.61 (0.2) | 111.67 (0.54) | 118.33 (0.73) | 90.93 (2.61) |
| Hawkes | 514.13 (16.19) | 112.14 (1.26) | 144.79 (11.89) | 122.44 (15.5) | 12.84 (0.23) | 110.83 (0.54) | 114.99 (0.75) | 89.5 (0.14) |
| GRU-EC-TO | 870.76 (29.88) | 152.98 (1.71) | 289.66 (41.0) | 154.62 (21.3) | 4.4 (0.16) | 111.66 (0.55) | 120.25 (0.69) | 89.96 (0.25) |
| NH | 869.55 (30.47) | 184.55 (2.35) | 272.66 (21.64) | 161.84 (21.59) | 4.66 (0.11) | 111.99 (0.54) | 131.2 (0.74) | 91.1 (0.15) |
| Poisson | 873.82 (30.24) | 188.13 (2.37) | 534.95 (131.14) | 168.37 (21.04) | 11.25 (0.46) | 113.46 (0.54) | 132.93 (0.74) | 92.11 (0.15) |
| | | | | | | | | |
| Hawkes | 514.13 (16.19) | 112.14 (1.26) | 144.79 (11.89) | 122.44 (15.5) | 12.84 (0.23) | 110.83 (0.54) | 114.99 (0.75) | 89.5 (0.14) |
| GRU-RMTPP-LCONCAT + B | 722.89 (18.13) | 86.32 (0.95) | 461.49 (117.39) | 123.67 (16.18) | 3.75 (0.25) | 110.31 (0.58) | 108.45 (0.99) | 82.74 (0.21) |
| GRU-LNM-CONCAT | 762.03 (22.43) | 92.47 (1.85) | 259.12 (21.41) | 124.34 (18.24) | 2.51 (0.14) | 110.15 (0.6) | 107.09 (0.6) | 85.16 (0.83) |
| GRU-MLP/MC-LCONCAT | 800.68 (22.15) | 117.88 (5.34) | 520.5 (162.31) | 147.26 (21.75) | 3.69 (0.34) | 110.85 (0.49) | 112.62 (0.9) | 83.37 (0.3) |
| GRU-FNN-LCONCAT | 791.0 (31.28) | 103.16 (4.6) | 346.6 (47.58) | 128.11 (19.55) | 4.34 (0.15) | 110.72 (0.61) | 109.83 (0.87) | 83.38 (0.23) |
| GRU-EC-TEMWL | 830.42 (33.2) | 98.53 (1.65) | 298.61 (38.42) | 150.4 (21.71) | 3.06 (0.18) | 110.06 (0.53) | 108.8 (0.93) | 85.55 (0.9) |
| SA-SA/MC-CONCAT | 862.04 (28.39) | 130.75 (2.06) | 281.81 (34.96) | 148.24 (23.17) | 3.14 (0.31) | 111.12 (0.5) | 111.44 (0.6) | 87.58 (0.46) |
| SA-SA/CM-LCONCAT | 860.87 (30.33) | 111.51 (3.0) | 321.69 (41.29) | 152.42 (20.93) | 4.43 (0.16) | 111.6 (0.51) | 116.43 (0.44) | 89.67 (0.34) |

| | LastFM | MOOC | Wikipedia | Github | MIMIC2 | Hawkes | SO | Retweets |
|---|---|---|---|---|---|---|---|---|
| | | | | ECE | | | | |
| GRU-LNM-TO | 0.5 (0.0) | 0.45 (0.01) | 0.49 (0.0) | 0.4 (0.01) | 0.41 (0.02) | 0.44 (0.0) | 0.28 (0.02) | 0.28 (0.03) |
| GRU-LN-TO | 0.5 (0.0) | 0.45 (0.01) | 0.5 (0.0) | 0.42 (0.01) | 0.4 (0.02) | 0.44 (0.0) | 0.26 (0.01) | 0.23 (0.02) |
| GRU-FNN-LCONCAT | 0.43 (0.05) | 0.31 (0.04) | 0.5 (0.01) | 0.15 (0.0) | 0.46 (0.0) | 0.39 (0.01) | 0.03 (0.0) | 0.06 (0.0) |
| GRU-SA/CM-LE | 0.5 (0.0) | 0.45 (0.0) | 0.5 (0.0) | 0.43 (0.01) | 0.46 (0.0) | 0.44 (0.0) | 0.26 (0.04) | 0.41 (0.0) |
| GRU-RMTPP-LCONCAT | 0.43 (0.03) | 0.06 (0.01) | 0.36 (0.06) | 0.15 (0.01) | 0.16 (0.02) | 0.4 (0.0) | 0.15 (0.02) | 0.08 (0.02) |
| GRU-SA/MC-LE | 0.5 (0.0) | 0.42 (0.0) | 0.49 (0.0) | 0.22 (0.02) | 0.33 (0.01) | 0.44 (0.0) | 0.15 (0.01) | 0.27 (0.02) |
| GRU-MLP/MC-LCONCAT | 0.42 (0.02) | 0.21 (0.04) | 0.47 (0.02) | 0.14 (0.03) | 0.32 (0.03) | 0.44 (0.01) | 0.03 (0.01) | 0.06 (0.0) |
| Hawkes | 0.03 (0.0) | 0.14 (0.0) | 0.11 (0.03) | 0.07 (0.02) | 0.19 (0.01) | 0.34 (0.01) | 0.09 (0.02) | 0.19 (0.0) |
| GRU-EC-LE | 0.5 (0.0) | 0.44 (0.01) | 0.48 (0.01) | 0.31 (0.01) | 0.38 (0.0) | 0.44 (0.0) | 0.23 (0.02) | 0.38 (0.01) |
| NH | 0.5 (0.0) | 0.5 (0.0) | 0.5 (0.0) | 0.47 (0.01) | 0.46 (0.0) | 0.48 (0.0) | 0.46 (0.0) | 0.45 (0.01) |
| Poisson | 0.5 (0.0) | 0.5 (0.0) | 0.5 (0.0) | 0.49 (0.01) | 0.47 (0.0) | 0.48 (0.0) | 0.46 (0.0) | 0.46 (0.0) |
| | | | | | | | | |
| GRU-LNM-CONCAT | 0.35 (0.03) | 0.18 (0.02) | 0.36 (0.07) | 0.31 (0.04) | 0.15 (0.01) | 0.4 (0.0) | 0.12 (0.02) | 0.08 (0.01) |
| GRU-LN-CONCAT | 0.29 (0.05) | 0.13 (0.02) | 0.46 (0.03) | 0.31 (0.04) | 0.2 (0.05) | 0.39 (0.01) | 0.13 (0.03) | 0.07 (0.01) |
| GRU-SA/CM-LEWL | 0.5 (0.0) | 0.42 (0.02) | 0.5 (0.0) | 0.43 (0.01) | 0.39 (0.03) | 0.44 (0.0) | 0.34 (0.03) | 0.23 (0.04) |
| GRU-RMTPP-TEMWL + B | 0.31 (0.02) | 0.06 (0.01) | 0.34 (0.08) | 0.19 (0.02) | 0.1 (0.01) | 0.4 (0.0) | 0.13 (0.02) | 0.07 (0.01) |
| GRU-SA/MC-TEMWL | 0.49 (0.01) | 0.36 (0.02) | 0.35 (0.06) | 0.25 (0.01) | 0.27 (0.02) | 0.44 (0.0) | 0.08 (0.04) | 0.26 (0.03) |
| GRU-MLP/MC-LEWL | 0.4 (0.05) | 0.23 (0.02) | 0.44 (0.05) | 0.22 (0.02) | 0.18 (0.03) | 0.44 (0.0) | 0.03 (0.01) | 0.17 (0.04) |
| GRU-EC-TEMWL | 0.48 (0.01) | 0.21 (0.03) | 0.49 (0.0) | 0.27 (0.04) | 0.17 (0.02) | 0.4 (0.0) | 0.21 (0.01) | 0.12 (0.02) |

| | LastFM | MOOC | Wikipedia | Github | MIMIC2 | Hawkes | SO | Retweets |
|---|---|---|---|---|---|---|---|---|
| | | | | F1-score | | | | |
| GRU-LNM-TO | 0.01 (0.0) | 0.06 (0.01) | 0.01 (0.0) | 0.4 (0.03) | 0.22 (0.02) | 0.14 (0.01) | 0.28 (0.01) | 0.54 (0.01) |
| GRU-LN-TO | 0.01 (0.0) | 0.06 (0.01) | 0.01 (0.0) | 0.4 (0.03) | 0.23 (0.02) | 0.15 (0.01) | 0.28 (0.0) | 0.55 (0.0) |
| GRU-FNN-LCONCAT | 0.02 (0.01) | 0.23 (0.04) | 0.02 (0.01) | 0.49 (0.02) | 0.22 (0.02) | 0.23 (0.0) | 0.33 (0.0) | 0.6 (0.0) |
| GRU-SA/CM-LE | 0.0 (0.0) | 0.04 (0.0) | 0.0 (0.0) | 0.4 (0.03) | 0.22 (0.02) | 0.13 (0.01) | 0.28 (0.0) | 0.45 (0.04) |
| GRU-RMTPP-LCONCAT | 0.03 (0.01) | 0.38 (0.01) | 0.31 (0.11) | 0.52 (0.01) | 0.74 (0.01) | 0.25 (0.01) | 0.33 (0.0) | 0.6 (0.0) |
| GRU-SA/MC-LE | 0.0 (0.0) | 0.03 (0.0) | 0.01 (0.0) | 0.4 (0.03) | 0.24 (0.03) | 0.16 (0.01) | 0.31 (0.0) | 0.55 (0.0) |
| GRU-MLP/MC-LCONCAT | 0.02 (0.0) | 0.24 (0.03) | 0.03 (0.02) | 0.4 (0.03) | 0.43 (0.09) | 0.2 (0.01) | 0.32 (0.0) | 0.6 (0.0) |
| Hawkes | 0.3 (0.0) | 0.29 (0.0) | 0.66 (0.02) | 0.54 (0.01) | 0.63 (0.0) | 0.28 (0.0) | 0.32 (0.0) | 0.56 (0.0) |
| GRU-EC-LE | 0.0 (0.0) | 0.06 (0.0) | 0.02 (0.0) | 0.4 (0.03) | 0.22 (0.02) | 0.14 (0.01) | 0.29 (0.0) | 0.53 (0.0) |
| NH | 0.0 (0.0) | 0.01 (0.0) | 0.0 (0.0) | 0.4 (0.03) | 0.22 (0.02) | 0.11 (0.0) | 0.26 (0.0) | 0.33 (0.0) |
| Poisson | 0.0 (0.0) | 0.01 (0.0) | 0.01 (0.0) | 0.4 (0.03) | 0.2 (0.02) | 0.11 (0.0) | 0.26 (0.0) | 0.33 (0.0) |
| | | | | | | | | |
| GRU-LNM-CONCAT | 0.1 (0.03) | 0.36 (0.01) | 0.22 (0.1) | 0.47 (0.03) | 0.64 (0.01) | 0.26 (0.0) | 0.33 (0.0) | 0.59 (0.01) |
| GRU-LN-CONCAT | 0.1 (0.02) | 0.36 (0.01) | 0.02 (0.01) | 0.47 (0.03) | 0.53 (0.09) | 0.26 (0.0) | 0.32 (0.0) | 0.6 (0.0) |
| GRU-SA/CM-LEWL | 0.0 (0.0) | 0.05 (0.01) | 0.0 (0.0) | 0.4 (0.03) | 0.25 (0.04) | 0.14 (0.0) | 0.28 (0.01) | 0.56 (0.01) |
| GRU-RMTPP-TEMWL + B | 0.11 (0.01) | 0.39 (0.01) | 0.46 (0.09) | 0.48 (0.02) | 0.73 (0.03) | 0.27 (0.0) | 0.32 (0.0) | 0.59 (0.0) |
| GRU-SA/MC-TEMWL | 0.01 (0.0) | 0.08 (0.02) | 0.1 (0.04) | 0.4 (0.03) | 0.57 (0.02) | 0.19 (0.02) | 0.32 (0.0) | 0.56 (0.0) |
| GRU-MLP/MC-LEWL | 0.02 (0.01) | 0.25 (0.01) | 0.07 (0.04) | 0.42 (0.02) | 0.59 (0.02) | 0.18 (0.01) | 0.33 (0.0) | 0.57 (0.01) |
| GRU-EC-TEMWL | 0.02 (0.0) | 0.28 (0.01) | 0.01 (0.0) | 0.41 (0.03) | 0.53 (0.03) | 0.27 (0.0) | 0.32 (0.0) | 0.58 (0.01) |

Table 13: Results with respect to the NLL-T and PCE for the combinations of models presented in Table 10 on all unmarked datasets. Standard error across all splits is reported in parenthesis.

| | NLL-T | | | | | | | |
|---|---|---|---|---|---|---|---|---|
| | Taxi | Twitter | Reddit Subs | Reddit Ask | PUBG | Yelp T. | Yelp A. | Yelp M. |
| GRU-EC-TEM + B | -136.78 (3.91) | -6.21 (0.46) | -4403.97 (24.0) | -917.48 (13.1) | -106.99 (0.21) | -2803.23 (85.58) | -5.76 (0.18) | -56.18 (0.76) |
| GRU-LNM-TEM | -137.02 (3.87) | -11.89 (0.62) | -4496.27 (37.01) | -926.36 (12.67) | -194.39 (18.07) | -2957.96 (92.81) | -6.8 (0.19) | -59.61 (0.9) |
| GRU-FNN-LTO | -132.51 (4.01) | -11.66 (0.53) | -4424.06 (24.21) | -926.25 (12.87) | -121.88 (0.18) | -2486.32 (80.2) | -5.72 (0.16) | -58.52 (0.78) |
| GRU-MLP/MC-LTO | -136.25 (3.95) | -10.4 (0.32) | -4403.41 (24.25) | -921.58 (12.77) | -110.44 (1.21) | -2808.32 (84.78) | -5.75 (0.26) | -58.21 (0.79) |
| GRU-RMTPP-LTO | -136.15 (3.98) | -9.96 (0.53) | -4407.58 (24.0) | -919.08 (12.91) | -107.04 (0.2) | -2906.91 (87.45) | -5.85 (0.14) | -58.31 (0.79) |
| SA-SA/CM-LE | -134.2 (4.13) | -10.54 (0.51) | -4320.9 (23.17) | -856.8 (12.31) | -104.81 (1.31) | -2739.73 (81.7) | -5.29 (0.15) | -55.49 (0.88) |
| GRU-SA/MC-LE | -124.53 (5.1) | -12.11 (0.5) | -4387.88 (19.64) | -922.39 (13.07) | -109.3 (0.9) | -2901.79 (92.0) | -6.83 (0.23) | -57.35 (0.84) |
| Hawkes | -134.09 (3.93) | -7.62 (0.51) | -4395.2 (23.74) | -919.51 (12.92) | -102.69 (0.2) | -2773.63 (87.1) | -4.62 (0.18) | -53.33 (0.8) |
| NH | -48.21 (24.44) | -1.94 (0.22) | -4158.41 (58.74) | -714.92 (11.7) | -94.95 (0.14) | -2084.84 (291.26) | -3.23 (0.14) | -39.19 (0.6) |
| Poisson | -41.71 (1.02) | 5.18 (0.11) | -3706.3 (17.61) | -676.53 (10.2) | -93.11 (0.15) | -673.76 (17.22) | -2.11 (0.1) | -25.2 (0.29) |

| | PCE | | | | | | | |
|---|---|---|---|---|---|---|---|---|
| | Taxi | Twitter | Reddit Subs | Reddit Ask | PUBG | Yelp T. | Yelp A. | Yelp M. |
| GRU-EC-TEM + B | 0.01 (0.0) | 0.13 (0.0) | 0.01 (0.0) | 0.02 (0.0) | 0.02 (0.0) | 0.05 (0.0) | 0.02 (0.0) | 0.05 (0.0) |
| GRU-LNM-TEM | 0.01 (0.0) | 0.01 (0.0) | 0.01 (0.0) | 0.0 (0.0) | 0.01 (0.0) | 0.01 (0.0) | 0.01 (0.0) | 0.01 (0.0) |
| GRU-FNN-LTO | 0.02 (0.0) | 0.01 (0.0) | 0.0 (0.0) | 0.0 (0.0) | 0.0 (0.0) | 0.05 (0.0) | 0.01 (0.0) | 0.01 (0.0) |
| GRU-MLP/MC-LTO | 0.01 (0.0) | 0.03 (0.01) | 0.01 (0.0) | 0.01 (0.0) | 0.01 (0.0) | 0.05 (0.0) | 0.01 (0.0) | 0.02 (0.0) |
| GRU-RMTPP-LTO | 0.01 (0.0) | 0.03 (0.0) | 0.01 (0.0) | 0.01 (0.0) | 0.01 (0.0) | 0.03 (0.0) | 0.01 (0.0) | 0.01 (0.0) |
| SA-SA/CM-LE | 0.01 (0.0) | 0.01 (0.0) | 0.01 (0.0) | 0.01 (0.0) | 0.01 (0.0) | 0.04 (0.0) | 0.01 (0.0) | 0.02 (0.0) |
| GRU-SA/MC-LE | 0.1 (0.03) | 0.02 (0.0) | 0.01 (0.0) | 0.01 (0.0) | 0.01 (0.0) | 0.03 (0.0) | 0.01 (0.0) | 0.02 (0.0) |
| Hawkes | 0.01 (0.0) | 0.01 (0.0) | 0.0 (0.0) | 0.01 (0.0) | 0.01 (0.0) | 0.06 (0.0) | 0.02 (0.0) | 0.05 (0.0) |
| NH | 0.3 (0.06) | 0.17 (0.0) | 0.1 (0.02) | 0.17 (0.0) | 0.06 (0.0) | 0.28 (0.06) | 0.04 (0.0) | 0.13 (0.0) |
| Poisson | 0.35 (0.0) | 0.2 (0.0) | 0.27 (0.0) | 0.25 (0.0) | 0.07 (0.0) | 0.47 (0.0) | 0.12 (0.0) | 0.27 (0.0) |

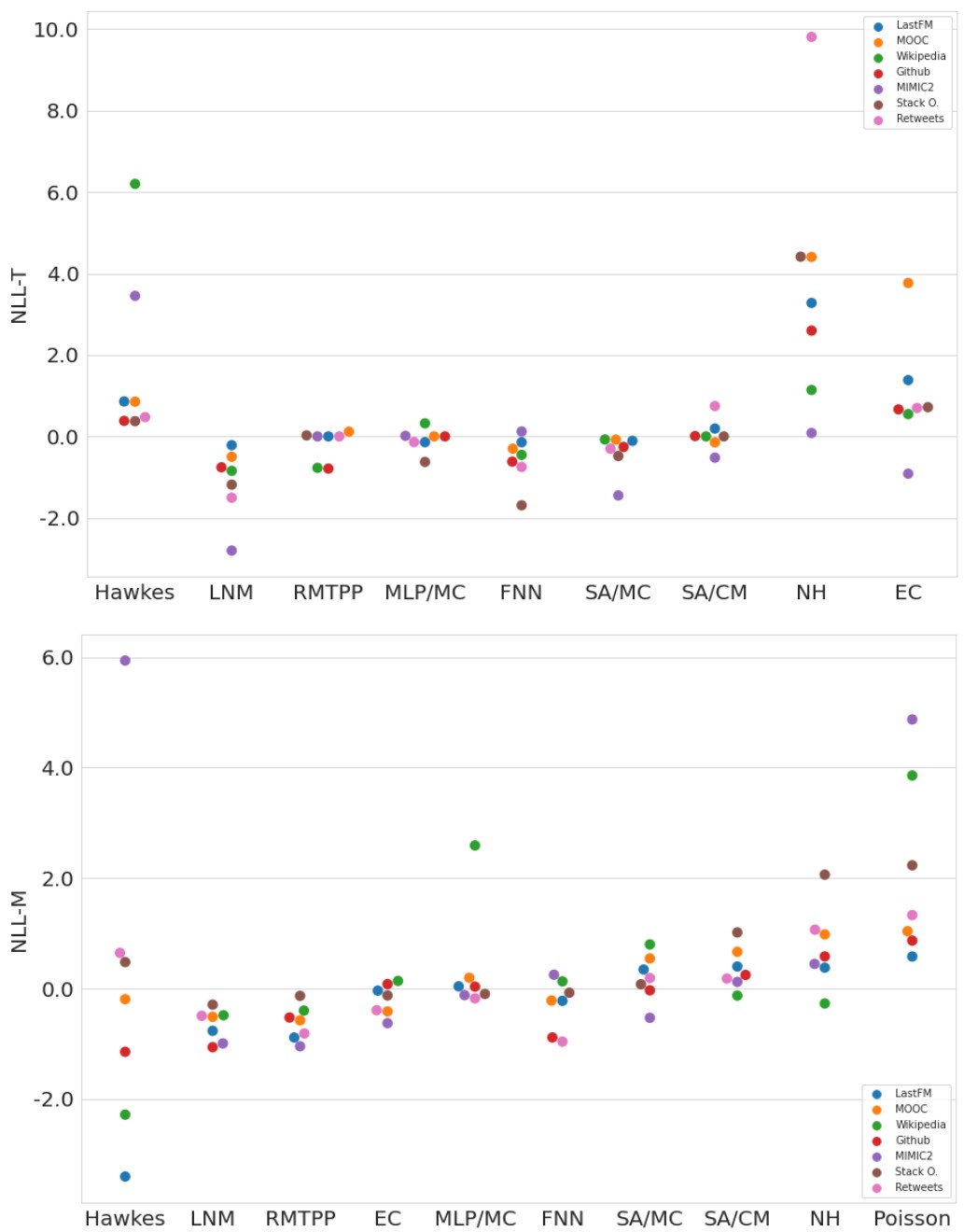

Figure 11: Standardized NLL-T and NLL-M for various decoders per dataset. The NLL-T was computed using the top row models of Table 6, while the NLL-M was computed using the bottom row models of Table 6. Lower NLL-T and NLL-M is better.

## C  TPP models considered

Table 14: Combinations included in the experimental study. "Unmarked" refers to whether the method is adapted for unmarked datasets. For EC, LNM and RMTPP, the setting where a baseline intensity term (B) is also considered.

| Decoder | Encoder | Event encoding | Name | Unmarked |
|---|---|---|---|---|
| EC | CONS | \ | CONS-EC | ✓ |
| | GRU | TO | GRU-EC-TO | ✓ |
| | | LTO | GRU-EC-LTO | ✓ |
| | | CONCAT | GRU-EC-CONCAT | ✗ |
| | | LCONCAT | GRU-EC-LCONCAT | ✗ |
| | | TEM | GRU-EC-TEM | ✓ |
| | | TEMWL | GRU-EC-TEMWL | ✗ |
| | | LE | GRU-EC-LE | ✓ |
| | | LEWL | GRU-EC-LEWL | ✗ |
| | SA | TO | SA-EC-TO | ✓ |
| | | LTO | SA-EC-LTO | ✓ |
| | | CONCAT | SA-EC-CONCAT | ✗ |
| | | LCONCAT | SA-EC-LCONCAT | ✗ |
| | | TEM | SA-EC-TEM | ✓ |
| | | TEMWL | SA-EC-TEMWL | ✗ |
| | | LE | SA-EC-LE | ✓ |
| | | LEWL | SA-EC-LEWL | ✗ |
| LNM | CONS | \ | CONS-LNM | ✓ |
| | GRU | TO | GRU-LNM-TO | ✓ |
| | | LTO | GRU-LNM-LTO | ✓ |
| | | CONCAT | GRU-LNM-CONCAT | ✗ |
| | | LCONCAT | GRU-LNM-LCONCAT | ✗ |
| | | TEM | GRU-LNM-TEM | ✓ |
| | | TEMWL | GRU-LNM-TEMWL | ✗ |
| | | LE | GRU-LNM-LE | ✓ |
| | | LEWL | GRU-LNM-LEWL | ✗ |
| | SA | TO | SA-LNM-TO | ✓ |
| | | LTO | SA-LNM-LTO | ✓ |
| | | CONCAT | SA-LNM-CONCAT | ✗ |
| | | LCONCAT | SA-LNM-LCONCAT | ✗ |
| | | TEM | SA-LNM-TEM | ✓ |
| | | TEMWL | SA-LNM-TEMWL | ✗ |
| | | LE | SA-LNM-LE | ✓ |
| | | LEWL | SA-LNM-LEWL | ✗ |
| RMTPP | CONS | \ | CONS-RMTPP | ✓ |
| | GRU | TO | GRU-RMTPP-TO | ✓ |
| | | LTO | GRU-RMTPP-LTO | ✓ |
| | | CONCAT | GRU-RMTPP-CONCAT | ✗ |
| | | LCONCAT | GRU-RMTPP-LCONCAT | ✗ |
| | | TEM | GRU-RMTPP-TEM | ✓ |
| | | TEMWL | GRU-RMTPP-TEMWL | ✗ |
| | | LE | GRU-RMTPP-LE | ✓ |
| | | LEWL | GRU-RMTPP-LEWL | ✗ |
| | SA | TO | SA-RMTPP-TO | ✓ |
| | | LTO | SA-RMTPP-LTO | ✓ |
| | | CONCAT | SA-RMTPP-CONCAT | ✗ |
| | | LCONCAT | SA-RMTPP-LCONCAT | ✗ |
| | | TEM | SA-RMTPP-TEM | ✓ |
| | | TEMWL | SA-RMTPP-TEMWL | ✗ |
| | | LE | SA-RMTPP-LE | ✓ |
| | | LEWL | SA-RMTPP-LEWL | ✗ |
| FNN | CONS | TO | CONS-FNN-TO | ✓ |
| | | LTO | CONS-FNN-LTO | ✓ |
| | | CONCAT | CONS-FNN-CONCAT | ✗ |
| | | LCONCAT | CONS-FNN-LCONCAT | ✗ |
| | | LE | CONS-FNN-LE | ✓ |
| | | LEWL | CONS-FNN-LEWL | ✗ |
| | GRU | TO | GRU-FNN-TO | ✓ |
| | | LTO | GRU-FNN-LTO | ✓ |
| | | CONCAT | GRU-FNN-CONCAT | ✗ |
| | | LCONCAT | GRU-FNN-LCONCAT | ✗ |
| | | LE | GRU-FNN-LE | ✓ |
| | | LEWL | GRU-FNN-LEWL | ✗ |
| | SA | TO | SA-FNN-TO | ✓ |
| | | LTO | SA-FNN-LTO | ✓ |
| | | CONCAT | SA-FNN-CONCAT | ✗ |
| | | LCONCAT | SA-FNN-LCONCAT | ✗ |
| | | LE | SA-FNN-LE | ✓ |
| | | LEWL | SA-FNN-LEWL | ✗ |

| Decoder | Encoder | Event encoding | Name | Unmarked |
|---|---|---|---|---|
| MLP/MC | CONS | TO | CONS-MLP/MC-TO | ✓ |
| | | LTO | CONS-MLP/MC-LTO | ✓ |
| | | CONCAT | CONS-MLP/MC-CONCAT | ✗ |
| | | LCONCAT | CONS-MLP/MC-LCONCAT | ✗ |
| | | TEM | CONS-MLP/MC-TEM | ✗ |
| | | TEMWL | CONS-MLP/MC-TEMWL | ✗ |
| | | LE | CONS-MLP/MC-LE | ✓ |
| | | LEWL | CONS-MLP/MC-LEWL | ✗ |
| | GRU | TO | GRU-MLP/MC-TO | ✓ |
| | | LTO | GRU-MLP/MC-LTO | ✓ |
| | | CONCAT | GRU-MLP/MC-CONCAT | ✗ |
| | | LCONCAT | GRU-MLP/MC-LCONCAT | ✗ |
| | | TEM | GRU-MLP/MC-TEM | ✓ |
| | | TEMWL | GRU-MLP/MC-TEMWL | ✗ |
| | | LE | GRU-MLP/MC-LE | ✓ |
| | | LEWL | GRU-MLP/MC-LEWL | ✗ |
| | SA | TO | SA-MLP/MC-TO | ✓ |
| | | LTO | SA-MLP/MC-LTO | ✓ |
| | | CONCAT | SA-MLP/MC-CONCAT | ✗ |
| | | LCONCAT | SA-MLP/MC-LCONCAT | ✗ |
| | | TEM | SA-MLP/MC-TEM | ✓ |
| | | TEMWL | SA-MLP/MC-TEMWL | ✗ |
| | | LE | SA-MLP/MC-LE | ✓ |
| | | LEWL | SA-MLP/MC-LEWL | ✗ |
| SA/CM | CONS | TO | CONS-SA/CM-TO | ✓ |
| | | LTO | CONS-SA/CM-LTO | ✓ |
| | | CONCAT | CONS-SA/CM-CONCAT | ✗ |
| | | LCONCAT | CONS-SA/CM-LCONCAT | ✗ |
| | | LE | CONS-SA/CM-LE | ✓ |
| | | LEWL | CONS-SA/CM-LEWL | ✗ |
| | GRU | TO | GRU-SA/CM-TO | ✓ |
| | | LTO | GRU-SA/CM-LTO | ✓ |
| | | CONCAT | GRU-SA/CM-CONCAT | ✗ |
| | | LCONCAT | GRU-SA/CM-LCONCAT | ✗ |
| | | LE | GRU-SA/CM-LE | ✓ |
| | | LEWL | GRU-SA/CM-LEWL | ✗ |
| | SA | TO | SA-SA/CM-TO | ✓ |
| | | LTO | SA-SA/CM-LTO | ✓ |
| | | CONCAT | SA-SA/CM-CONCAT | ✗ |
| | | LCONCAT | SA-SA/CM-LCONCAT | ✗ |
| | | LE | SA-SA/CM-LE | ✓ |
| | | LEWL | SA-SA/CM-LEWL | ✗ |
| SA/MC | CONS | TO | CONS-SA/MC-TO | ✓ |
| | | LTO | CONS-SA/MC-LTO | ✓ |
| | | CONCAT | CONS-SA/MC-CONCAT | ✗ |
| | | LCONCAT | CONS-SA/MC-LCONCAT | ✗ |
| | | TEM | CONS-SA/MC-TEM | ✗ |
| | | TEMWL | CONS-SA/MC-TEMWL | ✗ |
| | | LE | CONS-SA/MC-LE | ✓ |
| | | LEWL | CONS-SA/MC-LEWL | ✗ |
| | GRU | TO | GRU-SA/MC-TO | ✓ |
| | | LTO | GRU-SA/MC-LTO | ✓ |
| | | CONCAT | GRU-SA/MC-CONCAT | ✗ |
| | | LCONCAT | GRU-SA/MC-LCONCAT | ✗ |
| | | TEM | GRU-SA/MC-TEM | ✓ |
| | | TEMWL | GRU-SA/MC-TEMWL | ✗ |
| | | LE | GRU-SA/MC-LE | ✓ |
| | | LEWL | GRU-SA/MC-LEWL | ✗ |
| | SA | TO | SA-SA/MC-TO | ✓ |
| | | LTO | SA-SA/MC-LTO | ✓ |
| | | CONCAT | SA-SA/MC-CONCAT | ✗ |
| | | LCONCAT | SA-SA/MC-LCONCAT | ✗ |
| | | TEM | SA-SA/MC-TEM | ✓ |
| | | TEMWL | SA-SA/MC-TEMWL | ✗ |
| | | LE | SA-SA/MC-LE | ✓ |
| | | LEWL | SA-SA/MC-LEWL | ✗ |
| Neural Hawkes | \ | \ | NH | ✓ |
| Hawkes | \ | \ | Hawkes | ✓ |
| Poisson | \ | \ | Poisson | ✓ |

## D    Sequence distribution plots

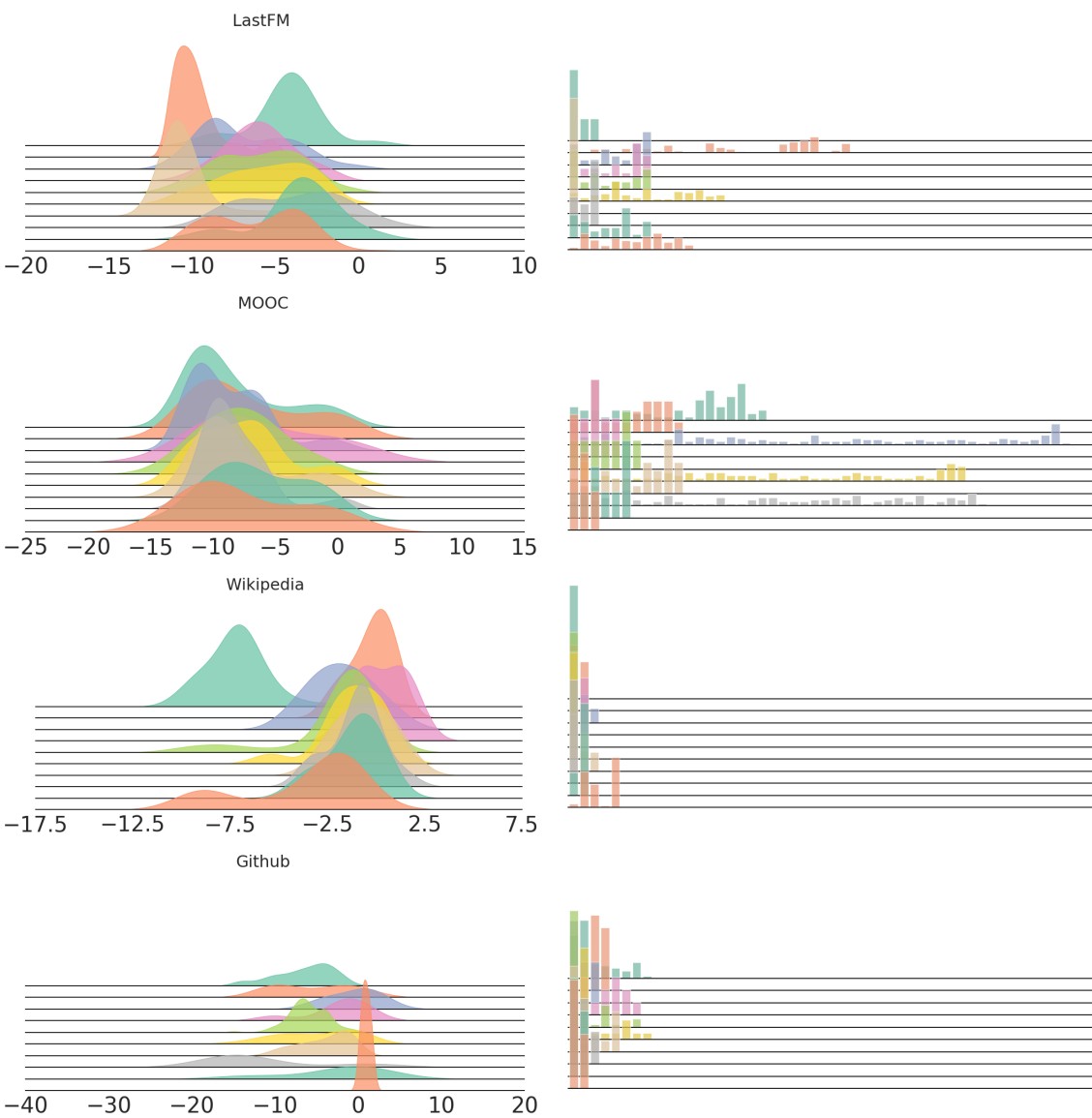

Figure 12: Distribution of log $\tau_i$ (left) and mark distribution (right) for 10 randomly sampled sequences in LastFM, MOOC, Wikipedia, and Github, after preprocessing.

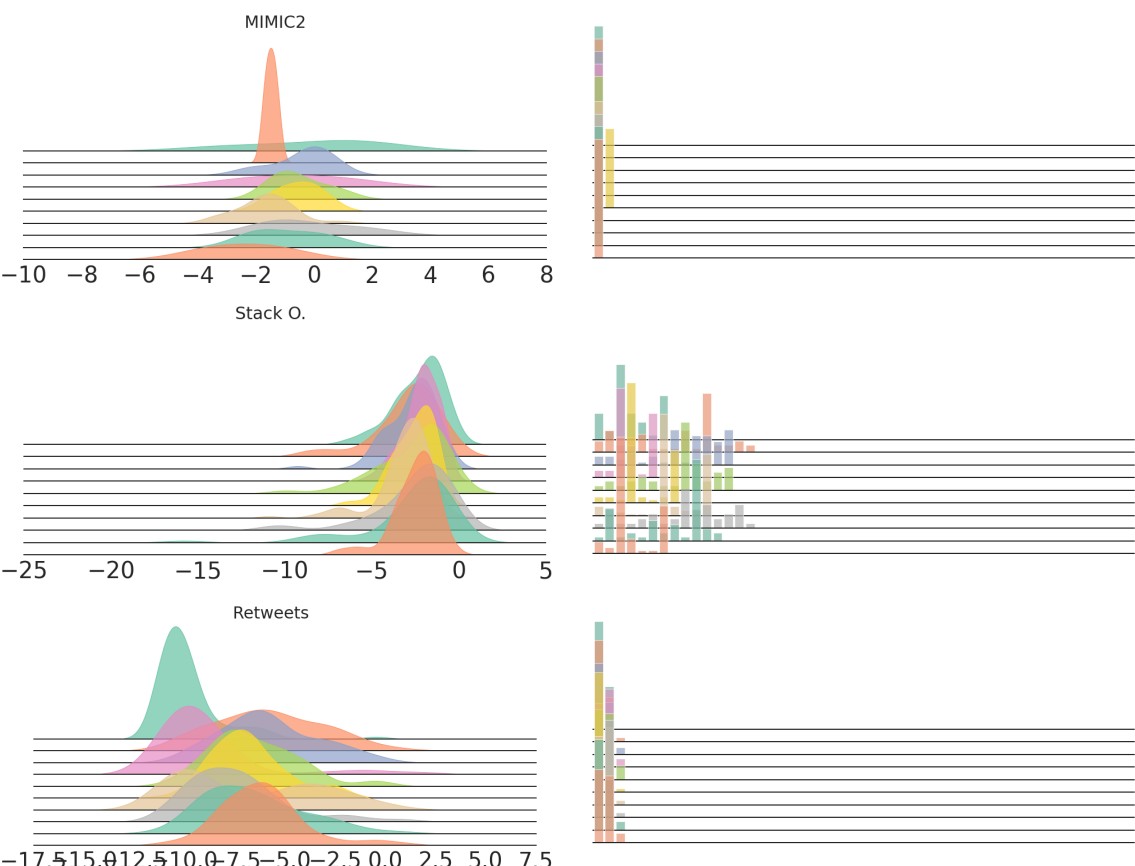

Figure 13: Distribution of $\log \tau_i$ (left) and mark distribution (right) for 10 randomly sampled sequences in MIMIC2, Stack Overflow, and Retweets, after preprocessing.

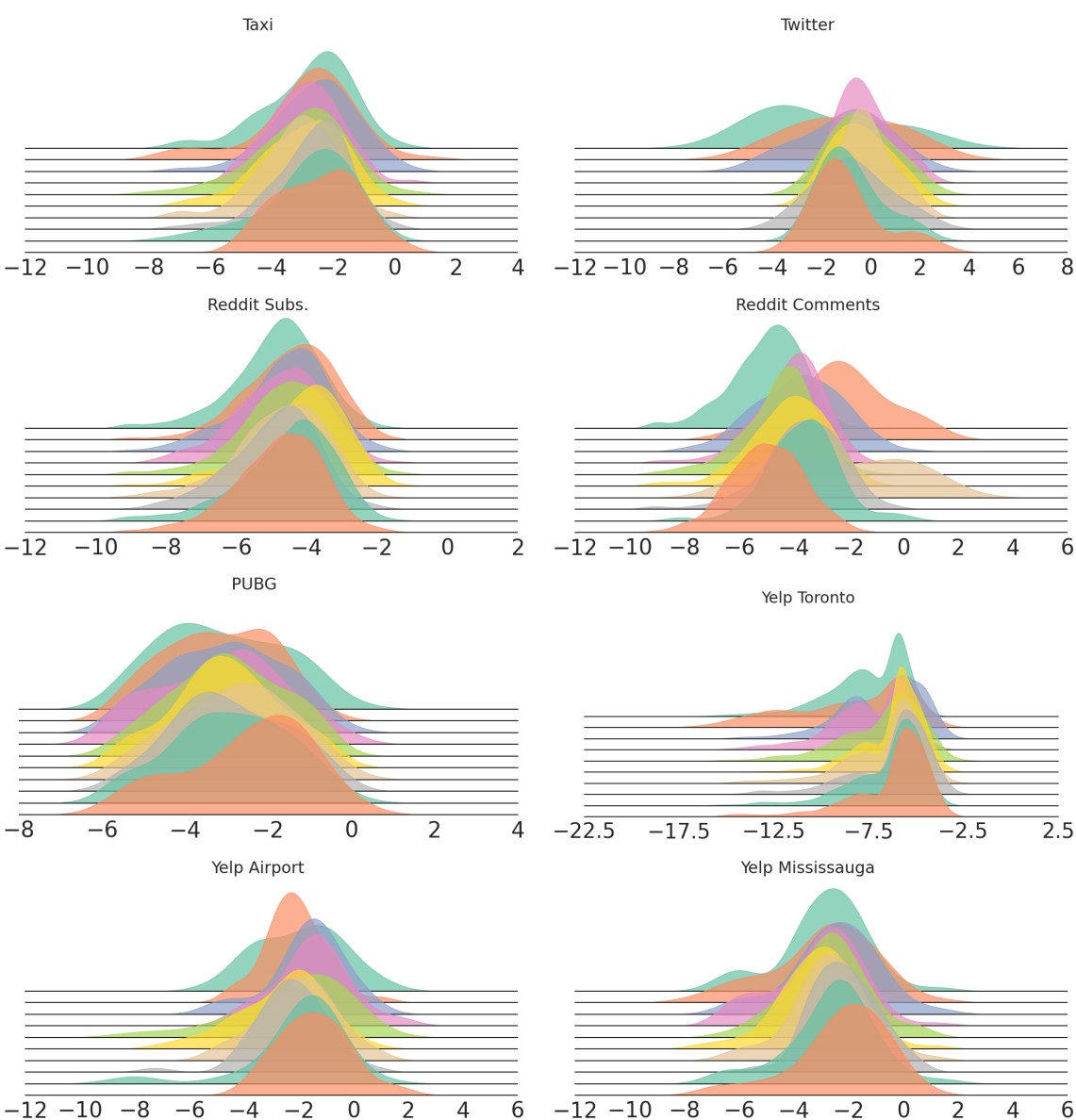

Figure 14: Distribution of log $\tau_i$ for 10 randomly sampled sequences in unmarked datasets, after preprocessing.

## E   Proofs

**1) Proof of Equation (6) in Section 2**. By definition of $\lambda^*(t)$ Rasmussen (2018), we have:

$$\lambda^*(t) = \frac{f^*(t)}{1 - F^*(t)} \tag{64}$$

$$= -\frac{d}{dt} \log\left(1 - F^*(t)\right). \tag{65}$$

Integrating both sides from $t_{i-1}$ to $t$, we get

$$\Lambda^*(t) = \int_{t_{i-1}}^{t} \lambda^*(s)ds = \int_{t_{i-1}}^{t} -d\log\left(1 - F^*(s)\right) \tag{66}$$

$$= -\log\left(1 - F^*(t)\right) + \log\left(1 - \underbrace{F^*(t_{i-1})}_{=0}\right), \tag{67}$$

where $F^*(t_{i-1}) = 0$ results from the point process being simple, i.e. two events cannot occur simultaneously. Rearranging the terms, we find:

$$F^*(t) = 1 - \exp\left(-\Lambda^*(t)\right) = 1 - \exp\left(-\sum_{k=1}^{K} \Lambda_k^*(t)\right). \tag{68}$$

Differentiating with respect to $t$ gives

$$f^*(t) = \frac{d}{dt} F^*(t) = \lambda^*(t)\exp\left(-\Lambda^*(t)\right). \tag{69}$$

Given that $\lambda_k^*(t) = \lambda^*(t)p^*(k|t)$, we finally find

$$f^*(t,k) = f^*(t)p^*(k|t) = \lambda_k^*(t)\exp\left(-\Lambda^*(t)\right) \tag{70}$$

# F   Results on relevant datasets only

The discussion in Section 6 highlighted that some datasets (MIMIC2, Stack Overflow, Taxi, Reddit Subs, Reddit Ask Comments, Yelp Toronto and Yelp Mississauga) might potentially be inappropriate for benchmarking neural TPP models. For completeness, we report in Tables 15, 16, 17, 18, 19 and 20 the results of the aggregation procedure discussed in Section 5.5 without these borderline inadequate datasets included. We found no significance differences with respect to the conclusions of Section 6.

Table 15: Average and median scores, and average ranks of the best combinations per decoder on the NLL-T (top rows) and NLL-M (bottom row) across all relevant marked datasets. Best results are highlighted in bold.

| | Marked Datasets | | | | | | | | | | | | | | |
| | NLL-T | | | PCE | | | NLL-M | | | ECE | | | F1-score | | |
| | Mean | Median | Rank | Mean | Median | Rank | Mean | Median | Rank | Mean | Median | Rank | Mean | Median | Rank |
| --- | --- | --- | --- | --- | --- | --- | --- | --- | --- | --- | --- | --- | --- | --- | --- |
| GRU-EC-LCONCAT | 0.07 | -0.01 | 8.5 | 0.22 | 0.23 | 9.17 | -0.97 | -1.34 | 4.5 | 0.29 | 0.28 | 4.33 | 0.27 | 0.27 | 4.0 |
| GRU-LNM-CONCAT | **-0.86** | **-0.84** | **1.5** | **0.02** | **0.01** | **1.5** | -2.97 | -1.76 | **2.33** | 0.26 | 0.27 | 3.33 | 0.34 | 0.31 | 2.67 |
| GRU-LN-LTO | -0.41 | -0.63 | 4.17 | 0.06 | 0.04 | 5.0 | -0.26 | -0.33 | 5.83 | 0.4 | 0.4 | 7.17 | 0.21 | 0.14 | 7.5 |
| GRU-FNN-LTO | -0.62 | -0.7 | 3.17 | 0.02 | 0.02 | 1.83 | -0.09 | -0.14 | 6.83 | 0.39 | 0.42 | 6.33 | 0.21 | 0.14 | 6.33 |
| GRU-MLP/MC-LCONCAT | -0.41 | -0.38 | 5.67 | 0.11 | 0.11 | 7.0 | -0.58 | -0.47 | 5.5 | 0.29 | 0.32 | 3.5 | 0.25 | 0.22 | 4.33 |
| GRU-RMTPP-LCONCAT | -0.59 | -0.5 | 4.67 | 0.05 | 0.04 | 4.67 | -1.94 | **-1.99** | 2.67 | 0.25 | 0.26 | 2.67 | 0.35 | 0.34 | 2.17 |
| GRU-SA/CM-LE | -0.33 | -0.39 | 6.33 | 0.06 | 0.04 | 4.0 | 0.11 | 0.11 | 8.17 | 0.45 | 0.45 | 8.33 | 0.17 | 0.09 | 9.0 |
| GRU-SA/MC-LE | -0.5 | -0.46 | 4.17 | 0.09 | 0.09 | 5.5 | 0.1 | -0.04 | 8.17 | 0.39 | 0.43 | 7.33 | 0.19 | 0.1 | 7.33 |
| Hawkes | 0.5 | -0.15 | 7.17 | 0.13 | 0.17 | 6.67 | **-6.27** | -1.58 | 3.17 | **0.15** | **0.13** | **2.0** | **0.44** | **0.42** | 2.0 |
| Poisson | 1.72 | 1.31 | 10.83 | 0.31 | 0.35 | 10.5 | 1.39 | 1.26 | 10.83 | 0.49 | 0.49 | 10.67 | 0.14 | 0.06 | **10.33** |
| NH | 1.53 | 1.07 | 9.83 | 0.3 | 0.35 | 10.17 | 0.16 | 0.41 | 8.0 | 0.48 | 0.49 | 10.33 | 0.14 | 0.06 | 10.33 |
| GRU-EC-TEMWL | 0.22 | 0.16 | 8.17 | 0.23 | 0.24 | 9.0 | -1.5 | -1.1 | 5.0 | 0.33 | 0.33 | 5.83 | 0.26 | 0.27 | 5.5 |
| GRU-LNM-CONCAT | **-0.86** | **-0.84** | **1.17** | **0.02** | **0.01** | 1.83 | -2.97 | -1.76 | **2.83** | 0.26 | 0.27 | 4.33 | 0.34 | 0.31 | 3.17 |
| GRU-LN-CONCAT | -0.37 | -0.57 | 4.33 | 0.08 | 0.04 | 4.5 | -3.0 | -1.81 | 3.5 | 0.28 | 0.3 | 3.83 | 0.3 | 0.31 | 4.0 |
| GRU-FNN-LCONCAT | -0.62 | -0.71 | 2.5 | 0.02 | 0.02 | **1.33** | -1.98 | -1.51 | 5.0 | 0.31 | 0.35 | 4.67 | 0.27 | 0.23 | 5.5 |
| GRU-MLP/MC-TEMWL | 0.16 | 0.24 | 7.83 | 0.22 | 0.25 | 8.17 | -0.77 | -0.64 | 7.17 | 0.35 | 0.39 | 6.17 | 0.27 | 0.26 | 6.33 |
| GRU-RMTPP-LCONCAT + B | -0.54 | -0.49 | 3.33 | 0.05 | 0.04 | 4.0 | -3.28 | **-2.0** | 3.17 | 0.2 | 0.16 | 2.67 | 0.37 | 0.33 | 2.17 |
| GRU-SA/CM-LEWL | -0.25 | -0.23 | 5.83 | 0.1 | 0.09 | 5.17 | -0.21 | -0.17 | 8.0 | 0.42 | 0.44 | 8.5 | 0.19 | 0.1 | 9.0 |
| GRU-SA/MC-TEMWL | -0.26 | -0.3 | 5.67 | 0.14 | 0.15 | 6.0 | -0.39 | -0.46 | 7.67 | 0.36 | 0.36 | 6.83 | 0.22 | 0.15 | 7.17 |

Table 16: Average and median scores, as well as average ranks per decoder and variation of event encoding, for relevant marked datasets. Refer to Section 5.5 for details on the aggregation procedure. Best results are highlighted in bold.

| | Marked Datasets | | | | | | | | | | | | | | |
| | NLL-T | | | PCE | | | NLL-M | | | ECE | | | F1-score | | |
| | Mean | Median | Rank | Mean | Median | Rank | Mean | Median | Rank | Mean | Median | Rank | Mean | Median | Rank |
|---|---|---|---|---|---|---|---|---|---|---|---|---|---|---|---|
| EC-TO | 0.83 | 0.59 | 7.17 | 0.26 | 0.28 | 6.0 | 0.15 | 0.19 | 6.17 | 0.45 | 0.46 | 7.17 | 0.17 | 0.08 | 7.0 |
| EC-LTO | 0.82 | 0.6 | 7.33 | 0.26 | 0.27 | 6.5 | 0.12 | 0.15 | 5.67 | 0.45 | 0.46 | 6.67 | 0.17 | 0.08 | 7.5 |
| EC-CONCAT | 0.22 | 0.22 | **2.5** | **0.22** | **0.23** | 2.5 | -0.39 | -0.75 | 3.0 | 0.37 | 0.39 | **1.67** | 0.25 | 0.23 | 2.17 |
| EC-LCONCAT | 0.41 | 0.29 | 4.0 | 0.24 | 0.24 | 4.83 | -0.38 | -0.66 | 3.67 | 0.36 | **0.35** | 2.67 | 0.24 | 0.22 | 3.17 |
| EC-TEM | 0.24 | 0.2 | 4.0 | 0.23 | 0.24 | 4.5 | 0.14 | 0.13 | 6.0 | 0.44 | 0.45 | 6.17 | 0.19 | 0.09 | 5.67 |
| EC-TEMWL | 0.32 | 0.3 | 4.17 | 0.23 | 0.25 | 5.0 | **-0.69** | **-0.84** | **1.83** | **0.35** | 0.35 | 2.0 | **0.27** | **0.24** | **1.33** |
| EC-LE | **0.2** | **0.16** | 3.0 | 0.22 | 0.24 | **2.0** | 0.11 | 0.1 | 5.17 | 0.43 | 0.45 | 5.5 | 0.19 | 0.09 | 5.5 |
| EC-LEWL | 0.26 | 0.25 | 3.83 | 0.23 | 0.25 | 4.67 | -0.15 | -0.45 | 4.5 | 0.39 | 0.41 | 4.17 | 0.24 | 0.22 | 3.67 |
| LNM-TO | -0.55 | -0.57 | 6.5 | **0.02** | 0.02 | 5.0 | 0.03 | 0.11 | 6.83 | 0.45 | 0.46 | 6.83 | 0.17 | 0.09 | 7.5 |
| LNM-LTO | -0.56 | -0.59 | 6.17 | 0.02 | 0.02 | 5.5 | 0.0 | 0.06 | 6.5 | 0.45 | 0.46 | 7.17 | 0.17 | 0.1 | 6.67 |
| LNM-CONCAT | **-0.8** | -0.8 | **1.67** | 0.02 | 0.02 | 4.0 | **-2.48** | -1.47 | **1.5** | **0.28** | **0.26** | 2.0 | **0.35** | **0.35** | **1.5** |
| LNM-LCONCAT | -0.79 | **-0.83** | 2.0 | 0.02 | **0.01** | **2.33** | -1.99 | -1.18 | 3.0 | 0.31 | 0.33 | 2.83 | 0.29 | 0.27 | 3.33 |
| LNM-TEM | -0.74 | -0.78 | 4.5 | 0.02 | 0.01 | 4.0 | -0.0 | 0.05 | 6.5 | 0.44 | 0.45 | 6.0 | 0.19 | 0.1 | 5.83 |
| LNM-TEMWL | -0.7 | -0.72 | 4.83 | 0.03 | 0.02 | 4.0 | -2.17 | **-1.49** | 2.17 | 0.3 | 0.32 | **1.83** | 0.32 | 0.29 | 2.0 |
| LNM-LE | -0.71 | -0.77 | 5.67 | 0.03 | 0.02 | 6.83 | -0.03 | -0.02 | 6.17 | 0.43 | 0.44 | 6.0 | 0.19 | 0.1 | 6.0 |
| LNM-LEWL | -0.71 | -0.75 | 4.67 | 0.02 | 0.02 | 4.33 | -2.13 | -1.18 | 3.33 | 0.32 | 0.31 | 3.33 | 0.29 | 0.26 | 3.17 |
| FNN-TO | 1.15 | 0.68 | 4.5 | 0.26 | 0.31 | 4.67 | 0.34 | 0.37 | 4.5 | 0.48 | 0.48 | 5.5 | 0.14 | 0.07 | 5.17 |
| FNN-LTO | -0.43 | -0.54 | 2.0 | 0.03 | **0.02** | 1.67 | 0.06 | 0.05 | 3.5 | 0.42 | 0.42 | 1.83 | 0.2 | 0.12 | 2.17 |
| FNN-CONCAT | 1.1 | 0.54 | 4.33 | 0.27 | 0.31 | 4.33 | 0.19 | 0.13 | 3.33 | 0.46 | 0.46 | 4.0 | 0.15 | 0.09 | 3.5 |
| FNN-LCONCAT | **-0.59** | **-0.66** | **1.0** | **0.02** | 0.02 | **1.33** | **-1.23** | **-1.17** | **1.67** | **0.33** | **0.35** | **1.17** | **0.25** | **0.22** | **1.0** |
| FNN-LE | 1.14 | 0.72 | 4.33 | 0.27 | 0.32 | 4.0 | 0.3 | 0.34 | 4.0 | 0.48 | 0.48 | 4.0 | 0.15 | 0.06 | 4.67 |
| FNN-LEWL | 1.19 | 0.65 | 4.83 | 0.27 | 0.32 | 4.83 | 0.24 | 0.34 | 4.0 | 0.47 | 0.47 | 4.5 | 0.15 | 0.08 | 4.5 |
| MLP/MC-TO | 0.23 | 0.22 | 6.67 | 0.19 | 0.25 | 5.83 | 0.34 | 0.33 | 6.33 | 0.43 | 0.43 | 6.67 | 0.18 | 0.09 | 7.33 |
| MLP/MC-LTO | -0.19 | -0.17 | 3.33 | **0.11** | **0.1** | **1.67** | 0.5 | 0.25 | 7.17 | 0.41 | 0.45 | 6.33 | 0.19 | 0.1 | 5.83 |
| MLP/MC-CONCAT | 0.03 | 0.06 | 4.83 | 0.2 | 0.22 | 5.83 | -0.03 | -0.18 | 3.17 | 0.37 | 0.39 | 3.0 | 0.24 | 0.22 | 3.33 |
| MLP/MC-LCONCAT | **-0.33** | **-0.28** | **1.83** | 0.11 | 0.1 | 1.67 | -0.17 | -0.42 | 3.17 | **0.33** | **0.34** | 2.67 | 0.24 | 0.21 | 2.83 |
| MLP/MC-TEM | 0.04 | 0.03 | 5.67 | 0.19 | 0.23 | 6.0 | 0.26 | 0.18 | 5.17 | 0.41 | 0.41 | 5.83 | 0.19 | 0.1 | 6.67 |
| MLP/MC-TEMWL | 0.24 | 0.32 | 7.33 | 0.22 | 0.25 | 7.83 | **-0.6** | **-0.43** | **2.17** | 0.35 | 0.38 | **2.5** | **0.26** | **0.26** | **1.67** |
| MLP/MC-LE | -0.11 | -0.13 | 3.17 | 0.16 | 0.19 | 3.5 | 0.38 | 0.22 | 5.67 | 0.4 | 0.42 | 6.33 | 0.19 | 0.1 | 5.83 |
| MLP/MC-LEWL | -0.09 | -0.13 | 3.17 | 0.17 | 0.2 | 3.67 | -0.39 | -0.33 | 3.17 | 0.34 | 0.35 | 2.67 | 0.24 | 0.2 | 2.5 |
| RMTPP-TO | 0.32 | 0.23 | 7.33 | 0.2 | 0.24 | 6.83 | 0.07 | 0.09 | 6.17 | 0.44 | 0.45 | 7.17 | 0.17 | 0.09 | 7.33 |
| RMTPP-LTO | -0.28 | -0.42 | 3.83 | 0.06 | 0.07 | 2.17 | 0.05 | 0.03 | 6.5 | 0.44 | 0.44 | 6.67 | 0.18 | 0.11 | 6.33 |
| RMTPP-CONCAT | 0.0 | 0.06 | 4.17 | 0.19 | 0.22 | 4.5 | -1.66 | -1.29 | 3.0 | 0.29 | 0.34 | 2.33 | 0.31 | 0.3 | 2.5 |
| RMTPP-LCONCAT | **-0.52** | **-0.47** | **1.5** | **0.05** | **0.05** | **1.0** | -1.85 | **-1.57** | 3.0 | **0.26** | **0.27** | **1.83** | 0.34 | 0.32 | **1.67** |
| RMTPP-TEM | 0.03 | 0.04 | 5.33 | 0.19 | 0.22 | 5.83 | 0.06 | 0.05 | 5.67 | 0.43 | 0.44 | 6.5 | 0.19 | 0.11 | 6.33 |
| RMTPP-TEMWL | 0.08 | 0.12 | 4.83 | 0.19 | 0.22 | 5.83 | **-2.28** | -1.42 | **1.67** | 0.27 | 0.29 | 2.17 | **0.35** | **0.37** | 1.83 |
| RMTPP-LE | 0.01 | 0.02 | 4.17 | 0.19 | 0.22 | 4.5 | 0.07 | 0.05 | 6.0 | 0.43 | 0.44 | 5.67 | 0.19 | 0.11 | 6.0 |
| RMTPP-LEWL | 0.05 | 0.12 | 4.83 | 0.19 | 0.23 | 5.5 | -1.24 | -1.03 | 4.0 | 0.32 | 0.38 | 3.67 | 0.28 | 0.28 | 4.0 |
| SA/CM-TO | -0.06 | -0.09 | 4.67 | 0.08 | 0.06 | 4.17 | 0.75 | 0.22 | 4.5 | 0.46 | 0.46 | 5.0 | 0.14 | 0.1 | 4.33 |
| SA/CM-LTO | -0.21 | -0.18 | 2.5 | **0.05** | 0.05 | **2.33** | 0.14 | 0.07 | 3.5 | 0.44 | **0.44** | 2.67 | **0.19** | 0.1 | 2.5 |
| SA/CM-CONCAT | -0.12 | -0.29 | 3.5 | 0.08 | 0.07 | 3.17 | 0.78 | 0.2 | 4.5 | 0.46 | 0.46 | 3.5 | 0.14 | 0.1 | 3.0 |
| SA/CM-LCONCAT | -0.02 | 0.05 | 4.33 | 0.08 | 0.06 | 3.5 | 0.16 | 0.08 | **2.33** | 0.45 | 0.45 | 3.0 | 0.19 | **0.11** | 3.0 |
| SA/CM-LE | **-0.3** | **-0.36** | **2.33** | 0.06 | **0.04** | 2.83 | 0.1 | 0.1 | 3.17 | 0.45 | 0.45 | 4.17 | 0.18 | 0.08 | 4.5 |
| SA/CM-LEWL | -0.19 | -0.19 | 3.67 | 0.1 | 0.08 | 5.0 | **-0.12** | **-0.11** | 3.0 | **0.42** | 0.44 | 2.67 | 0.19 | 0.09 | 3.67 |
| SA/MC-TO | 1.4 | 1.0 | 7.5 | 0.29 | 0.34 | 6.67 | 0.22 | 0.31 | 5.5 | 0.48 | 0.49 | 7.0 | 0.14 | 0.06 | 7.33 |
| SA/MC-LTO | 1.24 | 0.82 | 5.67 | 0.28 | 0.32 | 6.0 | 0.25 | 0.4 | 6.0 | 0.47 | 0.48 | 6.33 | 0.15 | 0.06 | 6.17 |
| SA/MC-CONCAT | 0.63 | 0.85 | 6.67 | 0.24 | 0.31 | 6.33 | 0.03 | 0.11 | 5.0 | 0.44 | 0.49 | 5.83 | 0.18 | 0.06 | 5.83 |
| SA/MC-LCONCAT | 0.52 | 0.74 | 5.17 | 0.24 | 0.3 | 6.5 | 0.1 | 0.15 | 5.17 | 0.42 | 0.48 | 5.0 | 0.18 | 0.06 | 5.67 |
| SA/MC-TEM | -0.35 | -0.3 | 3.33 | 0.13 | 0.14 | 3.33 | 0.18 | 0.05 | 5.0 | 0.41 | 0.44 | 4.33 | 0.18 | 0.08 | 4.17 |
| SA/MC-TEMWL | -0.25 | -0.26 | 4.17 | 0.14 | 0.14 | 4.17 | **-0.32** | **-0.27** | **2.17** | 0.37 | **0.38** | 2.0 | **0.22** | **0.14** | **1.67** |
| SA/MC-LE | **-0.48** | -0.42 | **1.5** | **0.09** | **0.09** | **1.33** | 0.12 | -0.03 | 4.67 | 0.4 | 0.43 | 3.67 | 0.19 | 0.09 | 3.5 |
| SA/MC-LEWL | -0.45 | **-0.45** | 2.0 | 0.1 | 0.1 | 1.67 | -0.1 | -0.13 | 2.5 | 0.37 | 0.42 | **1.83** | 0.2 | 0.1 | 1.67 |

Table 17: Average and median scores, as well as average ranks per decoder and variation of event encoding, for relevant marked datasets. Refer to Section 5.5 for details on the aggregation procedure. Best results are highlighted in bold.

| | NLL-T | | | PCE | | | NLL-M | | | ECE | | | F1-score | | |
|---|---|---|---|---|---|---|---|---|---|---|---|---|---|---|---|
| | Mean | Median | Rank | Mean | Median | Rank | Mean | Median | Rank | Mean | Median | Rank | Mean | Median | Rank |
| CONS-EC | 1.66 | 1.26 | 3.0 | 0.3 | 0.35 | 2.83 | 0.7 | 0.46 | 2.67 | 0.48 | 0.49 | 3.0 | 0.14 | 0.06 | **3.0** |
| SA-EC | 0.68 | 0.54 | 2.0 | 0.25 | 0.27 | 2.17 | -0.02 | -0.16 | 2.0 | 0.43 | 0.42 | 2.0 | 0.2 | 0.15 | 1.83 |
| GRU-EC | **0.15** | **0.15** | **1.0** | **0.22** | **0.23** | **1.0** | **-0.25** | **-0.4** | **1.33** | **0.39** | **0.38** | **1.0** | **0.22** | **0.17** | 1.17 |
| CONS-LNM | -0.34 | -0.5 | 2.83 | **0.02** | 0.02 | 2.33 | 0.65 | 0.39 | 3.0 | 0.48 | 0.49 | 3.0 | 0.14 | 0.06 | **3.0** |
| SA-LNM | -0.6 | -0.66 | 1.83 | 0.02 | 0.02 | 2.17 | -0.8 | -0.44 | 1.83 | 0.39 | 0.39 | 1.83 | 0.24 | 0.17 | 1.83 |
| GRU-LNM | **-0.79** | **-0.8** | **1.33** | 0.03 | **0.01** | **1.5** | **-1.39** | **-0.89** | **1.17** | **0.35** | **0.37** | **1.17** | **0.26** | **0.2** | 1.17 |
| CONS-FNN | 0.94 | 0.61 | 3.0 | 0.2 | 0.23 | 2.67 | 0.76 | 0.41 | 3.0 | 0.47 | 0.48 | 2.67 | 0.16 | 0.07 | **3.0** |
| SA-FNN | 0.68 | 0.38 | 1.83 | 0.19 | 0.22 | 2.0 | 0.15 | 0.16 | 2.0 | 0.45 | **0.45** | 1.83 | 0.17 | **0.11** | 1.83 |
| GRU-FNN | **0.51** | **0.12** | **1.17** | **0.18** | **0.21** | **1.33** | **-0.18** | **-0.08** | **1.0** | **0.43** | 0.45 | **1.5** | **0.18** | 0.1 | 1.17 |
| CONS-MLP/MC | 0.46 | 0.43 | 2.67 | 0.19 | 0.22 | 2.0 | 0.88 | 0.58 | 2.67 | 0.44 | 0.46 | 3.0 | 0.17 | 0.07 | **3.0** |
| SA-MLP/MC | 0.09 | 0.15 | 2.0 | **0.17** | 0.21 | 2.33 | 0.13 | -0.04 | 2.0 | 0.39 | 0.4 | 2.0 | 0.21 | 0.15 | 1.83 |
| GRU-MLP/MC | **-0.14** | **-0.12** | **1.33** | 0.17 | **0.19** | **1.67** | **-0.06** | **-0.23** | **1.33** | **0.36** | **0.37** | **1.0** | **0.22** | **0.17** | 1.17 |
| CONS-RMTPP | 0.81 | 0.56 | 3.0 | 0.18 | 0.21 | 2.33 | 1.1 | 0.59 | 2.67 | 0.47 | 0.48 | 3.0 | 0.13 | 0.05 | **3.0** |
| SA-RMTPP | 0.1 | 0.1 | 1.83 | 0.17 | 0.19 | 2.17 | -0.49 | -0.41 | 2.17 | 0.38 | 0.39 | 1.83 | 0.24 | 0.18 | 1.83 |
| GRU-RMTPP | **-0.18** | **-0.17** | **1.17** | **0.15** | **0.17** | **1.5** | **-1.2** | **-0.83** | **1.17** | **0.34** | **0.37** | **1.17** | **0.26** | **0.22** | 1.17 |
| CONS-SA/CM | -0.02 | -0.01 | 2.33 | 0.09 | 0.1 | 2.17 | 0.75 | 0.47 | 2.67 | **0.45** | **0.45** | 2.0 | **0.17** | 0.09 | 1.83 |
| SA-SA/CM | -0.14 | **-0.16** | 1.67 | **0.07** | **0.06** | 2.17 | **0.29** | **0.1** | 1.83 | 0.45 | 0.45 | **1.83** | 0.17 | **0.1** | 1.67 |
| GRU-SA/CM | **-0.16** | -0.07 | 2.0 | 0.07 | 0.07 | **1.67** | 0.31 | 0.1 | **1.5** | 0.45 | 0.45 | 2.17 | 0.17 | 0.1 | **2.5** |
| CONS-SA/MC | 0.8 | 0.52 | 3.0 | 0.22 | 0.25 | 2.83 | 0.76 | 0.44 | 3.0 | 0.46 | 0.47 | 3.0 | 0.16 | 0.07 | **3.0** |
| SA-SA/MC | 0.29 | 0.23 | 1.67 | **0.19** | **0.21** | 1.83 | 0.1 | 0.13 | 1.83 | **0.42** | 0.46 | 2.0 | **0.18** | **0.08** | 1.5 |
| GRU-SA/MC | **0.28** | **0.22** | **1.33** | 0.19 | 0.21 | **1.33** | **0.02** | **0.07** | **1.17** | 0.42 | **0.45** | **1.0** | 0.18 | 0.08 | 1.5 |

Table 18: Average, median, worst scores, and average ranks of the best combinations per decoder on the NLL-T across all relevant unmarked datasets. Best results are highlighted in bold.

| | Unmarked Datasets | | | | | | | |
| --- | --- | --- | --- | --- | --- | --- | --- | --- |
| | NLL-T | | | | PCE | | | |
| | Mean | Median | Worst | Rank | Mean | Median | Worst | Rank |
| GRU-EC-TEM + B | -0.21 | -0.28 | 0.16 | 7.0 | 0.06 | 0.02 | 0.13 | 8.33 |
| GRU-LNM-TEM | **-3.7** | **-1.03** | **-0.83** | **1.67** | **0.01** | **0.01** | **0.01** | **1.67** |
| GRU-LN-LE | -0.25 | -0.46 | 0.47 | 6.0 | 0.02 | 0.02 | 0.04 | 6.67 |
| GRU-FNN-LTO | -1.03 | -0.79 | -0.49 | 4.0 | 0.01 | 0.01 | 0.01 | 2.33 |
| GRU-MLP/MC-LTO | -0.57 | -0.57 | -0.5 | 4.67 | 0.02 | 0.01 | 0.03 | 5.67 |
| GRU-RMTPP-TEM + B | -0.51 | -0.54 | -0.36 | 5.67 | 0.02 | 0.01 | 0.05 | 5.0 |
| GRU-SA/CM-TO | -0.71 | -0.56 | 0.73 | 5.67 | 0.02 | 0.01 | 0.03 | 6.0 |
| GRU-SA/MC-LE | -0.81 | -0.87 | -0.51 | 2.33 | 0.01 | 0.01 | 0.02 | 5.0 |
| Hawkes | 0.05 | 0.06 | 0.16 | 8.0 | 0.01 | 0.01 | 0.02 | 4.33 |
| Poisson | 1.54 | 1.32 | 2.14 | 11.0 | 0.13 | 0.12 | 0.2 | 11.0 |
| NH | 0.87 | 0.9 | 0.96 | 10.0 | 0.09 | 0.06 | 0.17 | 10.0 |

Table 19: Average, median, and worst scores, as well as average ranks per decoder and variation of event encoding, for relevant unmarked datasets. Refer to Section 5.5 for details on the aggregation procedure. Best scores are highlighted in bold.

| | Unmarked Datasets | | | | | | | |
| --- | --- | --- | --- | --- | --- | --- | --- | --- |
| | NLL-T | | | | PCE | | | |
| | Mean | Median | Worst | Rank | Mean | Median | Worst | Rank |
| EC-TO | 0.36 | 0.35 | 0.6 | 3.33 | 0.08 | 0.04 | 0.16 | 2.67 |
| EC-LTO | 0.39 | 0.34 | 0.64 | 3.67 | 0.08 | 0.04 | 0.16 | 4.0 |
| EC-TEM | **-0.03** | **-0.13** | **0.31** | **1.0** | **0.06** | **0.03** | **0.14** | **1.0** |
| EC-LE | 0.17 | 0.04 | 0.46 | 2.0 | 0.07 | 0.03 | 0.15 | 2.33 |
| LNM-TO | -2.19 | -0.73 | -0.53 | 3.0 | **0.01** | **0.01** | **0.01** | 2.67 |
| LNM-LTO | -2.18 | -0.72 | -0.49 | 3.33 | 0.01 | 0.01 | 0.01 | 2.33 |
| LNM-TEM | **-2.64** | **-0.82** | **-0.76** | **1.33** | 0.01 | 0.01 | 0.01 | **1.67** |
| LNM-LE | -1.55 | -0.82 | -0.56 | 2.33 | 0.01 | 0.01 | 0.01 | 3.33 |
| FNN-TO | 0.54 | 0.51 | 0.62 | 2.33 | 0.08 | 0.04 | 0.15 | 2.33 |
| FNN-LTO | **-0.78** | **-0.65** | **-0.25** | **1.0** | **0.01** | **0.01** | **0.01** | **1.0** |
| FNN-LE | 0.51 | 0.52 | 0.54 | 2.67 | 0.08 | 0.05 | 0.15 | 2.67 |
| MLP/MC-TO | -0.03 | 0.01 | 0.1 | 3.33 | 0.04 | 0.03 | 0.08 | 3.33 |
| MLP/MC-LTO | -0.21 | -0.17 | 0.06 | 2.33 | **0.02** | 0.03 | **0.03** | **1.33** |
| MLP/MC-TEM | -0.12 | -0.14 | -0.04 | 2.33 | 0.04 | 0.03 | 0.09 | 3.0 |
| MLP/MC-LE | **-0.31** | **-0.39** | **-0.12** | **2.0** | 0.03 | **0.02** | 0.06 | 2.33 |
| RMTPP-TO | -0.11 | -0.16 | 0.11 | 3.33 | 0.04 | 0.02 | 0.08 | 3.33 |
| RMTPP-LTO | -0.12 | -0.15 | 0.2 | 3.0 | **0.02** | 0.02 | **0.03** | 2.33 |
| RMTPP-TEM | **-0.33** | **-0.29** | **-0.25** | **1.33** | 0.03 | **0.01** | 0.07 | 2.33 |
| RMTPP-LE | -0.25 | -0.24 | -0.14 | 2.33 | 0.03 | 0.01 | 0.08 | **2.0** |
| SA/CM-TO | **-0.34** | -0.5 | 0.72 | 2.0 | 0.02 | 0.02 | 0.03 | 2.33 |
| SA/CM-LTO | 0.67 | **-0.55** | 3.38 | 2.33 | 0.04 | 0.02 | 0.1 | 2.67 |
| SA/CM-LE | -0.25 | -0.3 | **0.19** | **1.67** | **0.01** | **0.01** | **0.01** | **1.0** |
| SA/MC-TO | 0.76 | 0.72 | 0.96 | 4.0 | 0.08 | 0.07 | 0.15 | 4.0 |
| SA/MC-LTO | 0.51 | 0.42 | 0.89 | 3.0 | 0.07 | 0.06 | 0.14 | 3.0 |
| SA/MC-TEM | -0.57 | -0.66 | -0.39 | 2.0 | **0.01** | **0.01** | **0.02** | **1.0** |
| SA/MC-LE | **-0.69** | **-0.8** | **-0.43** | **1.0** | 0.01 | 0.01 | 0.02 | 2.0 |

Table 20: Average, median, and worst scores, as well as average ranks per decoder and variation of history encoder, for relevant unmarked datasets. Refer to Section 5.5 for details on the aggregation procedure. Best scores are highlighted in bold.

| | Unmarked Datasets | | | | | | | |
| --- | --- | --- | --- | --- | --- | --- | --- | --- |
| | NLL-T | | | | PCE | | | |
| | Mean | Median | Worst | Rank | Mean | Median | Worst | Rank |
| CONS-EC | 1.29 | 1.14 | 2.14 | 3.0 | 0.1 | 0.07 | 0.2 | 3.0 |
| SA-EC | 0.61 | 0.55 | 0.75 | 2.0 | 0.09 | 0.05 | 0.17 | 2.0 |
| GRU-EC | **-0.17** | **-0.27** | **0.25** | **1.0** | **0.06** | **0.02** | **0.14** | **1.0** |
| CONS-LNM | -1.83 | -0.25 | 0.12 | 2.33 | **0.01** | **0.01** | 0.02 | **1.67** |
| SA-LNM | -1.24 | -0.64 | -0.24 | 2.67 | 0.01 | 0.01 | **0.01** | 2.33 |
| GRU-LNM | **-3.04** | **-0.93** | **-0.9** | **1.0** | 0.01 | 0.01 | 0.01 | 2.0 |
| CONS-FNN | 0.61 | 0.37 | 1.26 | 2.67 | 0.06 | **0.03** | 0.13 | 2.0 |
| SA-FNN | 0.15 | 0.16 | 0.34 | 2.0 | **0.05** | 0.03 | **0.1** | **1.33** |
| GRU-FNN | **0.03** | **0.14** | **0.17** | **1.33** | 0.06 | 0.03 | 0.11 | 2.67 |
| CONS-MLP/MC | 0.51 | 0.58 | 0.74 | 2.67 | 0.04 | 0.03 | 0.07 | 2.67 |
| SA-MLP/MC | 0.1 | 0.23 | 0.33 | 2.33 | 0.04 | 0.03 | **0.06** | 2.0 |
| GRU-MLP/MC | **-0.44** | **-0.41** | **-0.38** | **1.0** | **0.03** | **0.02** | 0.06 | **1.33** |
| CONS-RMTPP | 0.45 | 0.6 | 0.79 | 2.67 | **0.03** | 0.02 | **0.05** | **1.67** |
| SA-RMTPP | 0.03 | -0.04 | 0.29 | 2.33 | 0.03 | 0.02 | 0.07 | 2.67 |
| GRU-RMTPP | **-0.43** | **-0.37** | **-0.33** | **1.0** | 0.03 | **0.01** | 0.06 | 1.67 |
| CONS-SA/CM | 0.7 | 0.12 | 2.02 | 3.0 | 0.03 | 0.02 | **0.04** | 2.67 |
| SA-SA/CM | **0.01** | -0.43 | **0.99** | 1.67 | **0.02** | 0.02 | 0.04 | **1.33** |
| GRU-SA/CM | 0.04 | **-0.61** | 1.54 | **1.33** | 0.03 | **0.01** | 0.06 | 2.0 |
| CONS-SA/MC | 0.54 | 0.73 | 0.83 | 3.0 | **0.05** | 0.04 | 0.09 | 2.67 |
| SA-SA/MC | 0.05 | -0.06 | 0.27 | 2.0 | 0.05 | **0.03** | **0.08** | **1.33** |
| GRU-SA/MC | **-0.05** | **-0.16** | **0.25** | **1.0** | 0.05 | 0.04 | 0.08 | 2.0 |

# G   Critical distance diagrams for event encodings and history encoders

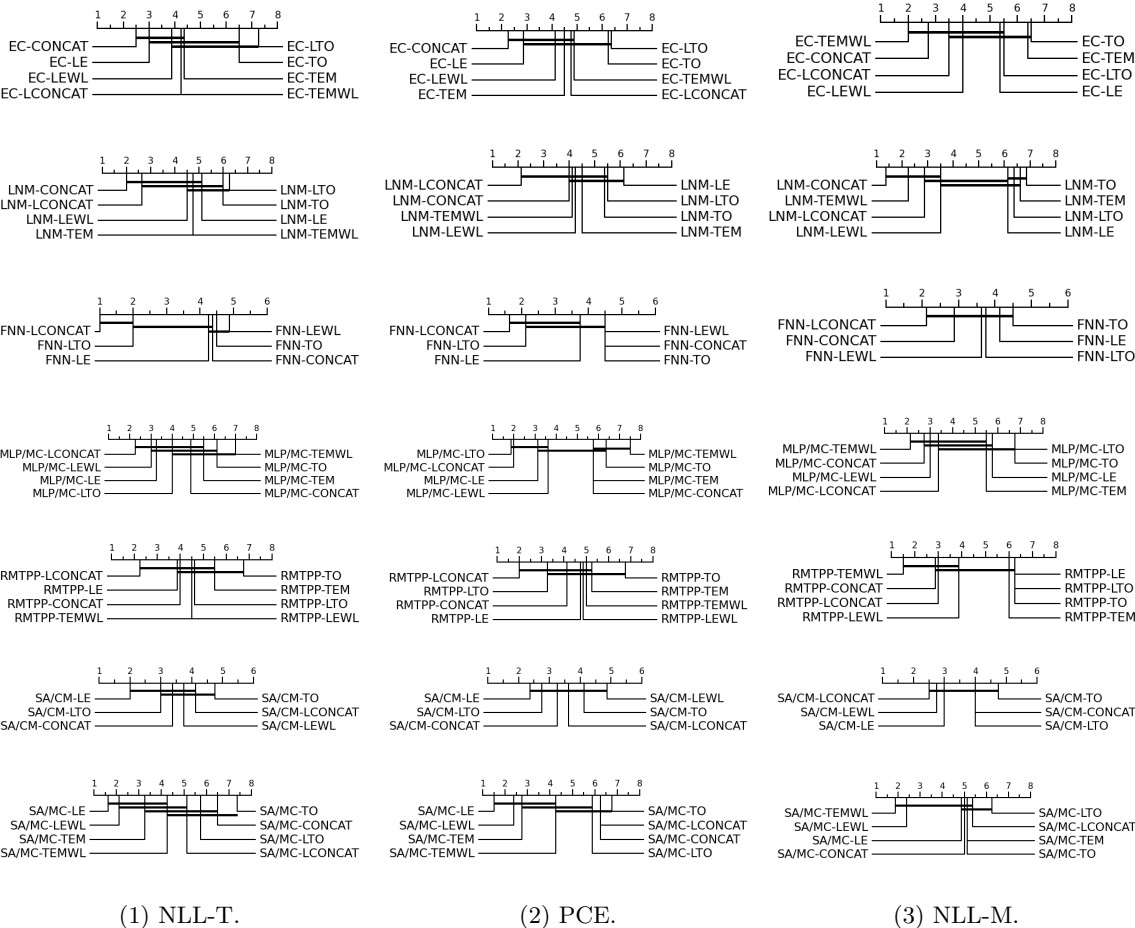

(1) NLL-T.          (2) PCE.          (3) NLL-M.

Figure 15: Critical Distance (CD) diagrams for the NLL-T, PCE and NLL-M at the $\alpha = 0.10$ significance level for all event encoding-decoder combinations of Table 4. The combinations' average ranks are displayed on top, and a bold line joins the combinations that are not statistically different.

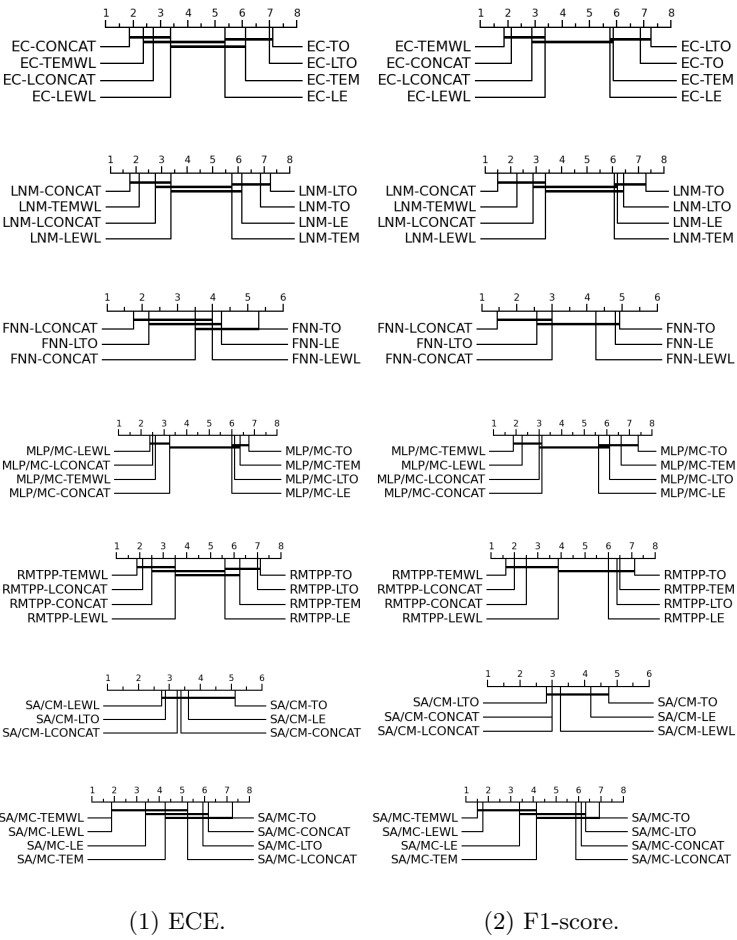

(1) ECE.                    (2) F1-score.

Figure 16: Critical Distance (CD) diagrams for the ECE and F1-score at the $\alpha = 0.10$ significance level for all event encoding-decoder combinations of Table 4. The combinations' average ranks are displayed on top, and a bold line joins the combinations that are not statistically different.

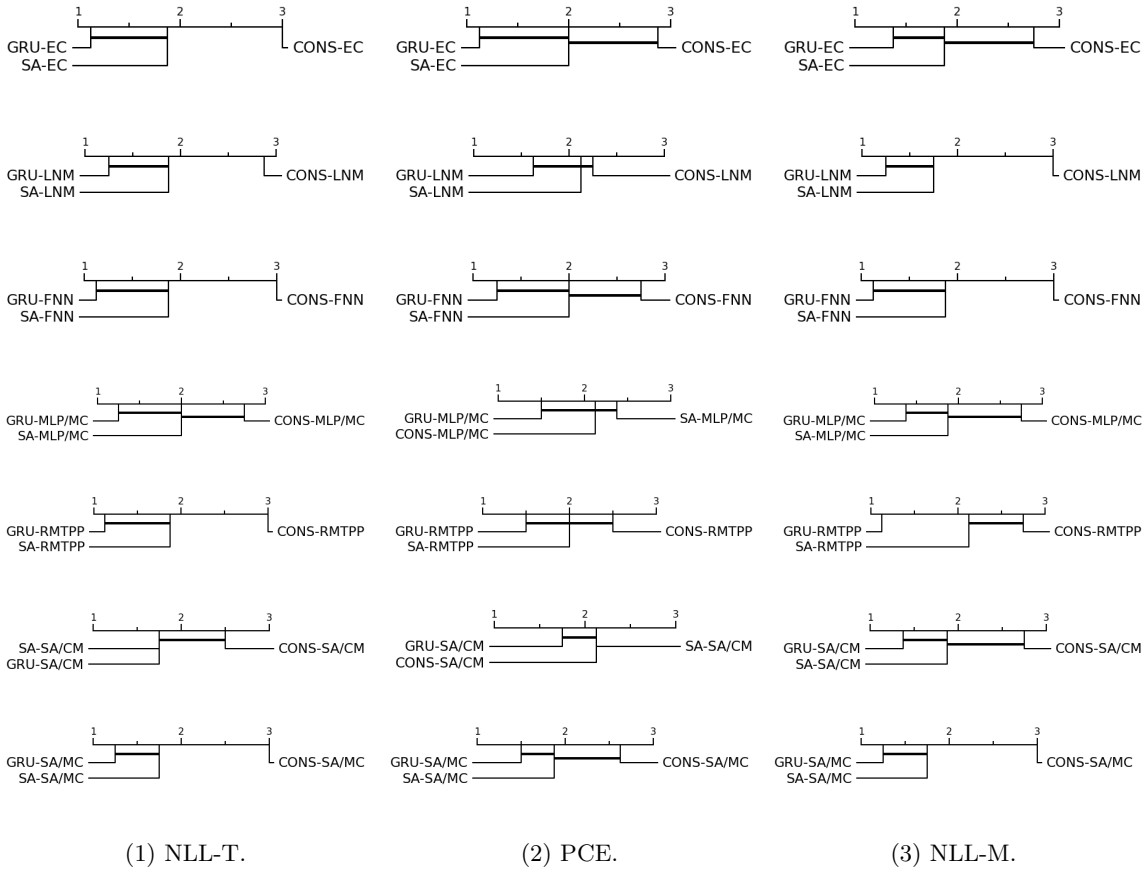

(1) NLL-T.                    (2) PCE.                    (3) NLL-M.

Figure 17: Critical Distance (CD) diagrams for the NLL-T, PCE and NLL-M at the $\alpha = 0.10$ significance level for all event history encoder-decoder combinations of Table 5. The combinations' average ranks are displayed on top, and a bold line joins the combinations that are not statistically different.

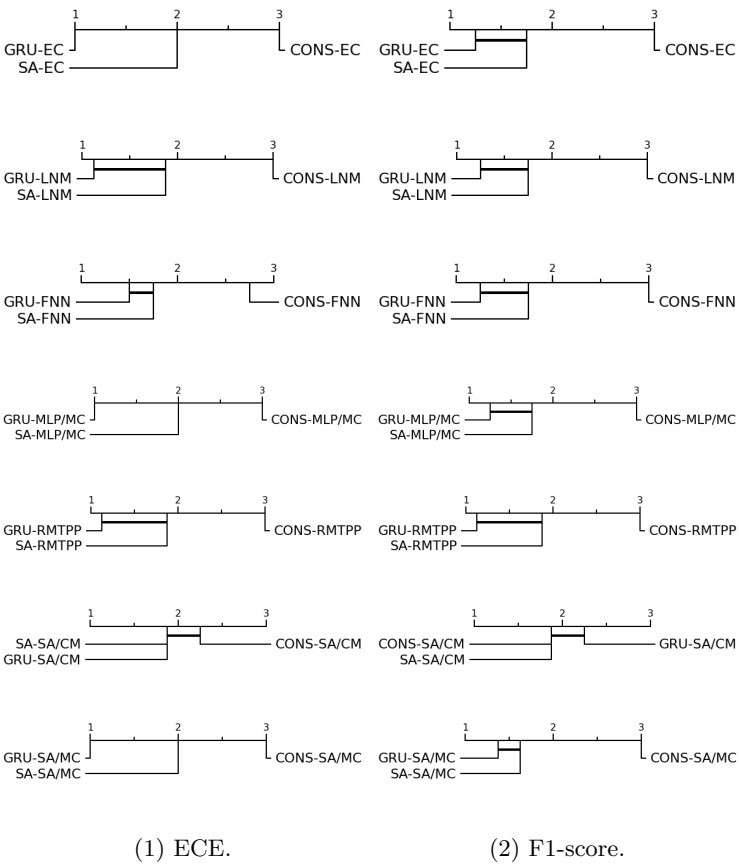

(1) ECE.                              (2) F1-score.

Figure 18: Critical Distance (CD) diagrams for the ECE and F1-score at the $\alpha = 0.10$ significance level for all event history encoder-decoder combinations of Table 5. The combinations' average ranks are displayed on top, and a bold line joins the combinations that are not statistically different.

