# OpenReview forum: "On the Predictive Accuracy of Neural Temporal Point Process Models for Continuous-time Event Data"
_TMLR — Accepted by TMLR_

### Review · Reviewer_ZZrh · 2023-04-20

**Summary Of Contributions:**


This work pursues a large-scale systematic evaluation of neural temporal point process (TPP) methods.

The major contributions seem to be

1) a systematic evaluation

- larger than previous works in terms of size (15 datasets)
- covers calibration as well as likelihood performance
- designed to pinpoint effect of each architectural component (event encoder, history encoder, decoder)

2) several insights from the evaluation could inform future practice

- several datasets studied here are discouraged as good benchmarks for neural TPPs due to small variability between small parametric and large neural models
- fixed-context-length encoders of only the last few events do about as well as encoders of arbitrarily long history
- mark distributions are not well calibrated by neural TPPs (basic parametric models do better)
- self-attention needs to use vectorial representations of time for good performance


**Audience:**

Yes

**Broader Impact Concerns:**

No concerns. I don't think a Broader Impact statement is necessary here, though there could be one.

Some data analyzed here do originally arise from humans (e.g. the health records data in MIMIC2, the social media data in Reddit).
However, these datasets seem to be suitably de-identified.

**Claims And Evidence:**

Yes

**Requested Changes:**

Requested Content Changes (critical)
-----------------------------------

## W1: Why include datasets you identify as borderline inadequate?

I like the critical eye this work puts on existing datasets on page 31, where 7 of the 15 datasets (MIMIC2, Stack overflow, Taxi, Reddit Subs, and 3 more) are identified as potentially inadequate because "all decoders achieve comparable performance"

However, this leaves me confused, because all the earlier tables about ranking methods used these datasets.

I'm eager to hear from the authors: Why not take a stronger stance and remove these less-suitable datasets from your own experiments? Would any of the conclusions from earlier sections change?

In my mind, steering the community away from un-interesting datasets could be a valuable contribution here, but the fact that you've used these datasets in all experiments seems to perpetuate these datasets. A reader who isn't careful to read all the way thru page 31 would totally miss the recommendation to not use these.


## W2: Computational time tradeoffs not assessed at all

A big missed opportunity here is the lack of reporting of runtime or othe resources required for each method. Given that there isn't often a clear "winner" in terms of the critical distance diagrams (Fig 4), in practice many readers will want to know how to manage tradeoffs in terms of performance (likelihood/calibration) vs resources required (training time).

I wonder if the authors could comment on this: can you say anything about which methods are recommended from a runtime perspective?

To be clear: I don't necessarily want you to go run more experiments, but I wonder if there's some insight you can extract from the experiments you already did that would be useful.


## W3: Tough to parse main message from Tables

I have to say I found Tables 4 and 5 rather tough to parse the main takeaway messages with so many methods listed

I wonder if these can be punted to appendix, and the main paper can focus more on the same kind of information but converted to Critical Distance diagrams like in Fig 4.  Those seem much easier for readers to browse to get the main idea about which approaches are working truly better than others versus similar to others.


## W4: Requested Presentations Changes (critical)

Regarding W4, as someone quite familiar with probabilistic modeling in general, but who doesn't work directly with TPPs often, I felt the presentation in Sec 2 could be improved.

#### 1) adjust text around Eq 1-3 to better introduce quantities like the MCIF $\lambda(\cdot)$ to unfamiliar readers.

I suggest first introducing f and F from Eq 1 as a pdf and a CDF over times in interval $[t_{i-1}, \infty]$, then explaining how the MCIF $\lambda$ is defined in terms of these. The "instantaneous probability" aspect was confusing to me, because it seems at first from Eq 1 you can treat $\lambda$ as a PDF, but then in requirement $R_2$ later we find it integrates to infinity, not one. I feel this integration requirement should be better explained.

#### 2) typo in Eq 3, replace "] t_i-1, t]" with "[ t_i-1, t]"

#### 3) adjust explanation around Eq 5 of the cumulative MCIF $\Lambda$

the exp term here seems unmotivated, but perhaps at least a pointer to page 8 of the lecture notes of Rasmussen (2018) would clear things up, as you can define $\lambda(t)$ as the negative derivative of the log of 1 -F wrt t.

#### 4) clarify in Eq 6 the sum of $\Lambda$ terms implied by the "..."

what time values are provided, just t_1, t_2, ... t_i-1, and then t? the bounds of the sum are vague to me, perhaps because the first t has a subscript but last one doesn't




Minor Presentation Changes (suggested)
--------------------------------------

page 2: "solemnly evaluated" should maybe be "solely evaluated"

page 6: the footnote defining e_i is awkwardly placed (looks like a superscript of the $\mathbb{R}$ term)

around Eq 10-11, should be clear that in addition to a given sequence of n events, you need to specify the max time T of your observation window

text above Eq 12: should say "the **expected** next arrival time is ..."

Eq 13: is the denominator needed?

Fig 3: x-tick labels needed to help appreciate timescale of this plot



**Strengths And Weaknesses:**


Strengths
---------

* comprehensive benchmark of recent neural TPP approaches, with reasonably thorough and systematic experimental practices
* focus on calibration was welcome, I wish more works would assess this
* use of Critical Distance diagrams was appreciated
* qualitative evaluation in Fig 3 helps provide qualitative insight (fast rise in $\lambda$ after an event indicates flexibility of neural models)

Weaknesses
----------

I list short phrases about weaknesses here, with longer discussion further down this review form.

- W1: Why include datasets you identify as borderline inadequate?

- W2: Computational time tradeoffs not assessed at all

- W3: Tough to parse main messages from Tables

- W4: Presentation quality of math background (Sec 2) could be improved (see detailed comments below), to make more accessible to readers new to TPPs

---

> ### Author Response · Authors · 2023-05-05
> **Response to reviewer ZZrh**
>
> Thank you for your valuable feedback and critical questions. We give clarifications to your questions and comments below. We have revised the manuscript according to your recommendations. All changes with respect to the first version are highlighted in blue in the revised version.
>
> **W1. Why include datasets you identify as borderline inadequate?**
>
> While we indeed found that these datasets might not be appropriate for benchmarking neural TPP models, we still wanted to consider them for evaluation as they are frequently used benchmarks in the TPP literature. In the Appendix, we added a version of Tables 4, 5 and 6 that do not include MIMIC2 and Stack Overflow. We also included a version of Tables 7, 8, 9 that do not include Taxi, Reddit Submissions, Reddit Comments, Yelp Toronto and Yelp Mississauga. We found no significant differences in the results that might have an impact on the conclusions of Section 6. Nonetheless, you are right that readers should be more carefully warned about the potential inadequacy of these datasets. We have added a statement at the beginning of Section 6 in the updated version to make it more explicit.
>
> **W2. Computational time trade-offs not assessed at all.**
>
> In Section 6, we added a Table and a discussion summarizing the average time (in seconds) required for a decoder to perform a single forward and backward pass on the MOOC dataset. For all decoders, we used a single Nvidia RTX A5000 GPU, and the batch size was 32. We found the LogNormMix decoder to provide the best performance/running time trade-offs among the considered baselines with an average epoch time of 2.05 seconds.
>
> **W3. Tough to parse main message from tables.**
>
> We understand that Tables 4 and 5 are difficult to interpret, but after several attempts to find a more appropriate presentation, we found these Tables to convey the most information in the most compact form. We agree that critical difference diagrams allow a more intelligible interpretation of the results, but they only inform us about the average rank differences between methods (and their statistical significance), while Tables 4 and 5 report additional information in the terms of average mean and median results for each metrics. In the Appendix, we added Tables 4 and 5 converted into critical difference diagrams. Each Table amounts to 35 critical difference diagrams in total, which we found harder to interpret.
>
>
> **W4. Requested presentation changes (Critical)**
>
> 1) We adjusted the text around equations 1-3 to make Section 2 more intelligible for someone who is familiar with probabilistic modeling, but not necessarily with Temporal Point Processes. In particular, we followed your recommendation by defining first $f(t,k|\mathcal{H}_t) = f(t|\mathcal{H}_t)p(k|t,\mathcal{H}_t$) before introducing the mark-wise conditional intensity function.
>
> 2) The typo has been corrected.
>
> 3) We added the demonstration to retrieve the density from the intensity in Appendix E. Additionally, while we mentioned that the MCIF could be heuristically interpreted as a probability around Eq. 3, the fact that it doesn't integrate to 1 in $R^{\lambda}2$ is due to the information on which this probability is conditioned changing as we move forward in time, i.e. the interval $[t_{i-1}, t]$ is function of $t$. Hence, by integrating the MCIF over $[t_{i-1}, \infty[$, we are actually integrating an infinity of instantaneously defined conditional probabilities, which is not required to equal 1.
>
> 4) As pointed by a fellow reviewer, this expression is ill-defined for any process that is not Poisson, as we cannot express the expected number of events as a function that depends on the observed arrival times. We removed it in the updated version.
>
> **Requested presentation changes (Requested)**
>
> All requested changes have been applied in the updated version.

---

> ### Comment · Reviewer_ZZrh · 2023-06-02
> **Update thoughts after recent revision**
>
> I never saw a comment from the authors about my original review, but looking through the revised PDF now (June 2), I see that there were in fact many changes to address my comments.
>
> # W1: Why include datasets that are identified as borderline inadequate
>
> I appreciate the new disclaimer text at the top of Sec 6.
>
> > For completeness, we added the results of the aggregation procedure with the aforementioned datasets discarded in Appendix D. We found no significant differences in the results.
>
> I think my concern is resolved **except for a typo**: I think Appendix D should be F, since in present revision the title of F is "Results on relevant datasets only"
>
> # W2: Computation times
>
> A new table (Table 7) has been added, which nicely illustrates how LNM has good runtime-performance, while NH is the most costly
>
> Thank you authors!
>
> # W3 Tough to parse main messages in the Tables
>
> Unfortunately does not look like there was anything done about this. I still think Tables 4/5 should be updated to help reviewers more easily access the major takeaway messages.
>
> # W4 Presentation concerns
>
> Looks like most of my requests have been updated suitably.
>
> Regarding the MCIF, I think the revisions are an improvement, but the concept is still tough to grasp from the present equations/text. I wonder if right after Eq 1, you might add an explanation of the output numerical range of the MCIF. Feels odd to encourage readers to "heuristically interpreted as the probability of observing the next event of mark k in an infinitesimal interval around t", if this really isn't a probability because f / 1-F can be arbitrarily larger than 1.

---

> > ### Author Response · Authors · 2023-06-06
> > **Reply to reviewer ZZrh's comment**
> >
> > Dear Reviewer ZZrh,
> >
> > Thank you for bringing this to our attention. While we did post a comment to your review on May 5, we now realize that it was not made public. We apologize for the oversight. Please find our original reply above.
> >
> > We appreciate your understanding, and if you have any further questions or concerns, please feel free to reach out. Thank you again for your feedback.

---

### Review · Reviewer_prAn · 2023-04-21

**Summary Of Contributions:**

This work performs a large-scale study of neural TPP models on various predictive tasks related to continuous-time event data.

The paper starts with an overview of the existing approaches, describing the important architectural choices when defining TPP models. These choices include representation of events, history encoding and time/mark decoders.

Next, a large-scale evaluation on 15 real-world datasets is performed to investigate the effects of these architectural choices on the predictive performance. Besides the standard evaluation metrics such as time/mark negative log-likelihood (NLL), the models are evaluated based on the calibration of their predictions. Finally, additional analysis produces two surprising findings: (1) in many cases the models achieve competitive results even without considering most of the event history and (2) many of the benchmark datasets used in the literature can be equivalently using trivial baseline models.

**Audience:**

Yes

**Claims And Evidence:**

Yes

**Requested Changes:**

1. Clarify the scope & setting:
	1. Mention that the paper only considers categorical marks, while other choices are also possible.
	2. Only the problem of mark prediction *conditioned on time* is considered. This means, the evaluation does not consider the question "What will be the type of the next event?" (page 6), but rather "What will be the type of the next event, given that it occurs at a certain time?"
2. Equation 6: The statement is only true for Poisson processes, where the compensator does not depend on the history. However, in the general case the right-hand side is ill-defined: the arrival times $t_1, ..., t_{i-1}$ are random variables, so we cannot express the expected # of events at time $t$ as a function that depends on the observed arrival times. If we decided to condition on the observed arrival times $t_1, ..., t_{i-1}$, we would end up with the following expression $$\mathbb{E}[N(t) | t_1, ..., t_{i-1}] = i - 1 + \Pr(t_i \in [t_{i-1}, t] | t_1, ..., t_{i-1}),$$ which is not very insightful.
4. The discussion in paragraph **Parametrization of TPP models and Learning** is only valid if we assume that we are dealing with non-terminating TPPs - for a terminating TPP assumptions $R_2^\lambda$, $R_2^f$, $R_3^\Lambda$ are violated. It's worth noting that this is the setting considered in the paper. We need to include the termination probability (i.e., violate these assumptions) if we model finite sequences of events on $[0, \infty)$ (instead of a bounded interval $[0, T]$, as in the paper), since otherwise any finite sequence would have the NLL of $-\infty$.
5. Scaling of arrival times to the interval $[0, 10]$ in the experiments (page 18): Are the arrival times scaled individually for each sequence in the dataset? What if different sequences are observed on intervals of different length $T$? In such case, the chosen scaling procedure can distort information - for example, if all sequences were generated by the same renewal process on intervals of different length, the ground truth renewal process would not perfectly model the rescaled event sequences. The normalization procedure should be described in more detail.


Minor comments:
- Page 2: solemnly -> solely
- Before Equation 15: inter-arrival times
- Paragraph **Encoding the mark**: The two works mentioned at the start of the paragraph also use learned embeddings for each mark, and are equivalent to the remaining approaches discussed in the paragraph (since one-hot encoding followed by a linear layer is equivalent to an embedding obtained from a learnable embedding matrix).
- Equation 18: The initial state $h_1$ is either fixed (e.g., all zeros) or is a learnable parameter, but usually doesn't have an associated distribution. If we assume such a distribution, we would need to perform poster inference over the state (e.g., with variational inference) and thus won't be able to compute the log-likelihood in a closed form.
- Equations 23, 25, 26: Potentially use notation $\lambda_k^*$ instead to highlight that the rate is constant over $[t_{i-1}, t]$.
- Equation 27: Highlight that the state $\boldsymbol{e}$ depends on the current time $t$.
- Equation 28: $\approx$ instead of =
- Section 5.1: Why are "raw inter- arrival times" abbreviated to TO, and "logarithms of inter-arrival times" abbreviated to LTO?
- Page 23: Potentially worth point out that the CONS encoder effectively reduces any neural TPP model to a renewal process.
- Page 25, last paragraph: Does Hawkes outperform approaches due to the fact that it's the only decoder that does not assume conditional independence between time/type of the event?

**Strengths And Weaknesses:**

This is an outstanding work: the overview in sections 2 and 3 can serve as a definitive reference on the topic, the experimental evaluation raises the bar in terms of empirical rigor compared to existing works, and the findings of the paper have important implications for future research in the field of neural TPPs.

- **Detailed survey**: The paper provides a unified description of the existing neural TPP architectures using consistent notation, which will be a useful reference for future work.
- **Significance**: The paper sheds light on the importance of various design choices in neural TPP architectures. This is interesting from the research perspective, evaluating the progress of the field by comparing proposed methods in a unified setup and highlighting promising future directions. This can also open the door for new applications of neural TPPs, helping practitioners pick among the many existing models. The findings on the inadequate quality of many datasets currently used in the literature will hopefully lead to development of more challenging benchmarks inspired by real-world applications.
* **Empirical rigor**: Multiple aspects of the experimental evaluation in the paper make it stand out among prior works on neural TPPs. These include:
	* Including various sanity-check baselines to provide a reference point for the performance of more sophisticated models
	* Performing visual analysis of the data (Figure 2) and individual model outputs (Figure 3) to better understand the reasons for good/bad performance of various models
	* Using a large number of datasets (15)
	* Using significance tests when reporting the results, instead of simply considering the mean score
	* Novel task - investigating the calibration of the probabilistic predictions produced by the models
* **Presentation**: Despite its length, the paper is easy to follow and the concepts build on top of each other. The experimental results are presented clearly, and a large set of experiments are summarized with a few key findings.

I haven't found major weaknesses in this paper, except several partially incorrect statements that need to be adjusted. These are listed below under "requested changes". One aspect of the evaluation procedure (normalization of the data) can have a significant effect on the results, but is not described clearly in the paper.

---

> ### Author Response · Authors · 2023-05-05
> **Response to reviewer prAn**
>
> Thank you for sharing your valuable insights on Temporal Point Processes. They greatly helped us to improve the manuscript. We summarize below the updates we made in the paper according to your concerns/recommendations. All changes with respect to the first version are highlighted in blue in the revised version.
>
> **Requested presentation changes (Critical)**
>
> 1.1 Indeed, we added a comment in the section 'Background and notations' to account for different types of marks.
>
> 1.2 We clarified the setting in the text.
>
> 2\. Thank you for pointing that out. There was indeed a mistake. However, we were not able to recover the expression you provided in the conditional case. We would be grateful if you could provide us with more details regarding your derivations. See Appendix E for our attempt.
>
> 3\. We wanted to exclusively focus on the non-terminating processes setting as it is the default setting considered in most of the neural TPP literature. We added a statement in the paragraph Parametrization of TPP models and Learning to make it explicit, and to warn readers that some assumptions are violated in the terminating case.
>
> 4\. Regarding the scaling procedure, each event $t_i$ is scaled as $t_{i,scaled} = 10 \times \frac{t_i}{t_{max}}$, where $t_{max}$ is the maximum observed timestamp among all sequences in the dataset. Consequently, as $\frac{10}{t_{max}}$ is constant for a given dataset, the relative distance between all events is maintained after rescaling, and the ground truth process should perfectly model the rescaled event sequences. We described the rescaling procedure in more details in the updated version.
>
> **Minor Comments**
>
> We applied all recommendations and corrected all the typos you mentioned in the updated version. Regarding the statement in the paragraph Encoding the Mark, you are right in the sense that these approaches are equivalent. We modified the text accordingly.
>
> *Why are "raw inter- arrival times" abbreviated to TO, and "logarithms of inter-arrival times" abbreviated to LTO?*
>
> TO is a shorthand for 'Times only', while 'LTO' is a shorthand for 'Log-times only'. We used these notations in the early stages of the paper, and we kept it throughout.
>
> *Does Hawkes outperform approaches due to the fact that it's the only decoder that does not assume conditional independence between time/type of the event?*
>
> Most decoders (MLP/MC, FNN, SA/MC, SA/CM, NH) do not assume conditional independence between arrival times and marks, yet we found them to perform significantly worse on the mark prediction task than the ones that make the aforementioned assumption (LNM, RMTPP). Trying to better understand this behavior is part of our future inquiries.

---

> > ### Comment · Reviewer_prAn · 2023-06-07
> > **Response to the authors**
> >
> > Thank you for your response.
> >
> > Regarding 2: As far as I know, $\mathbb{E}[N(t)]$ cannot be computed analytically for general TPPs. This can be done for Poisson processes ($\mathbb{E}[N(t)] = \Lambda(t)$) or, in the limit case, for [renewal processes](https://www.sciencedirect.com/topics/mathematics/elementary-renewal-theorem), but even for Hawkes process the calculations become rather involved (see, e.g., Equation 5 of [Zhang et al., ICML 2022](https://proceedings.mlr.press/v162/zhang22a.html)). I am not aware of any such results for neural TPPs.
> >
> > My statement referred to the fact that if we condition on the fact that **exactly** $i-1$ events occurred before time $t$, then $N(t)$ can be equal either to $i-1$ or $i$. I don't think this is a very insightful observation, and I would not include it in the paper.

---

> > > ### Author Response · Authors · 2023-06-07
> > > **Response to reviewer prAn**
> > >
> > > Thank you for your clarifications regarding point 2. We agree that it lacks insight and will be omitted from the Appendix in the revised version.

---

### Review · Reviewer_iQsN · 2023-05-08

**Summary Of Contributions:**

The paper benchmarks a wide range of state-of-the-art neural temporal point process models on a variety of datasets for a mixture of different tasks – flagging which architecture combinations work well on specific datasets. Furthermore, an in-depth analysis on the calibration of Neural TPP models is performed, finding that mark distributions for most models are poorly calibrated.

**Audience:**

Yes

**Broader Impact Concerns:**

--

**Claims And Evidence:**

No

**Requested Changes:**


1.	Detailed clarification is required on the precise prediction problem that we’d like to solve, along with the specific benchmark metric we should be using to rank models on it. While marked TPPs can forecast time-to-event problems and dynamic classification problems separately, there are other model types which can be used to solve each separately as well (e.g. time-to-event models and standard classifiers). With regards to NLL metrics, how does this reflect the predictive accuracy of the model, and does this account for the sparsely observed labels (in time)? Assuming the goal is to forecast both jointly, is time prediction or event prediction more important?

2.	Assuming that performance does vary based on dataset type, which models are more suited to what dataset characteristics? Can this be backed up by a data analysis of the underlying datasets used?

3.	Counter-intuitive results – a) why do neural forecasting models underperform in some datasets? b) why do self-attention architectures underperform recurrent ones? Seems to run contrary to findings of Transformer Hawkers Process (ICML 2020) where recurrent marked TPPs underperforms THPs.

4.	Related to the above, are 5 random search iterations enough for hyperparameter optimisation? Should learning rates be tuned and what regularisation (e.g. dropout) used on the models? How do results look if the settings from the original papers are used?

5.	I might be missing something, but are discrete event labels and inter-arrival times the only inputs into the model?


**Strengths And Weaknesses:**

Strengths
---

Given the complexity of many temporal point process models, it is helpful to understand when different architectural components should be deployed, and when they should be avoided.

Weaknesses
---

While the paper does provide an incredible amount of detail on empirical results (and what performs where), it is slightly lacking with regards to distilling insights into the various architectural components – particularly for counter-intuitive results, and best practices on what pipeline to deploy on a brand new dataset/problem. In addition, the paper also falls short of cleanly benchmarking predictive performance across neural TPPs, given the very dataset-specific outcomes, with no attempts to reconcile the underperformance on weaker datasets beyond saying that they should be avoided for neural TPP models. Main questions/improvements described below:

---

> ### Author Response · Authors · 2023-05-13
> **Response to reviewer iQsN - Part 1**
>
> Thank you for your review. We address your concerns below:
>
> 1\) As detailed in Section 2 and Section 5.4, the prediction problem we are trying to solve is twofold: modeling the distribution of future arrival-times through $f^\ast(\tau)$, and the distribution of future marks, through $p^\ast(k|t)$, conditional on the process' history. While we do not report point estimates for future arrival times (which could be obtained by taking the expectation as in Equation 12), we still generate point estimates for future marks through Equation 13, and we evaluate these point estimates on the F1-score. You are right that point estimates models could be used to solve both tasks separately, i.e. one for predicting the next inter-arrival time, and one for predicting the next mark. However, marked TPP are probabilistic models that allow to generate complete distributions over future trajectories, while point estimates models would only generate a single prediction for $\tau_i$. Such models could be trained to minimize the MSE/MAE, and would not need to model $f^\ast(\tau)$ to perform well with respect to these metrics. Additionally, point estimates models would not allow to measure uncertainty in the predictions, which is alternatively quantified by the NLL scores. The NLL is a proper scoring rule, meaning that is uniquely minimized (asymptotically) by the true joint distribution of arrival times and marks (Gneiting et al. [1]).
>
>  [1] Gneiting ,Tilmann and Balabdaoui,Fadoua and E. Raftery, Adrian, *Probabilistic forecasts, calibration and sharpness*, 2007, Journal of the Royal Statistical Society: Series B (Statistical Methodology).
>
> *Is time prediction or mark prediction more important?*
>
> This is very domain specific. From a research point of view, we believe that both tasks are equally important.
>
> 2\) We added in Appendix B a figure showing the standardized NLL-T and NLL-M (using Equation 63) of the decoders of Table 6, for each marked dataset separately. We found no trends in the NLL-T or NLL-M metrics that could be directly linked to a particular dataset's characteristic. For instance, LastFM, Wikipedia and MOOC all contain a high number of marks that are evenly spread out (see Figure 2). While the Hawkes decoder performs well with respect to the NLL-M on LastFM and Wikipedia, it underperforms LNM and RMTPP on MOOC. In other terms, it is difficult to infer a model's performance only from specific dataset's characteristics. Note that the main goal of our analysis was to pinpoint which combinations of models' components worked best on average across a wide range of real-world event sequence datasets.
>
> 3\)
>
> *Why do neural networks underperform?*
>
> Regarding time related metrics (NLL-T, ECE), the best results were still obtained from neural methods (while we classify LNM as a semi-parametric baseline, it still contains neural components). On the mark prediction task (measured by the NLL-M, ECE and F1-score), we indeed found the parametric Hawkes decoder to yield the best average results, while semi-parametric baselines often provided competitive performance. In essence, point processes are low dimensional data, and the fact that simpler models outperform more complicated ones was also recently highlighted in the related field of Temporal Graph Learning by Cong et al. [2].
>
> [2] Weilin Cong and Si Zhang and Jian Kang and Baichuan Yuan and Hao Wu and Xin Zhou and Hanghang Tong and Mehrdad Mahdavi, *Do We Really Need Complicated Model Architectures For Temporal Networks?*, 2023, ICLR.
>
> *Why RNN outperform SA mechanisms?*
>
> The code base of THP has been criticized for non-reproducibility issues and errors in the computation of the negative log-likelihood (https://github.com/SimiaoZuo/Transformer-Hawkes-Process/issues). Alternatively, Enguehard et al. [3] also ran experiments comparing an RNN and a SA history encoder, and found neither of these encoders to systematically outperform the other on the datasets they considered. Recall that our history encoder analysis is an average over multiple combinations of components and datasets. Particularly, we found the GRU encoder to be more stable to the choice of event encoding employed, while the performance of the SA encoder dropped when using the TO or LTO encodings (i.e. non vectorial representations of time). The results of Table 5 also take these underperforming combinations into account, which contributes to the superiority of the GRU encoding.
>
> [3] Joseph Enguehard and Dan Busbridge and Adam Bozson and Claire Woodcock and Nils Y. Hammerla, *Neural Temporal Point Processes For Modelling Electronic Health Records*, 2020, ML4H.

---

> > ### Author Response · Authors · 2023-05-13
> > **Response to reviewer iQsN - Part 2**
> >
> > 4\) More random search would have indeed allowed us to better explore the hyper-parameters space, leading to potentially better results for each model separately. However, given the high number of different models to train and to evaluate on 14 datasets (each made of 5 splits), using more than 5 random search would have quickly made the computational requirements unmanageable. Learning rates are dynamically tuned during training, i.e. starting at $10^{-3}$, the learning rate is halved if no improvement is observed in validation loss for 5 consecutive epochs. For a fair comparison, we didn't use dropout during training as most models didn't rely on it in their original implementation.
> >
> > *How do the results look if the original settings of the papers are used?*
> >
> > We re-trained RMTPP and LNM using the settings of the original papers, which in our nomenclature corresponds to GRU-RMTPP-CONCAT and GRU-LNM-CONCAT, respectively. To reproduce the THP model, we combined the SA encoder with the MLP/MC decoder with the TEMWL event encoding, corresponding to SA-MLP/MC-TEMWL (SA-MLP/MC in the table below). We used the hyper-parameter setting corresponding to 'Set 1' in the original paper for all datasets. Moreover, for a fair comparison, we also retrained the MLP/MC decoder with a GRU encoder (GRU-MLP/MC) using a comparable set of hyper-parameters, i.e. same number of encoder layers, number of hidden units and dropout rate. We did not retrain the FNN decoder as the original implementation did not define a valid non-terminating TPP. The results are consistent with our conclusions.
> > 1) With respect to the NLL-T, LNM outperforms all other baselines.
> > 2) There is no clear winner between the GRU and SA history encoder when equipped with the MLP/MC decoder.
> > 3) Semi-parametric baselines (RMTPP, LNM) display lower NLL-M compared to the non-parametric MLP/MC decoder.
> >
> > --------------- -------------- ---------------- ----------------- ----------------- ------------- ---------------
> >                                                                     NLL Time
> >                      LastFM             MOOC         Wikipedia         Github         MIMIC2       Stack O.
> >     SA-MLP/MC   -992.94 (40.93)   -228.15 (4.13)  -149.54 (18.08)  -231.48 (40.19)  1.67 (0.1)   -85.24 (1.32)
> >     GRU-MLP/MC  -995.31 (66.83)   -240.49 (3.68)  -101.36 (13.25)  -254.18 (50.54)  1.57 (0.05)  -86.59 (1.47)
> >     RMTPP       -1122.96 (48.83)  -146.06 (2.35)  -166.79 (23.78)  -273.68 (48.19)  1.67 (0.08)  -83.24 (1.4)
> >     LNM         -1358.26 (57.53)  -312.98 (3.33)  -290.32 (26.85)  -383.71 (60.14)  1.56 (0.08)  -90.7 (1.42)
> > -------------- --------------- ---------------- ----------------- ----------------- ------------- ---------------
> >
> >
> > ---------- ---------------- -------------- ----------------- ---------------- ------------- ---------------
> >                                                                     NLL Mark
> >                     LastfM           MOOC         Wikipedia       Github         MIMIC2        Stack O.
> >     SA-MLP/MC    878.3 (29.58)   86.67 (1.13)  381.06 (67.68)   141.21 (20.48)  2.64 (0.21)  106.94 (0.72)
> >     GRU-MLP/MC   757.35 (11.3)   88.4 (2.72)   409.32 (143.29)  145.38 (22.44)  3.8 (0.36)   105.81 (0.79)
> >     RMTPP        621.17 (16.63)  86.12 (1.19)  291.93 (39.43)   123.44 (15.98)  2.04 (0.15)  106.36 (0.67)
> >     LNM          647.53 (15.65)  82.87 (1.22)  281.64 (24.29)   118.54 (15.51)  2.04 (0.14)  106.46 (0.65)
> > ---------- ---------------- -------------- ----------------- ---------------- ------------- ---------------
> >
> > 5\) Yes, past inter-arrival times and discrete labels are the only input to the model when it comes to constructing the history embedding. At the decoder level, the current time is also fed to the model if the model is time-dependent.

---

### Comment · Action_Editors · 2023-05-19
**author discussion period**

Hi all,

Thanks to the reviewers and authors for kicking off the discussion period.  Would like to encourage reviewers and authors to continue engaging before the recommendation period begins next week.

Thanks!

---

### Author Response · Authors · 2023-06-29
**Camera-ready version.**

We thank the reviewers and the action editor for their constructive feedback during the review process, which contributed to enhancing the quality of our paper. We have now uploaded the camera-ready version, addressing the suggestions provided. In particular, we have incorporated the recommended key insights into the captions of Tables 4 and 5 to assist readers in extracting the main messages. Additionally, we have included a link to our code base within the paper.

---

### Decision · Action_Editors · 2023-06-16

**Recommendation:** Accept as is

**Comment:**

Reviewers agreed that the paper is well written, and its claims empirically supported, and quite detailed.  Reviewers also agreed that this work is of interest to the TMLR audience, and that this work is a valuable survey of neural temporal point process methods -- we recommend it for a survey certification.

A lingering criticism is the translation of empirical evaluations and comparison into insights about methodological choices.  Tables 4 and 5 remain difficult to parse -- perhaps the caption could highlight a few important points or insights that the results in the table reveal.

**Audience:**

Yes

**Claims And Evidence:**

Yes